# Enhancing the Transferability of Adversarial Examples via a Few Queries and Fuzzy Domain Eliminating

## Abstract

Due to the vulnerability of deep neural networks, the black-box attack has drawn great attention from the community. Though transferable priors decrease the query number of the black-box query attacks in recent efforts, the average number of queries is still larger than 100, which is easily affected by the query number limit policy. In this work, we propose a novel method called query prior-based method to enhance the attack transferability of the family of fast gradient sign methods by using a few queries. Specifically, for the untargeted attack, we find that the successful attacked adversarial examples prefer to be classified as the wrong categories with higher probability by the victim model. Therefore, the weighted augmented cross-entropy loss is proposed to reduce the gradient angle between the surrogate model and the victim model for enhancing the transferability of the adversarial examples. In addition, the fuzzy domain eliminating technique is proposed to avoid the generated adversarial examples getting stuck in the local optimum. Specifically, we define the fuzzy domain of the input example $x$ in the $\epsilon$-ball of $x$. Then, temperature scaling and fuzzy scaling are utilized to eliminate the fuzzy domain for enhancing the transferability of the generated adversarial examples. Theoretical analysis and extensive experiments demonstrate that our method could significantly improve the transferability of gradient-based adversarial attacks on CIFAR10/100 and ImageNet and outperform the black-box query attack with the same few queries.

## 1 Introduction

Deep Neural Network (DNN) has penetrated many aspects of life, e.g. autonomous cars, face recognition and malware detection. However, the imperceptible perturbations fool the DNN to make a wrong decision, which is dangerous in the field of security and will cause significant economic losses. To evaluate and increase the robustness of DNN, the advanced adversarial attack methods need to be researched. In recent years, the white-box attacks make a great success and the black-box attacks make great progress. However, because of the weak transferability (with the low attack strength) and the large number of queries, the black-box attacks can still be further improved.

Recently, a number of transferable prior-based black-box query attacks have been proposed to reduce the number of queries. For example, Cheng et al. (2019) proposed a prior-guided random gradient-free (P-RGF) method, which takes the advantage of a transfer-based prior and the query information simultaneously. Yang et al. (2020) also proposed a simple baseline approach (SimBA++), which combines transferability-based and query-based black-box attacks, and utilized the query feedback to update the surrogate model in a novel learning scheme. However, the average query number of the most query attacks is larger than 100 in the evaluations on ImageNet. In this scenario, the performance of these query attacks may be significantly affected when the query number limit policy is applied in the DNN application.

Besides, many black-box transfer attacks have been proposed to enhance the transferability of the adversarial examples, e.g. fast gradient sign method (FGSM) (Goodfellow et al., 2015), iterative FGSM (I-FGSM) (Kurakin et al., 2017), momentum I-FGSM (MI-FGSM) (Dong et al., 2018), diverse input I-FGSM (DI-FGSM) (Xie et al., 2019), scale-invariant Nesterov I-FGSM (SI-NI-FGSM)

(Lin et al., 2020) and variance-tuning MI-FGSM (VMI-FGSM) (Wang & He, 2021). Zhang et al. (2022a) also proposed the relative cross-entropy loss (RCE) to enhance the transferability by maximizing the logit's rank distance from the ground-truth class. However, these transfer attacks achieve weak transferability of adversarial examples under the constraint of low attack strength.

Therefore, to solve the above problems, we make the following contributions:

- First, we propose the query prior-based attacks to enhance the transferability of adversarial examples with few queries under the constraint of low attack strength. Specifically, we find that: (i) The better the transferability of the transfer black-box attack, the smaller the gradient angle between the surrogate model and the victim model. (ii) The successful attacked adversarial examples prefer to be classified as the wrong categories with higher probability by the victim model. Based on the aforementioned findings, the weighted augmented cross-entropy (WACE) loss is proposed to decrease the gradient angle between the surrogate model and the victim model for enhancing the transferability of adversarial examples, which is proved in Appendices A.4 and A.5. The proposed query prior-based method enhances the transferability of the family of FGSMs by integrating the WACE loss and a few queries (this contribution is described in detail in Appendix C).

- Second, when the query prior is not achieved, the fuzzy domain eliminating technique is used to enhance the transferability of adversarial examples. Specifically, we explore the effectiveness of the temperature scaling in eliminating the fuzzy domain and propose the fuzzy scaling to eliminate the fuzzy domain. By combining the temperature scaling and fuzzy scaling, fuzzy domain eliminating based cross-entropy (FECE) loss is proposed to enhance the transferability of the generated adversarial examples. In addition, the weighted augmented fuzzy domain eliminating based cross-entropy (WFCE) loss, which consists of the WACE and FECE loss, can further enhance the transferability of adversarial examples.

- Third, theoretical analysis and extensive experiments demonstrate that: (i) On the premise of allowing query, the WACE loss is better than cross-entropy (CE) and RCE losses. (ii) The temperature scaling and fuzzy scaling can effectively eliminate a part of the fuzzy domain. (iii) Under the constraint of low attack strength, the query prior-based method and fuzzy domain eliminating technique can significantly improve the attack transferability of the family of fast gradient sign methods on CIFAR10/100 (Krizhevsky, 2009) and ImageNet (Russakovsky et al., 2015).

## 2 PRELIMINARIES

The family of FGSMs and the RCE loss are briefly introduced, which is helpful to understand our methods in Section 3 and is regarded as the baselines in Section 4.

### 2.1 FAMILY OF FAST GRADIENT SIGN METHODS

The methods mentioned in this section are referred as the black-box transfer attacks with the objective of enhancing the transferability of adversarial examples.

**Fast gradient sign method (FGSM)** (Goodfellow et al., 2015) is the first transfer attack, which generates the adversarial examples $x^{adv}$ by maximizing the loss function $L(x^{adv}, y_o; \theta)$ with a one-step update:

$$x^{adv} = x + \epsilon \cdot sign\left(\nabla_x L\left(x, y_o; \theta\right)\right) \tag{1}$$

where $\epsilon$ is the attack strength, $y_o$ is the ground truth, $\theta$ is the model parameters, $sign(\cdot)$ is the sign function and $\nabla_x L\left(x, y_o; \theta\right)$ is the gradient of the loss function w.r.t. $x$.

**Iterative FGSM (I-FGSM)** (Kurakin et al., 2017) is the iterative version of FGSM by applying FGSM with a small step size:

$$x_0 = x, \quad x_{t+1}^{adv} = Clip_x^\epsilon\left\{x_t^{adv} + \alpha \cdot sign\left(\nabla_x L\left(x_t^{adv}, y_o; \theta\right)\right)\right\} \tag{2}$$

where $Clip_x^\epsilon(\cdot)$ function restricts the generated adversarial examples to be within the $\epsilon$-ball of $x$.

**Momentum I-FGSM (MI-FGSM)** (Dong et al., 2018) integrates the momentum into I-FGSM to escape from the poor local maxima and enhance the transferability of adversarial examples:

$$g_{t+1} = \mu \cdot g_t + \frac{\nabla_x L\left(x_t^{adv}, y_o; \theta\right)}{\left\|\nabla_x L\left(x_t^{adv}, y_o; \theta\right)\right\|_1},$$

$$x_{t+1}^{adv} = Clip_x^\epsilon \left\{x_t^{adv} + \alpha \cdot sign\left(g_{t+1}\right)\right\} \tag{3}$$

where $g_t$ is the accumulated gradient at iteration $t$, and $\mu$ is the decay factor of $g_t$.

**Diverse inputs I-FGSM (DI-FGSM)** (Xie et al., 2019) applies random transformations $Tr(\cdot)$ to the input images at each iteration with probability $p$ instead of only using the original images to generate adversarial examples.

**Scale-invariant Nesterov I-FGSM (SI-NI-FGSM)** (Lin et al., 2020) integrates Nesterov Accelerated Gradient (NAG) into I-FGSM to leverage the looking ahead property of NAG, i.e. substitutes $x_t^{adv}$ in Eq. 3 with $x_t^{adv} + \alpha \cdot \mu \cdot g_t$, and build a robust adversarial attack. Due to the scale-invariant property of DNN, the scale-invariant attack method is also proposed to optimize the adversarial perturbations over the scale copies of the input images.

**Variance tuning MI-FGSM (VMI-FGSM)** (Wang & He, 2021) further considered the gradient variance to stabilize the update direction and escape from the poor local maxima instead of directly using the current gradient for the momentum accumulation:

$$g_{t+1} = \mu \cdot g_t + \frac{\nabla_x L\left(x_t^{adv}, y_o; \theta\right) + v_t}{\left\|\nabla_x L\left(x_t^{adv}, y_o; \theta\right) + v_t\right\|_1}, \tag{4}$$

$$v_{t+1} = \frac{1}{N} \sum_{i=1}^{N} \nabla_x L\left(x_{ti}^{adv}, y_o; \theta\right) - \nabla_x L\left(x_t^{adv}, y_o; \theta\right), \tag{5}$$

$$x_{t+1}^{adv} = Clip_x^\epsilon \left\{x_t^{adv} + \alpha \cdot sign\left(g_{t+1}\right)\right\}$$

where $v_{t+1}$ is the gradient variance as the $t$-th iteration, $x_{ti}^{adv} = x_t^{adv} + r_i$, $r_i \sim U[-(\beta \cdot \epsilon)^d, (\beta \cdot \epsilon)^d]$, and $U\left[a^d, b^d\right]$ stands for the uniform distribution in $d$ dimensions and $\beta$ is a hyperparameter.

## 2.2 RELATIVE CROSS-ENTROPY (RCE) LOSS

To escape from the poor local maxima, RCE loss (Zhang et al., 2022a) is a new normalized CE loss that guides the logit to be updated in the direction of implicitly maximizing its rank distance from the ground-truth class:

$$Softmax(z_i) = \frac{e_{z_i}}{\sum_{c=1}^{C} e^{z_c}}, \tag{6}$$

$$L_{CE}(x, y_o; \theta) = -\log Softmax(z_o), \tag{7}$$

$$L_{RCE}(x, y_o; \theta) = L_{CE}(x, y_o; \theta) - \frac{1}{C} \sum_{c=1}^{C} L_{CE}(x, y_c; \theta) \tag{8}$$

where $z_o$ is the logit of the ground truth label $y_o$, $C$ is the number of category, $y_c$ is the category with index $c$. Note that, Proposition 7 explains the effectiveness of the RCE loss from the perspective of our domain fuzzy eliminating in the targeted transfer attacks.

## 3 METHODOLOGY

In this section, the motivation is introduced first. Then, the weighted augmented cross-entropy (WACE) loss is proposed and a corresponding theoretical analysis is described. By combining the WACE loss with a few queries, our query prior-based method is mentioned. The fuzzy domain eliminating based cross-entropy (FECE) loss is proposed and its theoretical analysis is described. Finally, By combining the advantages of the WACE and FECE losses, the WFCE loss is proposed.

### 3.1 MOTIVATION

First, though the transferable prior-based black-box query attacks (Cheng et al., 2019; Yang et al., 2020) significantly reduce the query number, the average number of queries is still larger than 100.

---

**Algorithm 1** Query prior-based VMI-FGSM (QVMI-FGSM)

---

**Input:** The surrogate model $f$ with parameters $\theta_f$; the victim model $h$ with parameters $\theta_h$; the WACE loss $L_{WACE}$; an example $x$ with ground truth label $y_o$; the magnitude of perturbation $\epsilon$; the number of iteration $T$ and decay factor $\mu$; the factor $\beta$ for the upper bound of neighborhood and the number of example $N$ for variance tuning; the maximum number of queries $Q$ and number of the wrong top-n categories $\bar{n}$.

**Output:** An adversarial example $x^{adv}$.

$\alpha = \epsilon/T$

$g_0 = 0; v_0 = 0; x_0^{adv} = x$

**for** $t = 0 \to T - 1$ **do**

    Query the logit output of the victim model:

$$
\boldsymbol{Z}_h = \begin{cases} h\left(x_t^{adv}\right), & if\ Q \geq T \\ h\left(x_t^{adv}\right), & if\ Q < T \wedge t \in \left\{\lfloor \frac{T}{Q} \rfloor \cdot i \mid i = 0, 1, \cdots, Q-1\right\} \\ \boldsymbol{Z}_h, & if\ Q < T \wedge t \notin \left\{\lfloor \frac{T}{Q} \rfloor \cdot i \mid i = 0, 1, \cdots, Q-1\right\} \end{cases} \tag{9}
$$

    Calculate the gradient $g_{t+1} = \nabla_x L_{WACE}\left(x_t^{adv}, y_o; \theta_f, \boldsymbol{Z}_h, \bar{n}\right)$

    Update $g_{t+1}$ by variance tuning-based momentum:

$$
g_{t+1} = \mu \cdot g_t + \frac{\nabla_x L_{WACE}\left(x_t^{adv}, y_o; \theta_f, \boldsymbol{Z}_h, \bar{n}\right) + v_t}{\left\|\nabla_x L_{WACE}\left(x_t^{adv}, y_o; \theta_f, \boldsymbol{Z}_h, \bar{n}\right) + v_t\right\|_1}
$$

    Update $v_{t+1}$ by sampling $N$ examples in the neighborhood of $x$:

$$
v_{t+1} = \frac{1}{N} \sum_{i=1}^{N} \nabla_x L_{WACE}\left(x_{ti}^{adv}, y_o; \theta_f, \boldsymbol{Z}_h, \bar{n}\right) - \nabla_x L_{WACE}\left(x_t^{adv}, y_o; \theta_f, \boldsymbol{Z}_h, \bar{n}\right)
$$

    Update $x_{t+1}^{adv}$ by applying the sign of gradient $x_{t+1}^{adv} = Clip_x^{\epsilon}\left\{x_t^{adv} + \alpha \cdot sign\left(g_{t+1}\right)\right\}$

**end for**

$x^{adv} = x_T^{adv}$

**return** $x^{adv}$

---

The performance of these query attacks may be greatly affected by the query number limit policy of the DNN applications. On the contrary, we can use the results of a few queries as the priors to enhance the transferability of the black-box transferable attacks. Specifically, we find the preference of the attacked victim model (i.e., Proposition 2). Then a novel black-box transfer attack is designed to achieve higher transferability through the combination of the preference and the results of a few queries. Note that the detailed motivation is described in Appendix D.

Second, a common phenomenon occurs in the black-box transfer attacks: under the same attack strength, with the increase of the attack strength, the attack success rate (ASR) of the white-box attacks fastly converges to 100%, but the ASR of the black-box transfer attacks is slowly approaching 100%. This phenomenon shows that there is a fuzzy domain between the surrogate model and the victim model for the black-box attacks. The fuzzy domain is a locally optimal region where the generated adversarial examples make the surrogate model wrong but the victim model still correct. Therefore, the fuzzy domain eliminating technique can enhance the transferability of the black-box transfer attacks. In this paper, the temperature scaling and fuzzy scaling are used to eliminate the fuzzy domain.

## 3.2 WEIGHTED AUGMENTED CROSS-ENTROPY LOSS

In this section, we first introduce the characteristics and preference of the victim model, and then propose the WACE loss based on the preference and give the theoretical analysis.

For the iterative gradient-based attacks, let $f$ and $h$ denote the surrogate model and the victim model, respectively. We use $\theta_f$ and $\theta_h$ to denote the parameters of the surrogate model and the victim model, respectively. In the following, Definitions 1 and 2 are mentioned to define the gradient angle between $f$ and $h$, and the top-n wrong categories and the top-n wrong categories attack success rate

(ASR) respectively, which are used in the introduction and Proofs of Propositions 1 and 2 to analyze the preference of the victim model. Propositions 1 and 2 are proved in Appendices A.1 and A.2.

**Definition 1** *(Gradient angle between the surrogate model and the victim model) For the $t$-th iteration adversarial example $x_t^{adv}$ of the surrogate model $f$, the angle between $\nabla_x L(x_t^{adv}, y_o; \theta_f)$ and $\nabla_x L(x_t^{adv}, y_o; \theta_h)$ is the gradient angle between $f$ and $h$ at iteration $t$.*

**Proposition 1** *When the step size $\alpha$ is small, the better the transferability of the transfer black-box attack, the smaller the gradient angle between the surrogate model and the victim model.*

**Definition 2** *(Top-n wrong categories and top-n wrong categories attack success rate (ASR)) For the example $(x, y_o)$, if the output of the victim model $h$ is $h(x)$, the top-n wrong categories are $\bar{n}$ number of categories with the largest value in $h(x)$ except the ground truth $y_o$, which is denoted as $\{y_{\tau_i} | i \leqslant \bar{n}\}$. The top-n wrong categories ASR denotes the accuracy of the adversarial example $x^{adv}$ classified as the wrong category in the top-n wrong categories.*

**Proposition 2** *When the victim model $h$ is attacked by the white-box gradient-based attacks, the successful attacked adversarial examples prefer to be classified as the wrong categories with higher probability (i.e. the top-n wrong categories $\{y_{\tau_i} | i \leqslant \bar{n}\}$). Meanwhile, the higher the probability of the wrong category, the more likely the adversarial example is to be classified as this category.*

Therefore, according to Propositions 1 and 2 (the details in Appendix A.3), for the untargeted attack, the weighted augmented CE (WACE) loss is proposed to enhance the transferability of the adversarial examples. Besides maximizing the loss function $L_{CE}(x^{adv}, y_o; \theta_f)$, the WACE loss also minimizes the loss function $L_{CE}(x^{adv}, y_{\tau_i}; \theta_f)$ where $y_{\tau_i}$ belongs to the top-n wrong categories $\{y_{\tau_i} | i \leqslant \bar{n}\}$:

$$L_{WACE}(x, y_o; \theta_f, \boldsymbol{Z}_h, \bar{n}) = L_{CE}(x, y_o; \theta_f) - \frac{1}{\bar{n}} \sum_{i=1}^{\bar{n}} w_i \cdot L_{CE}(x, y_{\tau_i}; \theta_f) \qquad (10)$$

$$w_i = \frac{e^{z_{h,\tau_i}}}{\sum_{j=1}^{\bar{n}} e^{z_{h,\tau_j}}} \qquad (11)$$

where $\boldsymbol{Z}_h = h(x) = [z_{h,1}, z_{h,2}, \cdots, z_{h,C}]$ is the query logit output of the victim model $h$ to $x$, $\bar{n}$ is the number of the top-n wrong categories. Note that $\sum_{i=1}^{\bar{n}} w_i = 1$, and the higher the logit value of the wrong category, the larger the weight $w_i$.

According to Proposition 1, the following Theorem 1 verified that the transferability of the transfer black-box attack based on the WACE loss is better than that based on the RCE and CE losses. Theorem 1 is proved in Appendices A.4 and A.5.

**Theorem 1** *The angle between $\nabla_x L_{WACE}(x_t^{adv}, y_o; \theta_f, \boldsymbol{Z}_h, \bar{n})$ and $\nabla_x L_{CE}(x_t^{adv}, y_o; \theta_h)$ is less than the angle between $\nabla_x L_{RCE}(x_t^{adv}, y_o; \theta_f)$ and $\nabla_x L_{CE}(x_t^{adv}, y_o; \theta_h)$, and the angle between $\nabla_x L_{CE}(x_t^{adv}, y_o; \theta_f)$ and $\nabla_x L_{CE}(x_t^{adv}, y_o; \theta_h)$.*

Propositions 1 and 2 and Theorem 1 are the theoretical analysis of the WACE loss, which explained the high transferability of the WACE loss-based attacks.

### 3.3 QUERY PRIOR-BASED ATTACKS

The family of fast gradient sign methods in Section 2.1 uses the CE loss. However, on the premise of allowing a few queries, the CE loss is replaced by the WACE loss in the family of fast gradient sign methods. Therefore, VMI-FGSM (Wang & He, 2021) is transformed into query prior-based VMI-FGSM, namely QVMI-FGSM, which is described in Algorithm 1 in detail. Specifically, two changes are made, compared with VMI-FGSM algorithm. First, the CE loss is replaced by our WACE loss. Second, according to Eq. 9, if the maximum number of queries $Q$ is greater than or equal to the number of attack iteration $T$, QVMI-FGSM queries the logit output of the victim model at each iteration, otherwise, QVMI-FGSM starts from 0 and performs equidistant query with $\lfloor \frac{T}{Q} \rfloor$ as the interval.

Similarly, FGSM (Goodfellow et al., 2015), I-FGSM (Kurakin et al., 2017), MI-FGSM (Dong et al., 2018), DI-FGSM (Xie et al., 2019) and SI-NI-FGSM (Lin et al., 2020) are transformed into Q-FGSM, QI-FGSM, QMI-FGSM, QDI-FGSM and QSI-NI-FGSM by combining the query priors.

### 3.4 FUZZY DOMAIN ELIMINATING BASED CROSS-ENTROPY LOSS

In this section, we first define the fuzzy domain in the untargeted attacks and targeted attacks, respectively. Then, the temperature scaling and fuzzy scaling are introduced. Finally, the FECE loss is proposed based on these two scaling techniques and gives the theoretical analysis.

In Definitions 3 and 4, $p$ is a probability threshold to identify whether the adversarial example $\hat{x}$ is locally optimal, $\hat{c}$ and $\tau$ are respectively the wrong category with the highest probability and the target category, and $p_{\hat{c}}$ and $p_{\tau}$ are their corresponding probability in the probability vector of the adversarial example $\hat{x}$ predicted by the surrogate model $f$, respectively. The ground truth of $x$ is $y_o$.

**Definition 3** *(The Fuzzy Domain in the untargeted attacks) In the spherical neighborhood $\mathcal{B}(x, \epsilon)$ with the input $x$ as the center and $\epsilon$ as the radius, the subdomain containing the local optimal region of the surrogate model $f$ is $\mathbb{A}_{f,-}(p) = \{\hat{x} | \hat{x} \in \mathcal{B}(x, \epsilon) \wedge p_{\hat{c}} < p\}$. On the contrary, the subdomain without the local optimal region is $\mathbb{A}_{f,+}(p) = \mathcal{B}(x, \epsilon) - \mathbb{A}_{f,-}(p) = \{\hat{x} | \hat{x} \in \mathcal{B}(x, \epsilon) \wedge p_{\hat{c}} \geqslant p\}$. For the victim model $h$, in the domain $\mathcal{B}(x, \epsilon)$, the subdomain with correct classification is $\mathbb{B}_{h,+} = \{\hat{x} | \hat{x} \in \mathcal{B}(x, \epsilon) \wedge arg\max h(\hat{x}) = y_o\}$, and the subdomain with wrong classification is $\mathbb{B}_{h,-} = \mathcal{B}(x, \epsilon) - \mathbb{B}_{h,+} = \{\hat{x} | \hat{x} \in \mathcal{B}(x, \epsilon) \wedge arg\max h(\hat{x}) \neq y_o\}$. Therefore, in the domain $\mathcal{B}(x, \epsilon)$, the fuzzy domain in the untargeted attacks (i.e., $\mathbb{M}^{NT}_{(x,f,h)}$ where $NT$ represents the non-targeted attacks) is the region that makes the surrogate model fall into the local optimum and the victim model classification correct:*

$$\mathbb{M}^{NT}_{(x,f,h)} = \mathbb{A}_{f,-}(p) \cap \mathbb{B}_{h,+} \tag{12}$$

**Definition 4** *(The Fuzzy Domain in the targeted attacks) In the spherical neighborhood $\mathcal{B}(x, \epsilon)$ with the input $x$ as the center and $\epsilon$ as the radius, the subdomain containing the local optimal region of the surrogate model $f$ is $\mathbb{A}_{f,-}(p) = \{\hat{x} | \hat{x} \in \mathcal{B}(x, \epsilon) \wedge p_{\tau} < p\}$. On the contrary, the subdomain without the local optimal region is $\mathbb{A}_{f,+}(p) = \mathcal{B}(x, \epsilon) - \mathbb{A}_{f,-}(p) = \{\hat{x} | \hat{x} \in \mathcal{B}(x, \epsilon) \wedge p_{\tau} \geqslant p\}$. For the victim model $h$, in the domain $\mathcal{B}(x, \epsilon)$, the subdomain classified as the target category $\tau$ is $\mathbb{B}_{h,+} = \{\hat{x} | \hat{x} \in \mathcal{B}(x, \epsilon) \wedge arg\max h(\hat{x}) = y_{\tau}\}$, and the subdomain classified as other categories is $\mathbb{B}_{h,-} = \mathcal{B}(x, \epsilon) - \mathbb{B}_{h,+} = \{\hat{x} | \hat{x} \in \mathcal{B}(x, \epsilon) \wedge arg\max h(\hat{x}) \neq y_{\tau}\}$. Therefore, in the domain $\mathcal{B}(x, \epsilon)$, the fuzzy domain in the targeted attacks (i.e., $\mathbb{M}^{Ta}_{(x,f,h)}$ where $Ta$ represents the targeted attacks) is the region that makes the surrogate model fall into the local optimum and is classified as other categories by the victim model:*

$$\mathbb{M}^{Ta}_{(x,f,h)} = \mathbb{A}_{f,-}(p) \cap \mathbb{B}_{h,-} \tag{13}$$

The recent researches (Dong et al., 2018; Xie et al., 2019; Lin et al., 2020; Wang & He, 2021) are trying to avoid stucking into the local optimum of the generated adversarial examples and make progress on the transferability. *Therefore, the local optimal region in $\mathbb{A}_{f,-}$ is closely related to $\mathbb{B}_{h,+}$ in the untargeted attacks and $\mathbb{B}_{h,-}$ in the targeted attacks.*

**Assumption 1** *Because the local optimal region in $\mathbb{A}_{f,-}$ is closely related to $\mathbb{B}_{h,+}$ in the untargeted attacks and $\mathbb{B}_{h,-}$ in the targeted attacks, eliminating the domain $\mathbb{A}_{f,-}$ can achieve the task of eliminating the fuzzy domain $\mathbb{M}^{NT}_{(x,f,h)}$ or $\mathbb{M}^{Ta}_{(x,f,h)}$.*

Based on Assumption 1, the temperature scaling and fuzzy scaling are used to eliminate $\mathbb{A}_{f,-}$. The temperature scaling was firstly proposed by Hinton et al. (2015) on knowledge distillation. The fuzzy scaling uses a penalty parameter $\mathcal{K}$ to apply to the logit of the correct category in the untargeted attacks ($\mathcal{K} > 1$) or the logit of the target category in the targeted attacks ($0 < \mathcal{K} < 1$). By combining the temperature scaling and fuzzy scaling, the fuzzy domain eliminating based cross-entropy (FECE) loss is proposed for the untargeted attacks:

$$FESoftmax(z_i; \mathcal{T}, \mathcal{K}) = \begin{cases} \frac{e^{\mathcal{K} \cdot z_o / \mathcal{T}}}{e^{\mathcal{K} \cdot z_o / \mathcal{T}} + \sum_{c=1 \wedge c \neq o}^{C} e^{z_c / \mathcal{T}}}, i = o \\ \frac{e^{z_i / \mathcal{T}}}{e^{\mathcal{K} \cdot z_o / \mathcal{T}} + \sum_{c=1 \wedge c \neq o}^{C} e^{z_c / \mathcal{T}}}, i \neq o \end{cases} \tag{14}$$

$$L_{FECE}(x, y_o; \theta, \mathcal{T}, \mathcal{K}) = -\log FESoftmax(z_o; \mathcal{T}, \mathcal{K}) \tag{15}$$

where $\mathcal{T}$ is the temperature parameter in the temperature scaling ($\mathcal{T} > 1$), FESoftmax is a fuzzy domain eliminating based Softmax. For the targeted attacks, the ground truth category $y_o$ replaces as $y_{\tau}$ in Equations 14 and 15.

Based on Assumption 1, Propositions 3 and 4 prove that the temperature scaling ($\mathcal{T} > 1$ and $\mathcal{K} = 1$) can eliminate a part of the fuzzy domain in the untargeted attacks and targeted attacks, respectively. Propositions 5 and 6 prove that the fuzzy scaling can eliminate a part of the fuzzy domain in the untargeted attacks ($\mathcal{T} = 1$ and $\mathcal{K} > 1$) and targeted attacks ($\mathcal{T} = 1$ and $0 < \mathcal{K} < 1$) respectively. Note that Propositions 3, 4, 5 and 6 are proved in Appendix A.6.

**Proposition 3** *In the untargeted attacks, when $p > 0.5$, the temperature scaling ($\mathcal{T} > 1$ and $\mathcal{K} = 1$) can eliminate a part of the fuzzy domain $\mathbb{M}^{NT}_{(x,f,h)}$.*

**Proposition 4** *In the targeted attacks, when $p > 0.5$, the temperature scaling ($\mathcal{T} > 1$ and $\mathcal{K} = 1$) can eliminate a part of the fuzzy domain $\mathbb{M}^{Ta}_{(x,f,h)}$.*

**Proposition 5** *In the untargeted attacks, the fuzzy scaling ($\mathcal{T} = 1$ and $\mathcal{K} > 1$) can eliminate a part of the fuzzy domain $\mathbb{M}^{NT}_{(x,f,h)}$.*

**Proposition 6** *In the targeted attacks, the fuzzy scaling ($\mathcal{T} = 1$ and $0 < \mathcal{K} < 1$) can eliminate a part of the fuzzy domain $\mathbb{M}^{Ta}_{(x,f,h)}$.*

## 3.5 WEIGHTED AUGMENTED FUZZY DOMAIN ELIMINATING BASED CROSS-ENTROPY LOSS

To combine the advantages of the WACE and FECE losses, the weighted augmented fuzzy domain eliminating based cross-entropy (WFCE) loss is proposed:

$$L_{WFCE}\left(x, y_o; \theta_f, \boldsymbol{Z}_h, \bar{n}, \mathcal{T}, \mathcal{K}\right) =$$
$$L_{FECE}\left(x, y_o; \theta_f, \mathcal{T}, \mathcal{K}\right) - \frac{1}{\bar{n}} \sum_{i=1}^{\bar{n}} w_i \cdot L_{FECE}\left(x, y_o; \theta_f, \mathcal{T}, \mathcal{K}\right) \tag{16}$$

## 4 EXPERIMENTS

To validate the effectiveness of the proposed query prior-based attacks and the fuzzy domain eliminating technique, we conduct extensive experiments on CIFAR10/100 (Krizhevsky, 2009) and ImageNet (Russakovsky et al., 2015). The detailed experimental setup is described in Appendix B.1 and all experimental results show in Appendix B. In this section, we compare our method with competitive baselines under various experimental settings. Experimental results demonstrate that our method can significantly improve the transferability of the baselines. Finally, we provide further investigation on hyper-parameters $\bar{n}$ and $Q$ used for the query prior-based attacks, and $\mathcal{T}$ and $\mathcal{K}$ used for the FECE loss based transfer attacks. The detailed experimental analysis is shown in Appendix B. All experiments are run on a single machine with four GeForce RTX 2080tis and the deep learning framework is Pytorch.

### 4.1 COMPARISON WITH OR WITHOUT THE QUERY PRIORS ON THE UNTARGETED ATTACKS

**Attacking a naturally trained model.** As shown in Tables 1, 2, 3, 7, 8 and 9, when the attack strength $\epsilon = 8/255$, in comparison with different loss functions (the CE and RCE losses), the query prior-based attacks with the WACE loss can not only significantly improve the transfer attack success rate of the black-box setting but also improve the attack success rate of the white-box setting on different surrogate models and datasets.

**Attacking an adversarially trained model.** As shown in Tables 3 and 9, in comparison with different loss functions (the CE and RCE losses), the query prior-based attacks with the WACE loss can enhance the transferability of the gradient iterative-based attacks when attacking the adversarially trained model and the attack strength $\epsilon = 8/255$ on different surrogate models.

**Attacking the other models.** VMI-FGSM is selected as the baseline to compare the transferability to the other models where a surrogate model and a query model are used to generate adversarial examples to attack many other models. As shown in Tables 13, 14, 15, 16, 17 and 18, when the attack strength $\epsilon = 8/255$, in comparison with different loss functions (the CE and RCE losses), the query prior-based VMI-FGSM with the WACE loss can enhance the transferability of the generated adversarial examples on different surrogate models, query models and datasets (except for the case of the RCE loss in Table 16).

### 4.2 COMPARISON WITH OR WITHOUT THE FUZZY DOMAIN ELIMINATING TECHNIQUE ON THE UNTARGETED ATTACKS

**Attacking a naturally trained model.** As shown in Tables 1 and 2, in comparison with different loss functions (the CE and RCE losses), our FECE loss can significantly enhance the transferability of the gradient iterative-based attacks when attacking the naturally trained model and the attack strength $\epsilon = 8/255$ on different datasets (CIFAR10/100). Table 3 shows that our FECE loss can enhance the transferability of the latest gradient iterative-based attacks.

**Attacking an adversarially trained model.** As shown in Table 3, in comparison with different loss functions (the CE and RCE losses), our FECE loss can enhance the transferability of VMI-FGSM and keep (or slightly decrease) the transferability of the other attacks when attacking the adversarially trained model on ImageNet.

**Combination of the query priors and fuzzy domain eliminating technique in the untargeted attacks.** As shown in Tables 1, 2 and 3, when attacking the naturally trained model, in comparison with our WACE and FECE losses, our WFCE loss can further improve the transferability of the gradient iterative-based attacks on different datasets.

### 4.3 COMPARISON WITH CURRENT BLACK-BOX QUERY ATTACKS ON THE UNTARGETED ATTACKS

As shown in Tables 1, 2, 3, 7, 8 and 9, when the attack strength $\epsilon = 8/255$ and the allowed query number $Q = 10$, the attack success rate of our QVMI-FGSM is much larger than that of Square and PRGF when attacking the naturally trained model and adversarially trained model on different surrogate models and datasets (except for ImageNet with VGG16 as the surrogate model to attack the adversarially trained models when compared with Square).

### 4.4 ABLATION STUDY ON THE UNTARGETED ATTACKS

**Different numbers of the top-n wrong categories $\bar{n}$ and the query number $Q$.** Figures 5 and 6 respectively evaluate the effect of different $\bar{n}$ and $Q$ on the attack success rates of five naturally trained victim models and two adversarially trained victim models when these victim models are attacked by QI-FGSM ($\epsilon = 8/255$) with VGG16 for CIFAR10/100 and ImageNet. As shown in Figure 5, when $\bar{n}$ is greater than a certain threshold, the attack success rate will not be improved. As shown in Figure 6, the more the query, the greater the attack success rate.

**Different sizes of the penalty parameter $\mathcal{K}$ and the temperature $\mathcal{T}$.** Figures 9 and 10 respectively evaluate the effect of different $\mathcal{K}$ and $\mathcal{T}$ on the attack success rates of ResNet50 to VGG16 using various transfer attacks for CIFAR10/100 and ImageNet. As shown in Figure 9, with the increase of $\mathcal{K}$, the attack success rates of the gradient iterative-based attacks are significantly increased on CIFAR10 except for SI-NI-FGSM. Complementarily, as shown in Figure 10, with the increase of $\mathcal{T}$, the attack success rate of the SI-NI-FGSM is increased on CIFAR10, the attack success rates of all gradient iterative-based attacks are significantly increased on CIFAR100 and the attack success rates of the latest gradient iterative-based attacks (MI-FGSM, SI-NI-FGSM and VMI-FGSM) are increased by a reasonable $\mathcal{T}$ on ImageNet. Figure 11 further explores the optimal parameter combinations of $\mathcal{K}$ and $\mathcal{T}$ on different datasets, which are summarized in Table 30.

### 4.5 COMPARISON WITH OR WITHOUT THE FUZZY DOMAIN ELIMINATING TECHNIQUE ON THE TARGETED ATTACKS

As shown in Figure 12, slightly decreasing $\mathcal{K}$ from 1 can slightly increase the targeted attack success rates of several gradient iterative-based attacks on CIFAR10/100. As shown in Figure 13, *with the increase of the $\mathcal{T}$, the targeted attack success rates of almost all the FECE ($\mathcal{K} = 1$) based attacks are increased and close to that of the RCE based attacks* (Propositions 4 and 7 explain the result).

## 5 RELATED WORK

### 5.1 ADVERSARIAL ATTACKS

**Black-box Transfer Attacks** are divided into five categories, i.e. feature destruction-based attacks, gradient generation-based attacks, data augmentation-based attacks, model ensemble-based attacks, model specific-based attacks, respectively. The feature destruction-based attacks (Wu et al., 2020b; Inkawhich et al., 2019; 2020; Huang et al., 2019; Zhang et al., 2022b; Zhou et al., 2018; Ganeshan et al., 2019; Wang et al., 2021b) enhance the transferability of adversarial examples by destroying the features of the intermediate layers or critical neurons. The gradient generation-based attacks (Li et al., 2020a; Guo et al., 2020; Xie et al., 2019; Gao et al., 2020; Lin et al., 2020; Dong et al., 2018; Han et al., 2022; Wang & He, 2021) enhance the transferability of adversarial examples by changing the way of gradient generation. The data augmentation-based attacks (Wang et al., 2021a; Li et al., 2020b; Zou et al., 2020; Huang et al., 2021) increase the input diversity to enhance the transferability of adversarial examples. The model ensemble-based attacks (Liu et al., 2017; Li et al., 2020c; Xiong et al., 2021) use the common attention of various models to enhance the transferability of adversarial examples. The model specific-based attacks (Wu et al., 2020a) use the high transferability of some structures to enhance the transferability of adversarial examples, e.g. skip connection.

**Black-box Query Attacks** are divided into two categories, i.e., pure query attacks and transferable prior-based query attacks. The pure query attacks (Chen et al., 2017; Andriushchenko et al., 2020; Zhang et al., 2020) estimate the gradient or update the attack optimization model by querying the output of the victim model. Recently, Zhang et al. (2020) utilized the feedback knowledge not only to craft adversarial examples but also to alter the searching directions to achieve efficient attacks. The transferable prior-based query attacks (Cheng et al., 2019; Yang et al., 2020; Tashiro et al., 2020) use the prior knowledge of the surrogate model to decrease the query number. Recently, Tashiro et al. (2020) proposed Output Diversified Sampling to maximize diversity in the target model's outputs among the generated samples.

### 5.2 ADVERSARIAL DEFENSES

Adversarial training (Madry et al., 2018; Zhang et al., 2019; Wong et al., 2020; Pang et al., 2021) is the most effective method to defend against adversarial examples. Recently, Zhang et al. (2019) designed a new defense method to trade off the adversarial robustness against accuracy. Wong et al. (2020) discovered that adversarial training can use a much weaker and cheaper adversary, an approach that was previously believed to be ineffective, rendering the method no more costly than standard training in practice. Pang et al. (2021) investigated the effects of mostly overlooked training tricks and hyperparameters for the adversarially trained models.

## 6 CONCLUSION

Though transferable priors decrease the query number of the black-box query attacks, the average number of queries is still larger than 100, which is easily affected by the number of queries limit policy. On the contrary, we can utilize the priors of a few queries to enhance the transferability of the transfer attacks. In this work, we propose the query prior-based method to enhance the transferability of the family of FGSMs. Specifically, we find that: (i) The better the transferability of the transfer attack, the smaller the gradient angle between the surrogate model and the victim model. (ii) The successful attacked adversarial examples prefer to be classified as the wrong categories with higher probability by the victim model. Based on the above findings, the weighted augmented cross-entropy (WACE) loss is proposed to decrease the gradient angle between the surrogate model and the victim model for enhancing the transferability of adversarial examples. In addition, because the existence of the fuzzy domain makes it difficult to transfer the adversarial examples generated by the surrogate model to the victim model, the fuzzy domain eliminating technique, which consists of the fuzzy scaling and the temperature scaling, is proposed to enhance the transferability of the generated adversarial examples. Theoretical analysis and extensive experiments demonstrate the effectiveness of the query prior-based attacks and fuzzy domain eliminating technique.

ETHICS STATEMENT

We do not anticipate any negative ethical implications of the proposed method. The datasets (CI-FAR10/100 and ImageNet) used in this paper are publicly available and frequently used in the domain of computer vision. The proposed method is beneficial to the development of AI security.

REPRODUCIBILITY STATEMENT

Appendix B.1 introduces the details experimental setup, including datasets, models, baselines and hyper-parameters. Most baselines are implemented in a popular pytorch repository Kim (2020). Python implementation of this paper and all baselines are available in the supplementary materials.

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

# A PROOFS

We provide the proofs in this section.

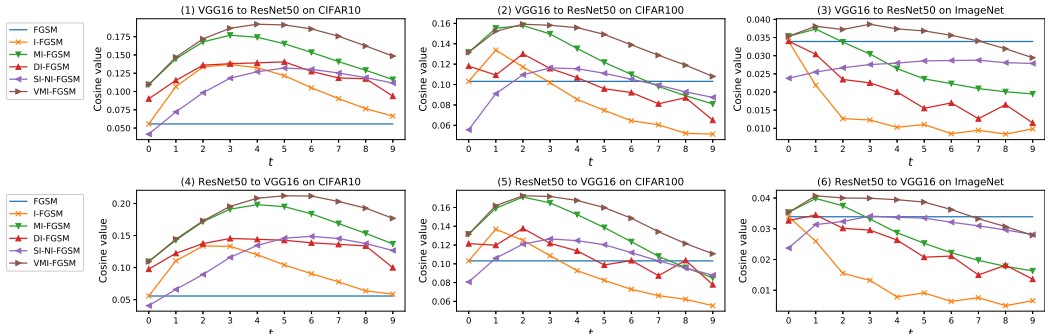

Figure 1: The cosine value of the gradient angle between the surrogate model and the victim model at each iteration when the surrogate model is attacked by different methods for CIFAR10/100 and ImageNet. For example, in subfigure (1), VGG16 as the surrogate model and ResNet50 as the victim model are attacked by different transfer attacks for CIFAR10. Note that the attack strength $\epsilon = 8/255$.

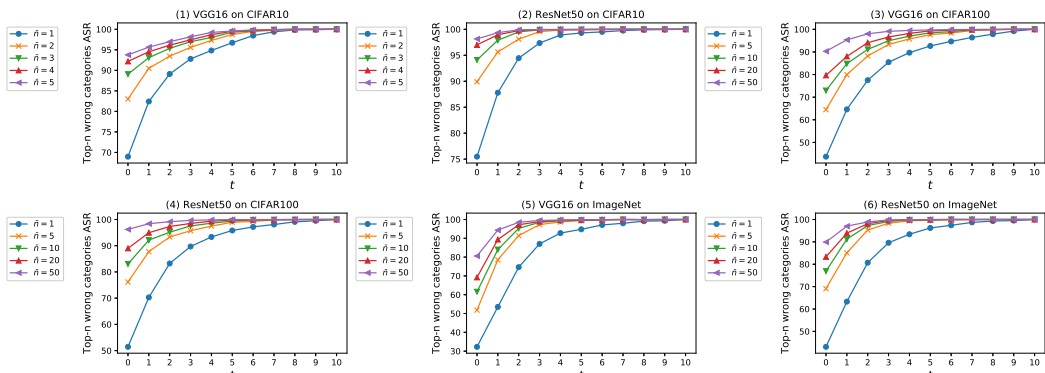

Figure 2: The top-n wrong categories attack success rate (ASR) (%) at each iteration $t$ when the model is attacked by I-FGSM (white-box setting) for CIFAR10/100 and ImageNet. For example, in subfigure (1), VGG16 is attacked by I-FGSM for CIFAR10. The successful attacked adversarial examples prefer to be classified as the top-n wrong categories. With the increase of the iteration, the top-n wrong categories ASR gradually increases and approaches 100%. Note that the attack strength $\epsilon = 8/255$.

## A.1 PROOF OF PROPOSITION 1

**Proposition 1:** *When the step size $\alpha$ is small, the better the transferability of the transfer black-box attack, the smaller the gradient angle between the surrogate model and the victim model.*

Note that we explore the relationship between the $\cos \vartheta$ ($\vartheta$ is the gradient angle between the surrogate model and the victim model) and the transferability on the same surrogate model and victim model pair using different transfer attack methods, but Liu et al. (2017) and Demontis et al. (2019) explore the relationship between the $\cos \vartheta$ and the transferability on the different surrogate and victim model pairs using the same transfer attack method.

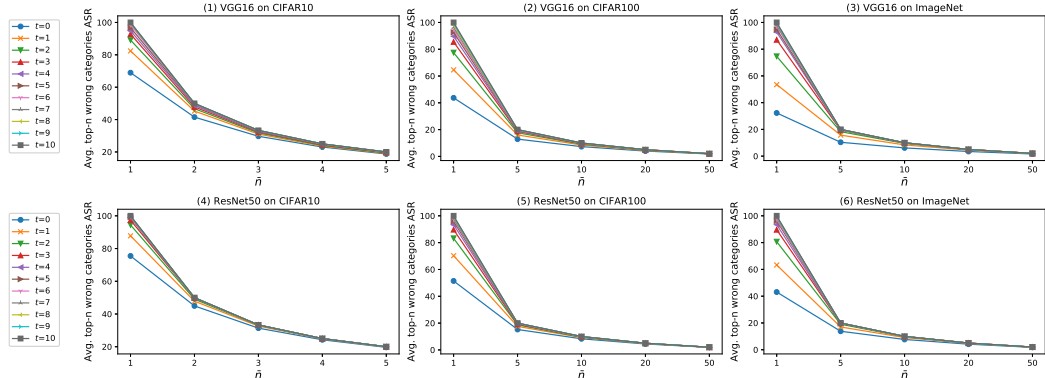

Figure 3: The average top-n wrong categories attack success rate (ASR) (%) at each iteration $t$ when the model is attacked by I-FGSM (white-box setting) for CIFAR10/100 and ImageNet. For example, in subfigure (1), VGG16 is attacked by I-FGSM for CIFAR10. The smaller $\bar{n}$, the higher the average top-n wrong categories ASR. Therefore, the higher the probability of the wrong category, the more likely the adversarial example is to be classified as this category. Note that the attack strength $\epsilon = 8/255$.

**Proof (Empirical Proof)** *To verify the correctness of Proposition 1, we compare the relationship between the attack success rates of the family of fast gradient sign methods and the cosine values (i.e., the average cosine values of the gradient angles between the surrogate model and the victim model at all iterations) of the family of fast gradient sign methods when the attack strength, number of iteration and step size are $\epsilon, T, \alpha = 8/255, 10, 0.8/255$. If the sort of the attack success rates is the same as the sort of the cosine values, Proposition 1 is correct with high confidence. Empirically, Proposition 1 is verified on different surrogate models and datasets as follows.*

*When VGG16 is the surrogate model and ResNet50 is the victim model for CIFAR10, Table 7 shows that the sort of the attack success rates is VMI-FGSM (76.60%) > MI-FGSM (70.75%) > SI-NI-FGSM (68.10%) > DI-FGSM (62.80%) > I-FGSM (61.45%) > FGSM (41.90%), and the Figure 1-(1) shows that the sort of the cosine values is also basically VMI-FGSM > MI-FGSM > SI-NI-FGSM > DI-FGSM > I-FGSM > FGSM.*

*When VGG16 is the surrogate model and ResNet50 is the victim model for CIFAR100, Table 8 shows that the sort of the attack success rates is VMI-FGSM (77.70%) > MI-FGSM (69.30%) > SI-NI-FGSM (64.35%) ≈ FGSM (63.15%) > DI-FGSM (59.30%) > I-FGSM (50.70%), and the Figure 1-(2) shows that the sort of the cosine values is also basically VMI-FGSM > MI-FGSM > SI-NI-FGSM ≈ FGSM > DI-FGSM > I-FGSM.*

*When VGG16 is the surrogate model and ResNet50 is the victim model for ImageNet, Table 9 shows that the sort of the attack success rates is VMI-FGSM (62.4%) > SI-NI-FGSM (56.6%) > MI-FGSM (46.5%) > DI-FGSM (38.1%) > FGSM (32.8%) > I-FGSM (27.8%), and the Figure 1-(3) shows that the sort of the cosine values is basically VMI-FGSM > FGSM > SI-NI-FGSM > MI-FGSM > DI-FGSM > I-FGSM.*

*When ResNet50 is the surrogate model and VGG16 is the victim model for CIFAR10, Table 1 shows that the sort of the attack success rates is VMI-FGSM (80.40%) > MI-FGSM (77.25%) > SI-NI-FGSM (73.00%) > DI-FGSM (67.65%) > I-FGSM (59.85%) > FGSM (44.20%), and the Figure 1-(4) shows that the sort of the cosine values is also basically VMI-FGSM > MI-FGSM > SI-NI-FGSM > DI-FGSM > I-FGSM > FGSM.*

*When ResNet50 is the surrogate model and VGG16 is the victim model for CIFAR100, Table 2 shows that the sort of the attack success rates is VMI-FGSM (84.40%) > MI-FGSM (77.35%) > SI-NI-FGSM (72.90%) > DI-FGSM (68.40%) > FGSM (64.80%) > I-FGSM (61.45%), and the Figure 1-(5) shows that the sort of the cosine values is also basically VMI-FGSM > MI-FGSM > SI-NI-FGSM > DI-FGSM > I-FGSM > FGSM.*

*When ResNet50 is the surrogate model and VGG16 is the victim model for ImageNet, Table 3 shows that the sort of the attack success rates is VMI-FGSM (69.1%) > SI-NI-FGSM (68.7%) > MI-FGSM (55.4%) > DI-FGSM (52.6%) > FGSM (42.6%) > I-FGSM (32.1%), and the Figure 1-(6) shows that the sort of the cosine values is also basically VMI-FGSM > FGSM > SI-NI-FGSM > MI-FGSM > DI-FGSM > I-FGSM.*

*In conclusion, by discussing the above six cases for CIFAR10/100 and ImageNet datasets, for the family of iterative fast gradient sign methods except for FGSM, Proposition 1 is correct with high confidence. Therefore, decreasing the cosine value of the gradient angle between the surrogate model and the victim model can enhance the transferability of adversarial examples.*

**Proof (Theoretical Proof)** *Assuming that the perturbation gradients of the surrogate model $f$ and the victim model $h$ at the attack iteration $t$ are $\nabla_x L(x_t^{adv}, y_o; \theta_f)$ and $\nabla_x L(x_t^{adv}, y_o; \theta_h)$, respectively, and $\vartheta$ is the angle of them. Then, the perturbation gradient of the surrogate model $f$ (i.e., $\nabla_x L(x_t^{adv}, y_o; \theta_f)$) is decomposed as the parallel component $\nabla_x^{\parallel} L(x_t^{adv}, y_o; \theta_f)$ and the vertical component $\nabla_x^{\perp} L(x_t^{adv}, y_o; \theta_f)$, which satisfied that*

$$\nabla_x^{\parallel} L\left(x_t^{adv}, y_o; \theta_f\right) \parallel \nabla_x L\left(x_t^{adv}, y_o; \theta_h\right) \tag{17}$$

$$\nabla_x^{\perp} L\left(x_t^{adv}, y_o; \theta_f\right) \perp \nabla_x L\left(x_t^{adv}, y_o; \theta_h\right) \tag{18}$$

$$\nabla_x L\left(x_t^{adv}, y_o; \theta_f\right) = \nabla_x^{\parallel} L\left(x_t^{adv}, y_o; \theta_f\right) + \nabla_x^{\perp} L\left(x_t^{adv}, y_o; \theta_f\right) \tag{19}$$

*For the victim model $h$, assuming that the variation of the loss funtion of the victim model $h$ caused by the perturbation $\nabla_x$ is*

$$\Delta_L^h\left(\nabla_x\right) = L\left(x_t^{adv} + \nabla_x, y_o; \theta_h\right) - L\left(x_t^{adv}, y_o; \theta_h\right) \tag{20}$$

*where $\Delta_L^h(\cdot)$ represents the variation of the loss function of the victim model $h$.*

*According to Lemma 1, for the victim model $h$, three properties (i.e., Equations 21, 22 and 23) are achieved.*

*For the first property, in the case of moving the same distance $\alpha$ (satisfied Lemma 1) along the gradient direction, the variation of the loss $L\left(x_t^{adv}, y_o; \theta_h\right)$ along the direction of the parallel component $\nabla_x^{\parallel} L(x_t^{adv}, y_o; \theta_f)$ is greater than that along the direction of the vertical component $\nabla_x^{\perp} L(x_t^{adv}, y_o; \theta_f)$, i.e.,*

$$\Delta_L^h\left(\alpha \cdot \frac{\nabla_x^{\parallel} L\left(x_t^{adv}, y_o; \theta_f\right)}{\left\|\nabla_x^{\parallel} L\left(x_t^{adv}, y_o; \theta_f\right)\right\|_2}\right) > \Delta_L^h\left(\alpha \cdot \frac{\nabla_x^{\perp} L\left(x_t^{adv}, y_o; \theta_f\right)}{\left\|\nabla_x^{\perp} L\left(x_t^{adv}, y_o; \theta_f\right)\right\|_2}\right) \tag{21}$$

*For the second property, the variation of the loss $L\left(x_t^{adv}, y_o; \theta_h\right)$ along the direction of the parallel component $\nabla_x^{\parallel} L(x_t^{adv}, y_o; \theta_f)$ is positively correlated with the moving distance $\alpha$, i.e.,*

$$\Delta_L^h\left(\alpha \cdot \frac{\nabla_x^{\parallel} L\left(x_t^{adv}, y_o; \theta_f\right)}{\left\|\nabla_x^{\parallel} L\left(x_t^{adv}, y_o; \theta_f\right)\right\|_2}\right) > \Delta_L^h\left((\alpha - \Delta_\alpha) \cdot \frac{\nabla_x^{\parallel} L\left(x_t^{adv}, y_o; \theta_f\right)}{\left\|\nabla_x^{\parallel} L\left(x_t^{adv}, y_o; \theta_f\right)\right\|_2}\right) \tag{22}$$

*where $\Delta_\alpha$ represents the variation of the moving distance $\alpha$.*

*For the third property, the degree of correlation between $\Delta_L^h\left(\alpha \cdot \frac{\nabla_x^{\parallel} L\left(x_t^{adv}, y_o; \theta_f\right)}{\left\|\nabla_x^{\parallel} L\left(x_t^{adv}, y_o; \theta_f\right)\right\|_2}\right)$ and $\alpha$ is greater than that between $\Delta_L^h\left(\alpha \cdot \frac{\nabla_x^{\perp} L\left(x_t^{adv}, y_o; \theta_f\right)}{\left\|\nabla_x^{\perp} L\left(x_t^{adv}, y_o; \theta_f\right)\right\|_2}\right)$ and $\alpha$, i.e.,*

$$\Delta_L^h\left((\alpha + \Delta_\alpha) \cdot \frac{\nabla_x^{\parallel} L\left(x_t^{adv}, y_o; \theta_f\right)}{\left\|\nabla_x^{\parallel} L\left(x_t^{adv}, y_o; \theta_f\right)\right\|_2}\right) + \Delta_L^h\left((\alpha - \Delta_\alpha) \cdot \frac{\nabla_x^{\perp} L\left(x_t^{adv}, y_o; \theta_f\right)}{\left\|\nabla_x^{\perp} L\left(x_t^{adv}, y_o; \theta_f\right)\right\|_2}\right) >$$

$$\Delta_L^h\left(\alpha \cdot \frac{\nabla_x^{\parallel} L\left(x_t^{adv}, y_o; \theta_f\right)}{\left\|\nabla_x^{\parallel} L\left(x_t^{adv}, y_o; \theta_f\right)\right\|_2}\right) + \Delta_L^h\left(\alpha \cdot \frac{\nabla_x^{\perp} L\left(x_t^{adv}, y_o; \theta_f\right)}{\left\|\nabla_x^{\perp} L\left(x_t^{adv}, y_o; \theta_f\right)\right\|_2}\right) \tag{23}$$

where $\Delta_\alpha$ represents the variation of the moving distance $\alpha$.

At the gradient attack iteration $t+1$, assuming that the generated adversarial example $x_{t+1}^{adv}$ in range $\mathcal{B}\left(x_t^{adv}, \alpha\right)$ (i.e., in the sphere with radius $\alpha$ centered on the adversarial example $x_t^{adv}$). Hence, for the surrogate model $f$, when the adversarial example $x_t^{adv}$ moves the allowed maximum distance $\alpha$ along the direction of the perturbation gradient $\nabla_x L(x_t^{adv}, y_o; \theta_f)$, the moving distance along the direction of the parallel component $\nabla_x^\parallel L(x_t^{adv}, y_o; \theta_f)$ is

$$\alpha^\parallel = \alpha \cdot \cos\vartheta, \tag{24}$$

the moving distance along the direction of the vertical component $\nabla_x^\perp L(x_t^{adv}, y_o; \theta_f)$ is

$$\alpha^\perp = \alpha \cdot \sin\vartheta \tag{25}$$

Then, we discuss the angle $\vartheta$ in three cases, i.e., $\frac{\pi}{4} < \vartheta \leqslant \frac{\pi}{2}$, $0 \leqslant \vartheta \leqslant \frac{\pi}{4}$ and $\frac{\pi}{2} < \vartheta \leqslant \pi$, respectively.

First, when $\frac{\pi}{4} < \vartheta \leqslant \frac{\pi}{2}$ or $0 \leqslant \cos\vartheta < \frac{\sqrt{2}}{2} \approx 0.707$, if the angle $\vartheta$ is reduced by $\Delta_\vartheta$,

$$\Delta_{\alpha^\parallel} = \alpha \cdot [\cos(\vartheta - \Delta_\vartheta) - \cos\vartheta] = \alpha \cdot (\Delta_\vartheta \cdot \sin\vartheta) \tag{26}$$

$$\Delta_{\alpha^\perp} = \alpha \cdot [\sin(\vartheta - \Delta_\vartheta) - \sin\vartheta] = \alpha \cdot (-\Delta_\vartheta \cdot \cos\vartheta) \tag{27}$$

$$\Rightarrow |\Delta_{\alpha^\parallel}| > |\Delta_{\alpha^\perp}| \tag{28}$$

Therefore, according to Equations 21, 22, 23 and 28, if the angle $\vartheta$ is reduced by $\Delta_\vartheta$,

$$\Delta_L^h((\alpha^\parallel + |\Delta_{\alpha^\parallel}|) \cdot \frac{\nabla_x^\parallel L(x_t^{adv}, y_o; \theta_f)}{\left\|\nabla_x^\parallel L(x_t^{adv}, y_o; \theta_f)\right\|_2}) + \Delta_L^h((\alpha^\perp - |\Delta_{\alpha^\perp}|) \cdot \frac{\nabla_x^\perp L(x_t^{adv}, y_o; \theta_f)}{\left\|\nabla_x^\perp L(x_t^{adv}, y_o; \theta_f)\right\|_2}) >$$

$$\Delta_L^h((\alpha^\parallel + |\Delta_{\alpha^\perp}|) \cdot \frac{\nabla_x^\parallel L(x_t^{adv}, y_o; \theta_f)}{\left\|\nabla_x^\parallel L(x_t^{adv}, y_o; \theta_f)\right\|_2}) + \Delta_L^h((\alpha^\perp - |\Delta_{\alpha^\perp}|) \cdot \frac{\nabla_x^\perp L(x_t^{adv}, y_o; \theta_f)}{\left\|\nabla_x^\perp L(x_t^{adv}, y_o; \theta_f)\right\|_2}) >$$

$$\Delta_L^h(\alpha^\parallel \cdot \frac{\nabla_x^\parallel L(x_t^{adv}, y_o; \theta_f)}{\left\|\nabla_x^\parallel L(x_t^{adv}, y_o; \theta_f)\right\|_2}) + \Delta_L^h(\alpha^\perp \cdot \frac{\nabla_x^\perp L(x_t^{adv}, y_o; \theta_f)}{\left\|\nabla_x^\perp L(x_t^{adv}, y_o; \theta_f)\right\|_2}) \tag{29}$$

According to Assumption 2, in a small local area (i.e., $\alpha$ is small enough, which is satisfied Lemma 1), the variation of the loss $L(x_t^{adv}, y_o; \theta_h)$ along the direction of the perturbation gradient of the surrogate model $f$ (i.e., $\Delta_L^h(\alpha \cdot \frac{\nabla_x L(x_t^{adv}, y_o; \theta_f)}{\left\|\nabla_x L(x_t^{adv}, y_o; \theta_f)\right\|_2})$) is positively correlated with the sum of the variation of the loss $L(x_t^{adv}, y_o; \theta_h)$ along the directions of the parallel and vertical components (i.e., $\Delta_L^h(\alpha^\parallel \cdot \frac{\nabla_x^\parallel L(x_t^{adv}, y_o; \theta_f)}{\left\|\nabla_x^\parallel L(x_t^{adv}, y_o; \theta_f)\right\|_2}) + \Delta_L^h(\alpha^\perp \cdot \frac{\nabla_x^\perp L(x_t^{adv}, y_o; \theta_f)}{\left\|\nabla_x^\perp L(x_t^{adv}, y_o; \theta_f)\right\|_2})$). Therefore, according to Equation 29, when the angle $\vartheta$ is reduced by $\Delta_\vartheta$, the loss $L(x_t^{adv}, y_o; \theta_h)$ is increased. According to the statistical results of Figure 1, the average cosine of the angle $\vartheta$ between the latest transfer attack generated perturbation gradients of the surrogate model and the victim model is far less than 0.707 (such as the maximum average cosine value is less than 0.25 on CIFAR10, 0.18 on CIFAR100 and 0.05 on ImageNet). Therefore, with the latest transfer attack method as the baseline, increasing the cosine of the angle $\vartheta$ can effectively improve the transferable attack success rate.

Second, when $0 \leqslant \vartheta \leqslant \frac{\pi}{4}$ or $0.707 \leqslant \cos\vartheta \leqslant 1$, if the angle $\vartheta$ is reduced by $\Delta_\vartheta$, Equation 29 may not be satisfied. Therefore, increasing the cosine of the angle $\vartheta$ may not effectively improve the transferable attack success rate.

Third, when $\frac{\pi}{2} < \vartheta \leqslant \pi$, if the angle $\vartheta$ is reduced by $\Delta_\vartheta$, Equation 29 does not hold. Therefore, increasing the cosine of the angle $\vartheta$ can not improve the transferable attack success rate. According to the statistical results of Figure 1, the average cosine of the angle $\vartheta$ is greater than 0 on the latest attack methods, so $\frac{\pi}{2} < \vartheta \leqslant \pi$ usually does not happen.

Overall, with the latest transfer attack method as the baseline, increasing the cosine of the angle $\vartheta$ can effectively improve the transferable attack success rate.

**Lemma 1** *When $\alpha$ is small enough, the loss $L(x_t^{adv}, y_o; \theta_h)$ increases fastest along the direction of the perturbation gradient $\nabla_x L(x_t^{adv}, y_o; \theta_h)$ in the region $\mathcal{B}(x_t^{adv}, \alpha)$ (i.e., in the sphere with radius $\alpha$ centered on the adversarial example $x_t^{adv}$).*

**Proof** *When $\alpha$ is small enough, in the region $\mathcal{B}(x_t^{adv}, \alpha)$, the rate of variation of the loss $L(x_t^{adv}, y_o; \theta_h)$ along the direction of the perturbation gradient $\nabla_x L(x_t^{adv}, y_o; \theta_h)$ is almost equal to the Lipschitz constant of the loss function $L(x_t^{adv}, y_o; \theta_h)$ in this region.*

**Assumption 2** *In a small local area (i.e., $\alpha$ is small enough, which is satisfied Lemma 1), the variation of the loss $L(x_t^{adv}, y_o; \theta_h)$ along the direction of the perturbation gradient of the surrogate model $f$ (i.e., $\Delta_L^h(\alpha \cdot \frac{\nabla_x L(x_t^{adv}, y_o; \theta_f)}{\left\| \nabla_x L(x_t^{adv}, y_o; \theta_f) \right\|_2})$) is positively correlated with the sum of the variation of the loss $L(x_t^{adv}, y_o; \theta_h)$ along the directions of the parallel and vertical components (i.e., $\Delta_L^h(\alpha^{\|} \cdot \frac{\nabla_x^{\|} L(x_t^{adv}, y_o; \theta_f)}{\left\| \nabla_x^{\|} L(x_t^{adv}, y_o; \theta_f) \right\|_2}) + \Delta_L^h(\alpha^{\perp} \cdot \frac{\nabla_x^{\perp} L(x_t^{adv}, y_o; \theta_f)}{\left\| \nabla_x^{\perp} L(x_t^{adv}, y_o; \theta_f) \right\|_2}))$.*

Lemma 1 and Assumption 2 are used in the Theoretical Proof of Proposition 1. Note that, Assumption 2 is reasonable. When $\alpha$ is small enough, in the region $\mathcal{B}(x_t^{adv}, \alpha)$, the rate of variation of the loss $L(x_t^{adv}, y_o; \theta_h)$ along the direction of the parallel component $\nabla_x^{\|} L(x_t^{adv}, y_o; \theta_f)$ (which is parallel to $\nabla_x L(x_t^{adv}, y_o; \theta_h)$) is almost equal to the Lipschitz constant of the loss function $L(x_t^{adv}, y_o; \theta_h)$ in this region. Meanwhile, in the region $\mathcal{B}(x_t^{adv}, \alpha)$, the rate of variation of the loss $L(x_t^{adv}, y_o; \theta_h)$ along the direction of the vertical component $\nabla_x^{\perp} L(x_t^{adv}, y_o; \theta_f)$ (which is perpendicular to $\nabla_x L(x_t^{adv}, y_o; \theta_h)$) is almost 0. Therefore, $\Delta_L^h(\alpha \cdot \frac{\nabla_x L(x_t^{adv}, y_o; \theta_f)}{\left\| \nabla_x L(x_t^{adv}, y_o; \theta_f) \right\|_2})$ is positively correlated with $\Delta_L^h(\alpha^{\|} \cdot \frac{\nabla_x^{\|} L(x_t^{adv}, y_o; \theta_f)}{\left\| \nabla_x^{\|} L(x_t^{adv}, y_o; \theta_f) \right\|_2}) + \Delta_L^h(\alpha^{\perp} \cdot \frac{\nabla_x^{\perp} L(x_t^{adv}, y_o; \theta_f)}{\left\| \nabla_x^{\perp} L(x_t^{adv}, y_o; \theta_f) \right\|_2})$.

However, as $\alpha$ increases, the Lipschitz constant of the loss function $L(x_t^{adv}, y_o; \theta_h)$ in the region $\mathcal{B}(x_t^{adv}, \alpha)$ also increases and is greater than the rate of variation of the loss $L(x_t^{adv}, y_o; \theta_h)$ along the direction of the parallel component $\nabla_x^{\|} L(x_t^{adv}, y_o; \theta_f)$. Meanwhile, in the region $\mathcal{B}(x_t^{adv}, \alpha)$, the rate of variation of the loss $L(x_t^{adv}, y_o; \theta_h)$ along the direction of the vertical component $\nabla_x^{\perp} L(x_t^{adv}, y_o; \theta_f)$ becomes uncertain. Therefore, the positive correlation between $\Delta_L^h(\alpha \cdot \frac{\nabla_x L(x_t^{adv}, y_o; \theta_f)}{\left\| \nabla_x L(x_t^{adv}, y_o; \theta_f) \right\|_2})$ and $\Delta_L^h(\alpha^{\|} \cdot \frac{\nabla_x^{\|} L(x_t^{adv}, y_o; \theta_f)}{\left\| \nabla_x^{\|} L(x_t^{adv}, y_o; \theta_f) \right\|_2}) + \Delta_L^h(\alpha^{\perp} \cdot \frac{\nabla_x^{\perp} L(x_t^{adv}, y_o; \theta_f)}{\left\| \nabla_x^{\perp} L(x_t^{adv}, y_o; \theta_f) \right\|_2})$ will weaken, which is why the transferable attack success rate of the gradient iterative-based attacks (I-FGSM, MI-FGSM, DI-FGSM, SI-NI-FGSM and VMI-FGSM) is higher than that of FGSM (because the attack step length of the gradient iterative-based attacks is less than that of FGSM).

**Corollary 1** *When $0 \leqslant \cos \vartheta < 0.707$, increasing $\cos \vartheta$ is a necessary and insufficient condition to improve the transferability of the generated adversarial examples. To effectively improve the transferability, a small step size $\alpha$ is also needed.*

**Proof (Theoretical Proof)** *For $0 \leqslant \cos \vartheta < 0.707$, if $\alpha$ is small and satisfied Lemma 1, so Proposition 1 is correct. As $\alpha$ increases, the Lipschitz constant of the loss function $L(x_t^{adv}, y_o; \theta_h)$ in the region $\mathcal{B}(x_t^{adv}, \alpha)$ also increases and is greater than the rate of variation of the loss $L(x_t^{adv}, y_o; \theta_h)$ along the direction of the parallel component $\nabla_x^{\|} L(x_t^{adv}, y_o; \theta_f)$. Meanwhile, in the region $\mathcal{B}(x_t^{adv}, \alpha)$, the rate of variation of the loss $L(x_t^{adv}, y_o; \theta_h)$ along the direction of the vertical component $\nabla_x^{\perp} L(x_t^{adv}, y_o; \theta_f)$ becomes uncertain. Therefore, the positive correlation between $\Delta_L^h(\alpha \cdot \frac{\nabla_x L(x_t^{adv}, y_o; \theta_f)}{\left\| \nabla_x L(x_t^{adv}, y_o; \theta_f) \right\|_2})$ and $\Delta_L^h(\alpha^{\|} \cdot \frac{\nabla_x^{\|} L(x_t^{adv}, y_o; \theta_f)}{\left\| \nabla_x^{\|} L(x_t^{adv}, y_o; \theta_f) \right\|_2}) + \Delta_L^h(\alpha^{\perp} \cdot \frac{\nabla_x^{\perp} L(x_t^{adv}, y_o; \theta_f)}{\left\| \nabla_x^{\perp} L(x_t^{adv}, y_o; \theta_f) \right\|_2})$ will weaken. When $\alpha$ is large to a certain extent, $\Delta_L^h(\alpha \cdot \frac{\nabla_x L(x_t^{adv}, y_o; \theta_f)}{\left\| \nabla_x L(x_t^{adv}, y_o; \theta_f) \right\|_2})$ and $\Delta_L^h(\alpha^{\|} \cdot \frac{\nabla_x^{\|} L(x_t^{adv}, y_o; \theta_f)}{\left\| \nabla_x^{\|} L(x_t^{adv}, y_o; \theta_f) \right\|_2}) + \Delta_L^h(\alpha^{\perp} \cdot \frac{\nabla_x^{\perp} L(x_t^{adv}, y_o; \theta_f)}{\left\| \nabla_x^{\perp} L(x_t^{adv}, y_o; \theta_f) \right\|_2})$ are no longer positively correlated. Then Equation 29 does not hold, so Proposition 1 is incorrect. Therefore, a small step size $\alpha$ is a key parameter to effectively improve the transferability.*

**Proof (Empirical Proof)** *As shown in Figure 1-(3), when VGG16 is the surrogate model and ResNet50 is the victim model for ImageNet, the sort of the cosine values is basically FGSM > SI-NI-FGSM > MI-FGSM > DI-FGSM > I-FGSM, but the sort of the attack success rate is SI-NI-FGSM (56.6%) > MI-FGSM (46.5%) > DI-FGSM (38.1%) > FGSM (32.8%) > I-FGSM (27.8%).*

*As shown in Figure 1-(6), when ResNet50 is the surrogate model and VGG16 is the victim model for ImageNet, the sort of the cosine values is basically FGSM > SI-NI-FGSM > MI-FGSM > DI-FGSM > I-FGSM, but the sort of the attack success rate is SI-NI-FGSM (68.7%) > MI-FGSM (55.4%) > DI-FGSM (52.6%) > FGSM (42.6%) > I-FGSM (32.1%).*

*Therefore, to effectively improve the transferability, a small step size $\alpha$ is also needed. Note that, in Figure 1, the attack strength $\epsilon$ is the same, the step size $\alpha$ of FGSM is greater than SI-NI-FGSM, MI-FGSM and DI-FGSM.*

## A.2    PROOF OF PROPOSITION 2

**Proposition 2:** *When the victim model $h$ is attacked by the white-box gradient-based attacks, the successful attacked adversarial examples prefer to be classified as the wrong categories with higher probability (i.e. the top-n wrong categories $\{y_{\tau_i}|i \leqslant \bar{n}\}$). Meanwhile, the higher the probability of the wrong category, the more likely the adversarial example is to be classified as this category.*

**Proof (Empirical Proof)** *To verify the correctness of Proposition 2, we explore the top-n wrong categories attack success rate (ASR) on I-FGSM at each iteration where the attack strength, number of iterations and step of size are $\epsilon, T, \alpha = 8/255, 10, 0.8/255$. Assuming that $ASR^{\bar{n}=n}$ denotes the top-n wrong categories ASR, and $\overline{ASR}^{\bar{n}=n}$ denotes the average top-n wrong categories, namely $\overline{ASR}^{\bar{n}=n} = \frac{ASR^{\bar{n}=n}}{\bar{n}}$. First, if the top-n wrong categories ASR is significantly higher than the average level (i.e., $\frac{1}{C-1}$ where $C$ is the number of categories of the classification task), namely $ASR^{\bar{n}=n} \gg \frac{1}{C-1}$, the previous sentence of Proposition 2 is correct with high confidence. Second, when $n1 < n2$, if the average top-$n1$ wrong categories ASR is higher than the average top-$n2$ wrong categories ASR, namely $\overline{ASR}^{\bar{n}=n1} > \overline{ASR}^{\bar{n}=n2}$, the last sentence of Proposition 2 is correct with high confidence. Empirically, Proposition 2 is verified on different surrogate models and datasets as follows.*

*When VGG16 for CIFAR10 is attacked by I-FGSM, Figure 2-(1) shows that $ASR^{\bar{n}=5} > ASR^{\bar{n}=4} > ASR^{\bar{n}=3} > ASR^{\bar{n}=2} > ASR^{\bar{n}=1} \geq 69\% \gg \frac{1}{10-1}$ at each iteration. With the increase of the iteration t, $ASR^{\bar{n}=n}$ gradually increases and approaches 100%. Figure 3-(1) shows that $\overline{ASR}^{\bar{n}=1} > \overline{ASR}^{\bar{n}=2} > \overline{ASR}^{\bar{n}=3} > \overline{ASR}^{\bar{n}=4} > \overline{ASR}^{\bar{n}=5}$ at each iteration.*

*When ResNet50 for CIFAR10 is attacked by I-FGSM, Figure 2-(2) shows that $ASR^{\bar{n}=5} > ASR^{\bar{n}=4} > ASR^{\bar{n}=3} > ASR^{\bar{n}=2} > ASR^{\bar{n}=1} \geq 76\% \gg \frac{1}{10-1}$ at each iteration. With the increase of the iteration t, $ASR^{\bar{n}=n}$ gradually increases and approaches 100%. Figure 3-(4) shows that $\overline{ASR}^{\bar{n}=1} > \overline{ASR}^{\bar{n}=2} > \overline{ASR}^{\bar{n}=3} > \overline{ASR}^{\bar{n}=4} > \overline{ASR}^{\bar{n}=5}$ at each iteration.*

*When VGG16 for CIFAR100 is attacked by I-FGSM, Figure 2-(3) shows that $ASR^{\bar{n}=50} > ASR^{\bar{n}=20} > ASR^{\bar{n}=10} > ASR^{\bar{n}=5} > ASR^{\bar{n}=1} \geq 54\% \gg \frac{1}{100-1}$ at each iteration. With the increase of the iteration t, $ASR^{\bar{n}=n}$ gradually increases and approaches 100%. Figure 3-(2) shows that $\overline{ASR}^{\bar{n}=1} > \overline{ASR}^{\bar{n}=5} > \overline{ASR}^{\bar{n}=10} > \overline{ASR}^{\bar{n}=20} > \overline{ASR}^{\bar{n}=50}$ at each iteration.*

*When ResNet50 for CIFAR100 is attacked by I-FGSM, Figure 2-(4) shows that $ASR^{\bar{n}=50} > ASR^{\bar{n}=20} > ASR^{\bar{n}=10} > ASR^{\bar{n}=5} > ASR^{\bar{n}=1} \geq 52\% \gg \frac{1}{100-1}$ at each iteration. With the increase of the iteration t, $ASR^{\bar{n}=n}$ gradually increases and approaches 100%. Figure 3-(5) shows that $\overline{ASR}^{\bar{n}=1} > \overline{ASR}^{\bar{n}=5} > \overline{ASR}^{\bar{n}=10} > \overline{ASR}^{\bar{n}=20} > \overline{ASR}^{\bar{n}=50}$ at each iteration.*

*When VGG16 for ImageNet is attacked by I-FGSM, Figure 2-(5) shows that $ASR^{\bar{n}=50} > ASR^{\bar{n}=20} > ASR^{\bar{n}=10} > ASR^{\bar{n}=5} > ASR^{\bar{n}=1} \geq 33\% \gg \frac{1}{1000-1}$ at each iteration. With the increase of the iteration t, $ASR^{\bar{n}=n}$ gradually increases and approaches 100%. Figure 3-(3) shows that $\overline{ASR}^{\bar{n}=1} > \overline{ASR}^{\bar{n}=5} > \overline{ASR}^{\bar{n}=10} > \overline{ASR}^{\bar{n}=20} > \overline{ASR}^{\bar{n}=50}$ at each iteration.*

*When ResNet50 for ImageNet is attacked by I-FGSM, Figure 2-(6) shows that $ASR^{\bar{n}=50} > ASR^{\bar{n}=20} > ASR^{\bar{n}=10} > ASR^{\bar{n}=5} > ASR^{\bar{n}=1} \geq 43\% \gg \frac{1}{1000-1}$ at each iteration. With the increase of the iteration t, $ASR^{\bar{n}=n}$ gradually increases and approaches 100%. Figure 3-(6) shows that $\overline{ASR}^{\bar{n}=1} > \overline{ASR}^{\bar{n}=5} > \overline{ASR}^{\bar{n}=10} > \overline{ASR}^{\bar{n}=20} > \overline{ASR}^{\bar{n}=50}$ at each iteration.*

*In conclusion, by discussing the above six cases on CIFAR10/100 and ImageNet datasets, the Proposition 2 is correct with high confidence. Therefore, after knowing the output of the victim model, directly classifying the adversarial examples into the category in the top-n wrong categories can remove the gradient perturbation of the other wrong categories.*

**Proof (Theoretical Proof)** *To explore whether the successful adversarial examples prefer to be classified as the wrong categories with higher probability or not, the derivation formula of $L_{CE}$ w.r.t. the input $x$ is*

$$
\frac{\partial L_{CE}}{\partial x} = \frac{\partial L_{CE}}{\partial z_o} \cdot \frac{\partial z_o}{\partial x} + \sum_{i=1(i\neq o)}^{C} \frac{\partial L_{CE}}{\partial z_i} \cdot \frac{\partial z_i}{\partial x}
$$

$$
= -\frac{1}{\ln 2} \cdot \left(1 - \frac{e^{z_o}}{\sum_{i=1}^{C} e^{z_i}}\right) \cdot \frac{\partial z_o}{\partial x} + \frac{1}{\ln 2} \cdot \sum_{i=1(i\neq o)}^{C} \frac{e^{z_i}}{\sum_{j=1}^{C} e^{z_j}} \cdot \frac{\partial z_i}{\partial x} \tag{30}
$$

*According to Equation 30, the coefficient of $\frac{\partial z_o}{\partial x}$ (i.e., $-\frac{1}{\ln 2} \cdot (1 - \frac{e^{z_o}}{\sum_{i=1}^{C} e^{z_i}})$) is less than 0, and the coefficient of $\frac{\partial z_i}{\partial x}$ (i.e., $\frac{1}{\ln 2} \cdot \frac{e^{z_i}}{\sum_{j=1}^{C} e^{z_j}}$) is greater than 0. The greater the logit output $z_i$ of the wrong category $y_i$, the larger the coefficient of $\frac{\partial z_i}{\partial x}$ (i.e., $\frac{1}{\ln 2} \cdot \frac{e^{z_i}}{\sum_{j=1}^{C} e^{z_j}}$). Therefore, in the process of the gradient ascent of the loss function $L_{CE}$, the greater the logit output $z_i$ of the wrong category $y_i$ is, the faster $z_i$ grows. Due to the fact that the greater the logit output $z_i$ of the wrong category $y_i$, the larger the probability $p_i$, the successful adversarial examples prefer to be classified as the wrong categories with higher probability.*

## A.3 THE DETAIL DESIGN PROCESS OF THE WACE LOSS

According to Proposition 1, decreasing the gradient angle between the surrogate model $f$ and the victim model $h$, i.e., the angle between $\nabla_x L_{CE}(x_t^{adv}, y_o; \theta_f)$ and $\nabla_x L_{CE}(x_t^{adv}, y_o; \theta_h)$, can enhance the transferability of adversarial examples.

According to the previous sentence of Proposition 2, because the successful attacked adversarial examples prefer to be classified as the wrong categories with higher probability, to avoid the gradient perturbation of the other wrong categories, besides maximizing the loss function $L_{CE}(x^{adv}, y_o; \theta_f)$, we also minimize the distance between the model output and the top-n wrong categories with higher probability, namely maximizing $L_{CE}(x^{adv}, y_o; \theta_f) - \sum_{i=1}^{\bar{n}} \frac{1}{\bar{n}} \cdot L_{CE}(x^{adv}, y_{\tau_i}; \theta_f)$ where each category in $\{y_{\tau_i} | i \leqslant \bar{n}\}$ is equally important.

According to the last sentence of Proposition 2, the higher the probability of the wrong category, the more likely the adversarial example is to be classified as this category. Therefore, we add weight to the distance calculation of the top-n wrong categories according to the logit of each category in the top-n wrong categories, namely maximizing $L_{CE}(x^{adv}, y_o; \theta_f) - \sum_{i=1}^{\bar{n}} \frac{e^{z_{h,\tau_i}}}{\sum_{j=1}^{\bar{n}} e^{z_{h,\tau_j}}} \cdot L_{CE}(x^{adv}, y_{\tau_i}; \theta_f)$.

Therefore, the WACE loss is:

$$
L_{WACE}(x, y_o; \theta_f, \boldsymbol{Z}_h, \bar{n}) = L_{CE}(x, y_o; \theta_f) - \frac{1}{\bar{n}} \sum_{i=1}^{\bar{n}} \frac{e^{z_{h,\tau_i}}}{\sum_{j=1}^{\bar{n}} e^{z_{h,\tau_j}}} \cdot L_{CE}(x, y_{\tau_i}; \theta_f) \tag{31}
$$

## A.4 PROOF OF THEOREM 1

**Theorem 1:** *The angle between $\nabla_x L_{WACE}(x_t^{adv}, y_o; \theta_f, \boldsymbol{Z}_h, \bar{n})$ and $\nabla_x L_{CE}(x_t^{adv}, y_o; \theta_h)$ is less than the angle between $\nabla_x L_{RCE}(x_t^{adv}, y_o; \theta_f)$ and $\nabla_x L_{CE}(x_t^{adv}, y_o; \theta_h)$, and the angle between $\nabla_x L_{CE}(x_t^{adv}, y_o; \theta_f)$ and $\nabla_x L_{CE}(x_t^{adv}, y_o; \theta_h)$.*

**Proof** *According to Proposition 2,*

$$\nabla_x L_{CE}\left(x_t^{adv}, y_o; \theta_h\right) = \frac{\partial L_{CE}}{\partial x_t^{adv}} = -\frac{\partial L_{CE}}{\partial z_{h,o}} \cdot \left(-\frac{\partial z_{h,o}}{\partial x_t^{adv}}\right) + \sum_{i=1(i\neq o)}^{C} \frac{\partial L_{CE}}{\partial z_{h,i}} \cdot \frac{\partial z_{h,i}}{\partial x_t^{adv}}$$

$$= \frac{1}{\ln 2} \cdot \left(\left(1 - \frac{e^{z_{h,o}}}{\sum_{i=1}^{C} e^{z_{h,i}}}\right) \cdot \left(-\frac{\partial z_{h,o}}{\partial x_t^{adv}}\right) + \sum_{i=1(i\neq o)}^{C} \frac{e^{z_{h,i}}}{\sum_{j=1}^{C} e^{z_{h,j}}} \cdot \frac{\partial z_{h,i}}{\partial x_t^{adv}}\right)$$

$$\approx \frac{1}{\ln 2} \cdot \left(\left(1 - \frac{e^{z_{h,o}}}{\sum_{i=1}^{C} e^{z_{h,i}}}\right) \cdot \left(-\frac{\partial z_{h,o}}{\partial x_t^{adv}}\right) + \sum_{i=1}^{\bar{n}} \frac{e^{z_{h,\tau_i}}}{\sum_{j=1}^{\bar{n}} e^{z_{h,\tau_j}}} \cdot \frac{\partial z_{h,\tau_i}}{\partial x_t^{adv}}\right) \quad (32)$$

*Eq. 6, 7 and 9 in the main paper are the CE loss, the RCE loss and the WACE loss, respectively. The gradient of each loss w.r.t. $x_t^{adv}$ on the surrogate model $f$ is as follows, respectively.*

$$\nabla_x L_{CE}\left(x_t^{adv}, y_o; \theta_f\right) = \frac{\partial L_{CE}}{\partial x_t^{adv}} = -\frac{\partial L_{CE}}{\partial z_o} \cdot \left(-\frac{\partial z_o}{\partial x_t^{adv}}\right) + \sum_{i=1(i\neq o)}^{C} \frac{\partial L_{CE}}{\partial z_i} \cdot \frac{\partial z_i}{\partial x_t^{adv}}$$

$$= \frac{1}{\ln 2} \cdot \left(\left(1 - \frac{e^{z_o}}{\sum_{i=1}^{C} e^{z_i}}\right) \cdot \left(-\frac{\partial z_o}{\partial x_t^{adv}}\right) + \sum_{i=1(i\neq o)}^{C} \frac{e^{z_i}}{\sum_{j=1}^{C} e^{z_j}} \cdot \frac{\partial z_i}{\partial x_t^{adv}}\right) \quad (33)$$

$$\nabla_x L_{RCE}\left(x_t^{adv}, y_o; \theta_f\right) = \frac{\partial L_{RCE}}{\partial x_t^{adv}} = -\frac{\partial L_{RCE}}{\partial z_o} \cdot \left(-\frac{\partial z_o}{\partial x_t^{adv}}\right) + \sum_{i=1(i\neq o)}^{C} \frac{\partial L_{RCE}}{\partial z_i} \cdot \frac{\partial z_i}{\partial x_t^{adv}}$$

$$= \frac{1}{\ln 2} \cdot \left(\left(-\frac{\partial z_o}{\partial x_t^{adv}}\right) + \sum_{i=1}^{C} \frac{1}{C} \cdot \frac{\partial z_i}{\partial x_t^{adv}}\right) \quad (34)$$

$$\nabla_x L_{WACE}\left(x_t^{adv}, y_o; \theta_f\right) = \frac{\partial L_{WACE}}{\partial x_t^{adv}} = -\frac{\partial L_{WACE}}{\partial z_o} \cdot \left(-\frac{\partial z_o}{\partial x_t^{adv}}\right) + \sum_{i=1(i\neq o)}^{C} \frac{\partial L_{WACE}}{\partial z_i} \cdot \frac{\partial z_i}{\partial x_t^{adv}}$$

$$= \frac{1}{\ln 2} \cdot \left(\left(-\frac{\partial z_o}{\partial x_t^{adv}}\right) + \sum_{i=1}^{\bar{n}} \frac{e^{z_{h,\tau_i}}}{\sum_{j=1}^{\bar{n}} e^{z_{h,\tau_j}}} \cdot \frac{\partial z_{\tau_i}}{\partial x_t^{adv}}\right) \quad (35)$$

*Assuming that $x^{adv}$ is a successful attacked adversarial example and $x_0^{adv}$ is correctly classified by the surrogate model $f$ and the victim model $h$ with almost 100% probability. **When the iteration $t$ is equal to 0**, $(1 - \frac{e^{z_o}}{\sum_{i=1}^{C} e^{z_i}})$ and $(1 - \frac{e^{z_{h,o}}}{\sum_{i=1}^{C} e^{z_{h,i}}})$ are approximately 0, and $\frac{e^{z_i}}{\sum_{j=1}^{C} e^{z_j}}$ and $\frac{e^{z_{h,i}}}{\sum_{j=1}^{C} e^{z_{h,j}}}$ are approximately 0. Eq. 32 and 33 are transformed as follows, respectively.*

$$\nabla_x L_{CE}\left(x_t^{adv}, y_o; \theta_h\right) \approx \frac{1}{\ln 2} \cdot \sum_{i=1}^{\bar{n}} \frac{e^{z_{h,\tau_i}}}{\sum_{j=1}^{\bar{n}} e^{z_{h,\tau_j}}} \cdot \frac{\partial z_{h,\tau_i}}{\partial x_t^{adv}} \quad (36)$$

$$\nabla_x L_{CE}\left(x_t^{adv}, y_o; \theta_f\right) \approx 0 \quad (37)$$

*Therefore, according to Proposition 2, in comparison with the gradient of the CE and RCE losses, Eq. 34, 35, 36 and 37 show that the gradient of the WACE loss remove the gradient of the unrelated wrong categories (i.e., the wrong categories with minimum probability).*

*__With the increase of the iteration__ $t$, according to Proposition 2, in comparison with the CE loss, Eq. 32, 33 and 35 show that the gradient of the WACE loss removes the gradient of the unrelated wrong categories and enhances the weight (or coefficient) of the gradient of the ground truth. In comparison with the RCE loss, Eq. 32, 34 and 35 show that the gradient of the WACE loss removes the gradient of the unrelated wrong categories.*

*Therefore, Theorem 1 is correct.*

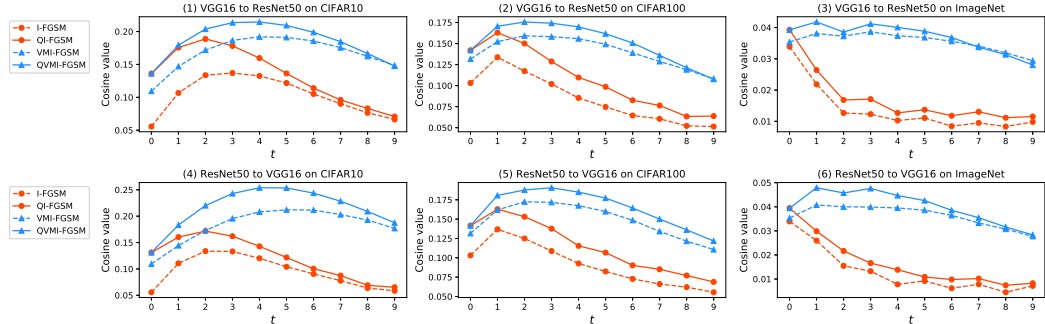

Figure 4: The cosine value of the gradient angle between the surrogate model and the victim model at each iteration $t$ when the surrogate model is attacked by different methods for CIFAR10/100 and ImageNet. For example, in subfigure (1), VGG16 as the surrogate model and ResNet50 as the victim model are attacked by different transfer attacks for CIFAR10. The query prior-based attacks can significantly improve the cosine value of the gradient angle between the surrogate model and the victim model, i.e. decrease the gradient angle between the surrogate model and the victim model. Note that the attack strength $\epsilon = 8/255$.

## A.5 THE REDUCTION OF THE GRADIENT ANGLE WITH THE WACE LOSS

To verify that the WACE loss can reduce the gradient angle between the surrogate model and the victim model, we compare the cosine value of the gradient angle between the family of iterative FGSM and their query prior-based version. As shown in Figure 4, for CIFAR10/100 and ImageNet, at each iteration, the cosine value of the gradient angle between the surrogate model and the victim model on the family of iterative FGSM are smaller than that on their query prior-based version. Therefore, the WACE loss can reduce the gradient angle between the surrogate model and the victim model.

## A.6 PROOF OF PROPOSITIONS 3, 4, 5 AND 6

**Proposition 3:** *In the untargeted attacks, when $p > 0.5$, the temperature scaling ($\mathcal{T} > 1$ and $\mathcal{K} = 1$) can eliminate a part of the fuzzy domain $\mathbb{M}_{(x,f,h)}^{NT}$.*

**Proof** *When $\mathcal{T} > 1$ and $\mathcal{K} = 1$, Eq. 14 is transformed as:*

$$FESoftmax\left(z_i; \mathcal{T}, 1\right) = \frac{e^{z_i/\mathcal{T}}}{\sum_{c=1}^{C} e^{z_c/\mathcal{T}}} \tag{38}$$

*Assuming that (i) the adversarial example $\hat{x}$ is generated without the temperature scaling and the logit output of the surrogate model $f$ is $\boldsymbol{Z}_f = f\left(\hat{x}\right) = [z_{f,1}, z_{f,2}, \cdots, z_{f,C}]$; (ii) the adversarial example $\hat{x}'$ is generated with the temperature scaling and the logit output of the surrogate model $f$ is $\boldsymbol{Z}'_f = f\left(\hat{x}\right) = \left[z'_{f,1}, z'_{f,2}, \cdots, z'_{f,C}\right]$; (iii) the same attack eventually makes that $\forall c, z'_{f,c}/\mathcal{T} = z_{f,c}$. If $\hat{x} \in \mathbb{M}_{(x,f,h)}^{NT}$,*

$$p_{\hat{c}} = \frac{e^{z_{f,\hat{c}}}}{\sum_{c=1}^{C} e^{z_{f,c}}} = \frac{e^{z'_{f,\hat{c}}/\mathcal{T}}}{\sum_{c=1}^{C} e^{z'_{f,c}/\mathcal{T}}} < p \tag{39}$$

*where $\hat{c}$ is the wrong category with the highest probability. To make $\hat{x}'$ not belong to $\mathbb{M}_{(x,f,h)}^{NT}$, according to Assumption 1, $\hat{x}'$ does not belong to $\mathbb{A}_{f,-}\left(p\right)$. Therefore, the probability threshold $p$ should satisfied:*

$$\frac{e^{z'_{f,\hat{c}}/\mathcal{T}}}{\sum_{c=1}^{C} e^{z'_{f,c}/\mathcal{T}}} < p \leqslant \frac{e^{z'_{f,\hat{c}}}}{\sum_{c=1}^{C} e^{z'_{f,c}}}$$

$$\Rightarrow \frac{1}{\sum_{c=1}^{C} e^{\frac{1}{\mathcal{T}}\left(z'_{f,c} - z'_{f,\hat{c}}\right)}} < p \leqslant \frac{1}{\sum_{c=1}^{C} e^{z'_{f,c} - z'_{f,\hat{c}}}} \tag{40}$$

When the condition that $\forall c \leqslant C \wedge c \neq \hat{c}, z'_{f,\hat{c}} > z'_{f,c}$ is satisfied,

$$0.5 < \frac{1}{\sum_{c=1}^{C} e^{\frac{1}{\mathcal{T}}\left(z'_{f,c} - z'_{f,\hat{c}}\right)}} < \frac{1}{\sum_{c=1}^{C} e^{z'_{f,c} - z'_{f,\hat{c}}}}. \tag{41}$$

*Hence,*

$$p > 0.5$$
$$\Rightarrow \forall c \leqslant C \wedge c \neq \hat{c}, z'_{f,\hat{c}} > z'_{f,c}$$
$$\Rightarrow 0.5 < \frac{1}{\sum_{c=1}^{C} e^{\frac{1}{\mathcal{T}}\left(z'_{f,c} - z'_{f,\hat{c}}\right)}} < p \leqslant \frac{1}{\sum_{c=1}^{C} e^{z'_{f,c} - z'_{f,\hat{c}}}}. \tag{42}$$

*Therefore, the temperature scaling can eliminate the fuzzy domain with $0.5 < \frac{1}{\sum_{c=1}^{C} e^{\frac{1}{\mathcal{T}}\left(z'_{f,c} - z'_{f,\hat{c}}\right)}} < p$.*

**Proposition 4:** *In the targeted attacks, when $p > 0.5$, the temperature scaling ($\mathcal{T} > 1$ and $\mathcal{K} = 1$) can eliminate a part of the fuzzy domain $\mathbb{M}^{Ta}_{(x,f,h)}$.*

**Proof** *The Proof of Proposition 4 is the same as that of Proposition 3. Note that the category $y_{\hat{c}}$ is changed to $y_{\tau}$.*

**Proposition 5:** *In the untargeted attacks, the fuzzy scaling ($\mathcal{T} = 1$ and $\mathcal{K} > 1$) can eliminate a part of the fuzzy domain $\mathbb{M}^{NT}_{(x,f,h)}$.*

**Proof** *When $\mathcal{T} = 1$ and $\mathcal{K} > 1$, Eq. 14 is transformed as:*

$$FESoftmax\left(z_i; 1, \mathcal{K}\right) = \begin{cases} \frac{e^{\mathcal{K} \cdot z_o}}{e^{\mathcal{K} \cdot z_o} + \sum_{c=1 \wedge c \neq o}^{C} e^{z_c}}, i = o \\ \frac{e^{z_i}}{e^{\mathcal{K} \cdot z_o} + \sum_{c=1 \wedge c \neq o}^{C} e^{z_c}}, i \neq o \end{cases} \tag{43}$$

*Assuming that (i) the adversarial example $\hat{x}$ is generated without the fuzzy scaling and the logit output of the surrogate model $f$ is $\boldsymbol{Z}_f = f(\hat{x}) = [z_{f,1}, z_{f,2}, \cdots, z_{f,C}]$; (ii) the adversarial example $\hat{x}'$ is generated with the fuzzy scaling and the logit output of the surrogate model $f$ is $\boldsymbol{Z}'_f = f(\hat{x}) = \left[z'_{f,1}, z'_{f,2}, \cdots, z'_{f,C}\right]$; (iii) the same attack eventually makes that $\forall c \leqslant C \wedge c \neq \hat{c}, z_{f,c} = z'_{f,c}, z_{f,o} = \mathcal{K} \cdot z'_{f,o}$. If $\hat{x} \in \mathbb{M}^{NT}_{(x,f,h)}$,*

$$p_{\hat{c}} = \frac{e^{z_{f,\hat{c}}}}{\sum_{c=1}^{C} e^{z_{f,c}}} = \frac{e^{z'_{f,\hat{c}}}}{e^{\mathcal{K} \cdot z'_{f,o}} + \sum_{c=1 \wedge c \neq o}^{C} z'_{f,c}} < p \tag{44}$$

*To make $\hat{x}'$ not belong to $\mathbb{M}^{NT}_{(x,f,h)}$, according to Assumption 1, we need to make $\hat{x}'$ does not belong to $\mathbb{A}_{f,-}(p)$. Therefore, when $z_{f,o} = \mathcal{K} \cdot z'_{f,o} > 0$,*

$$\frac{e^{z'_{f,\hat{c}}}}{e^{\mathcal{K} \cdot z'_{f,o}} + \sum_{c=1 \wedge c \neq o}^{C} e^{z'_{f,c}}} < p \leqslant \frac{e^{z'_{f,\hat{c}}}}{\sum_{c=1}^{C} e^{z'_{f,c}}} \tag{45}$$

*When $z_{f,o} = \mathcal{K} \cdot z'_{f,o} \leqslant 0$, $\hat{x}$ and $\hat{x}'$ are almost successfully attacked, i.e., $\hat{x}, \hat{x}' \notin \mathbb{M}^{NT}_{(x,f,h)}$.*

*Therefore, the fuzzy scaling can eliminate the fuzzy domain with $\frac{e^{z'_{f,\hat{c}}}}{e^{\mathcal{K} \cdot z'_{f,o}} + \sum_{c=1 \wedge c \neq o}^{C} e^{z'_{f,c}}} < p$ when $z'_{f,o} > 0$.*

**Proposition 6:** *In the targeted attacks, the fuzzy scaling ($\mathcal{T} = 1$ and $0 < \mathcal{K} < 1$) can eliminate a part of the fuzzy domain $\mathbb{M}^{Ta}_{(x,f,h)}$.*

**Proof** *When $\mathcal{T} = 1$ and $0 < \mathcal{K} < 1$, Eq. 14 is transformed as:*

$$FESoftmax\left(z_i; 1, \mathcal{K}\right) = \begin{cases} \frac{e^{\mathcal{K} \cdot z_{\tau}}}{e^{\mathcal{K} \cdot z_{\tau}} + \sum_{c=1 \wedge c \neq \tau}^{C} e^{z_c}}, i = \tau \\ \frac{e^{z_i}}{e^{\mathcal{K} \cdot z_{\tau}} + \sum_{c=1 \wedge c \neq \tau}^{C} e^{z_c}}, i \neq \tau \end{cases} \tag{46}$$

Table 1: The untargeted attack success rates (%) on six naturally trained models for CIFAR10 using various transfer attacks and two query attacks with the attack strength $\epsilon = 8/255$. The adversarial examples are generated by ResNet50. $*$ denotes the attack success rates under white-box attacks. **Average** means to calculate the average value except $*$. Note that $Q = 1$ in Q-FGSM.

| Model | Attack | Loss | V16 | V19 | R50 | WRN-16-4 | D121 | M-v2 | **Average** |
|---|---|---|---|---|---|---|---|---|---|
| | Square($Q = 10$) | - | 12.15 | 12.70 | 15.20 | 19.00 | 18.85 | 27.30 | 17.53 |
| | PRGF($Q = 10$) | - | 4.25 | 4.15 | 26.95* | 8.10 | 7.75 | 6.60 | 6.17 |
| | FGSM | CE | 44.20 | 46.20 | 65.50* | 56.15 | 53.15 | 55.10 | 50.96 |
| | | RCE | 37.95 | 38.95 | 54.70* | 49.25 | 45.65 | 49.70 | 44.3 |
| | | FECE(Ours) | **50.40** | **52.50** | **75.80*** | **63.00** | **59.10** | **59.95** | **56.99** |
| | Q-FGSM(Ours) | WACE(Ours) | 48.70 | 47.05 | 71.55* | 59.75 | 56.20 | 58.00 | 53.94 |
| | | WFCE(Ours) | 48.30 | 46.75 | 71.30* | 59.80 | 55.65 | 57.65 | 53.63 |
| | I-FGSM | CE | 59.85 | 62.55 | 99.85* | 93.95 | 88.70 | 84.35 | 77.88 |
| | | RCE | 58.05 | 60.10 | 99.05* | 90.60 | 86.15 | 82.25 | 75.43 |
| | | FECE(Ours) | 66.75 | 69.35 | **100*** | 95.70 | 91.75 | 87.35 | 82.18 |
| | QI-FGSM(Ours) | WACE(Ours) | 68.05 | 68.50 | 99.90* | **96.90** | 92.80 | 89.45 | 83.14 |
| | | WFCE(Ours) | **71.20** | **70.60** | 99.95* | 96.85 | **93.60** | **90.50** | **84.55** |
| | MI-FGSM | CE | 77.25 | 79.60 | 99.05* | 93.60 | 89.70 | 87.15 | 85.46 |
| | | RCE | 70.95 | 72.90 | 97.35* | 88.50 | 84.90 | 82.55 | 79.96 |
| | | FECE(Ours) | 83.30 | **84.70** | **100*** | 96.30 | 92.40 | 90.05 | 89.35 |
| ResNet50 | QMI-FGSM(Ours) | WACE(Ours) | 84.40 | 82.90 | 99.70* | 97.00 | 93.55 | 91.85 | 89.94 |
| | | WFCE(Ours) | **85.00** | 82.55 | 99.70* | **97.15** | **93.85** | **92.00** | **90.11** |
| | DI-FGSM | CE | 67.65 | 69.65 | 98.20* | 90.80 | 87.00 | 82.10 | 79.44 |
| | | RCE | 63.75 | 63.65 | 94.50* | 83.05 | 80.45 | 76.90 | 73.56 |
| | | FECE(Ours) | 75.05 | **76.05** | **99.35*** | 93.30 | 91.15 | 87.95 | 84.70 |
| | QDI-FGSM(Ours) | WACE(Ours) | 75.05 | 73.20 | 99.25* | **94.20** | 91.50 | 88.75 | 84.54 |
| | | WFCE(Ours) | **77.20** | 75.20 | 99.15* | 93.95 | 91.10 | **89.00** | **85.29** |
| | SINI-FGSM | CE | 73.00 | 75.80 | 98.10* | 92.35 | 89.95 | 86.45 | 83.51 |
| | | RCE | 82.45 | 84.20 | 99.55* | 96.75 | 94.85 | 92.35 | 90.12 |
| | | FECE(Ours) | 83.05 | 85.55 | 99.65* | 97.10 | 95.25 | 92.85 | 90.76 |
| | QSINI-FGSM(Ours) | WACE(Ours) | 87.55 | **87.85** | **99.85*** | 98.40 | **96.25** | **95.35** | **93.08** |
| | | WFCE(Ours) | **87.65** | 87.40 | 99.80* | **98.60** | **96.25** | 95.25 | 93.03 |
| | VMI-FGSM | CE | 80.40 | 82.20 | 99.20* | 94.60 | 90.80 | 88.95 | 87.39 |
| | | RCE | 77.75 | 78.05 | 97.50* | 92.15 | 89.00 | 87.60 | 84.91 |
| | | FECE(Ours) | 83.95 | 85.50 | 100* | 96.35 | 92.70 | 90.00 | 89.7 |
| | QVMI-FGSM(Ours) | WACE(Ours) | **88.60** | **86.10** | **99.65*** | **97.95** | **95.40** | **94.50** | **92.51** |
| | | WFCE(Ours) | 87.75 | 85.50 | 99.60* | 97.65 | 94.65 | 93.55 | 91.82 |

*Assuming that (i) the adversarial example $\hat{x}$ is generated without the fuzzy scaling and the logit output of the surrogate model $f$ is $\mathbf{Z}_f = f(\hat{x}) = [z_{f,1}, z_{f,2}, \cdots, z_{f,C}]$; (ii) the adversarial example $\hat{x}'$ is generated with the fuzzy scaling and the logit output of the surrogate model $f$ is $\mathbf{Z}'_f = f(\hat{x}) = \left[z'_{f,1}, z'_{f,2}, \cdots, z'_{f,C}\right]$; (iii) the same attack eventually makes that $\forall c \leqslant C \wedge c \neq \hat{c}, z_{f,c} = z'_{f,c}, z_{f,\tau} = \mathcal{K} \cdot z'_{f,\tau}$. If $\hat{x} \in \mathbb{M}^{Ta}_{(x,f,h)}$,*

$$p_\tau = \frac{e^{z_{f,\tau}}}{\sum_{c=1}^{C} e^{z_{f,c}}} = \frac{e^{\mathcal{K} \cdot z'_{f,\tau}}}{e^{\mathcal{K} \cdot z'_{f,\tau}} + \sum_{c=1 \wedge c \neq \tau}^{C} z'_{f,c}} < p \tag{47}$$

*To make $\hat{x}'$ not belong to $\mathbb{M}^{Ta}_{(x,f,h)}$, according to Assumption 1, we need to make $\hat{x}'$ does not belong to $\mathbb{A}_{f,-}(p)$. Therefore, when $z_{f,\tau} = \mathcal{K} \cdot z'_{f,\tau} > 0$, because $\left(z'_{f,c} - z'_{f,\tau}\right) < \left(z'_{f,c} - \mathcal{K} \cdot z'_{f,\tau}\right)$,*

$$\frac{1}{1 + \sum_{c=1 \wedge c \neq \tau}^{C} e^{z'_{f,c} - \mathcal{K} \cdot z'_{f,\tau}}} < p \leqslant \frac{1}{\sum_{c=1}^{C} e^{z'_{f,c} - z'_{f,\tau}}}$$

$$\Rightarrow \frac{e^{\mathcal{K} \cdot z'_{f,\tau}}}{e^{\mathcal{K} \cdot z'_{f,\tau}} + \sum_{c=1 \wedge c \neq \tau}^{C} z'_{f,c}} < p \leqslant \frac{e^{z'_{f,\tau}}}{\sum_{c=1}^{C} e^{z'_{f,c}}} \tag{48}$$

*When $z_{f,\tau} = \mathcal{K} \cdot z'_{f,\tau} < 0$, $\hat{x}$ and $\hat{x}'$ are almost failed attacked.*

*Therefore, the fuzzy scaling can eliminate the fuzzy domain with $\frac{e^{\mathcal{K} \cdot z'_{f,\tau}}}{e^{\mathcal{K} \cdot z'_{f,\tau}} + \sum_{c=1 \wedge c \neq \tau}^{C} z'_{f,c}} < p$ when $z'_{f,\tau} > 0$.*

Table 2: The untargeted attack success rates (%) on six naturally trained models for CIFAR100 using various transfer attacks and two query attacks with the attack strength $\epsilon = 8/255$. The adversarial examples are generated by ResNet50. $*$ denotes the attack success rates under white-box attacks. **Average** means to calculate the average value except $*$. Note that $Q = 1$ in Q-FGSM.

| Model | Attack | Loss | V16 | R50 | RN50 | WRN-16-4 | D121 | M-v2 | **Average** |
|---|---|---|---|---|---|---|---|---|---|
| | Square($Q = 10$) | - | 34.35 | 36.30 | 41.45 | 42.45 | 34.05 | 51.35 | 39.99 |
| | PRGF($Q = 10$) | - | 8.70 | 43.65* | 12.45 | 12.15 | 12.50 | 8.10 | 10.78 |
| | FGSM | CE | 64.80 | 83.40* | 68.75 | 72.15 | 71.60 | 66.60 | 68.78 |
| | | RCE | 63.85 | 81.75* | 67.55 | 70.55 | 69.95 | 66.35 | 67.65 |
| | Q-FGSM(Ours) | WACE(Ours) | **65.30** | **83.75*** | **69.65** | **72.80** | **72.60** | **67.65** | **69.6** |
| | I-FGSM | CE | 61.45 | 99.00* | 81.40 | 82.05 | 81.10 | 58.05 | 72.81 |
| | | RCE | 69.45 | 98.50* | 84.10 | 85.80 | 84.35 | 68.00 | 78.34 |
| | | FECE(Ours) | 69.45 | 98.60* | 83.80 | 87.00 | 83.85 | 67.75 | 78.37 |
| | QI-FGSM(Ours) | WACE(Ours) | 71.25 | **99.40*** | 88.55 | 90.30 | 87.65 | 70.50 | 81.65 |
| | | WFCE(Ours) | **72.40** | **99.40*** | **89.15** | **90.90** | **88.50** | **72.10** | **82.61** |
| | MI-FGSM | CE | 77.35 | 97.80* | 85.55 | 86.70 | 85.70 | 74.05 | 81.87 |
| | | RCE | 79.10 | 96.90* | 86.50 | 88.10 | 87.10 | 78.20 | 83.8 |
| | | FECE(Ours) | 80.45 | 97.15* | 87.15 | 88.25 | 87.30 | 78.25 | 84.28 |
| | QMI-FGSM(Ours) | WACE(Ours) | 80.90 | **98.80*** | 89.45 | 91.40 | **90.05** | 79.35 | 86.23 |
| ResNet50 | | WFCE(Ours) | **81.25** | 98.55* | **89.90** | **91.65** | 90.05 | **79.90** | **86.55** |
| | DI-FGSM | CE | 68.40 | 97.55* | 83.40 | 83.05 | 81.25 | 65.85 | 76.39 |
| | | RCE | 72.55 | 96.05* | 83.50 | 85.25 | 83.10 | 71.70 | 79.22 |
| | | FECE(Ours) | 73.25 | 96.40* | 85.00 | 85.35 | 83.65 | 71.85 | 79.82 |
| | QDI-FGSM(Ours) | WACE(Ours) | **74.70** | 98.25* | 88.25 | **89.65** | **88.30** | 75.15 | 83.21 |
| | | WFCE(Ours) | 74.55 | **98.45*** | **89.05** | 89.35 | 88.15 | **75.85** | **83.39** |
| | SINI-FGSM | CE | 72.90 | 97.00* | 82.60 | 84.85 | 82.65 | 70.75 | 78.75 |
| | | RCE | 84.85 | 99.55* | 91.70 | 92.75 | 92.00 | **84.05** | 89.07 |
| | | FECE(Ours) | 84.70 | 99.50* | 91.75 | 92.95 | 92.55 | 83.15 | 89.02 |
| | QSINI-FGSM(Ours) | WACE(Ours) | 86.10 | 99.65* | 93.70 | 93.75 | **93.90** | 82.20 | 89.93 |
| | | WFCE(Ours) | **86.40** | **99.70*** | **93.80** | **94.10** | 93.85 | 82.25 | **90.08** |
| | VMI-FGSM | CE | 84.40 | 98.20* | 88.75 | 90.35 | 89.25 | 80.70 | 86.69 |
| | | RCE | 84.00 | 97.90* | 90.45 | 91.20 | 90.55 | 82.70 | 87.78 |
| | | FECE(Ours) | 85.30 | 98.00* | 90.10 | 91.40 | 90.40 | 82.75 | 87.99 |
| | QVMI-FGSM(Ours) | WACE(Ours) | 87.05 | **98.95*** | **92.70** | 94.35 | 93.20 | 84.60 | 90.38 |
| | | WFCE(Ours) | **87.20** | 98.90* | 92.60 | **94.45** | **93.30** | **85.05** | **90.52** |

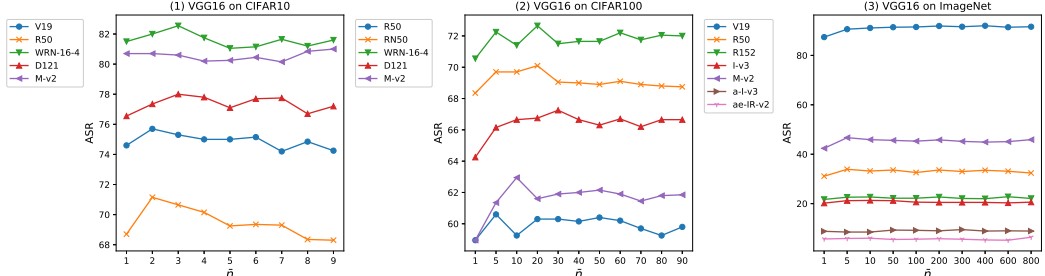

Figure 5: The untargeted attack success rates (%) on the victim models with adversarial examples generated by QI-FGSM ($\epsilon = 8/255$) for CIFAR10/100 and ImageNet (the surrogate model is VGG16) when varying the number of the top-n wrong categories $\bar{n}$.

# B    DETAILED EXPERIMENTAL ANALYSIS

In this section, we first introduce the experimental setup, then we compare our method with competitive baselines under various experimental settings.

## B.1    EXPERIMENTAL SETUP

**Datasets.** Different methods are compared on CIFAR10/100 (Krizhevsky, 2009) and ImageNet (Russakovsky et al., 2015). We randomly pick 2,000 clean images from the CIFAR10/100 test dataset and 1,000 clean images from the ILSVRC 2012 validation set (Russakovsky et al., 2015), where the selected images are correctly classified by both surrogate model and victim model.

**Models.** We consider nine naturally trained networks, including VGG16 (V16) (Simonyan & Zisserman, 2015), VGG19 (V19) (Simonyan & Zisserman, 2015), ResNet50 (R50) (He et al.,

Table 3: The untargeted attack success rates (%) of six naturally trained models and two adversarially trained models on ImageNet using various transfer attacks and two query attacks with the attack strength $\epsilon = 8/255$. The adversarial examples are generated on ResNet50. $*$ denotes the attack success rates under white-box attacks. **Avg.** means to calculate the average value of the naturally trained models except $*$. Note that $Q = 1$ in Q-FGSM.

| Model | Attack | Loss | V16 | V19 | R50 | R152 | I-v3 | M-v2 | **Avg.** | a-I-v3 | ae-IR-v2 |
|---|---|---|---|---|---|---|---|---|---|---|---|
| | Square($Q = 0$) | - | 37.5 | 35.3 | 18.1 | 12.9 | 18.2 | 35.8 | 26.3 | 13.8 | 13.4 |
| | Square($Q = 1$) | - | 38.8 | 36.5 | 20.2 | 14.6 | 19.1 | 37.4 | 27.8 | 16.3 | 14.8 |
| | Square($Q = 10$) | - | 44.4 | 41.4 | 25.6 | 18.3 | 24.1 | 43.7 | 32.9 | 21.7 | 17.6 |
| | PRGF($Q = 10$) | - | 6.9 | 5.5 | 80.2* | - | 3.2 | 7.1 | 5.7 | 3.2 | 1.7 |
| | | CE | 42.6 | 40.8 | 90.4* | 42.3 | 29.9 | 41.2 | 39.4 | **20.5** | **11.5** |
| | FGSM | RCE | 37.7 | 36.2 | 80.5* | 30.6 | 24.6 | 35.2 | 32.9 | 19.0 | 10.2 |
| | | FECE(Ours) | 43.50 | 41.4 | 90.0* | 42.4 | 30.7 | 41.1 | 39.8 | 20.0 | 11.0 |
| | Q-FGSM | WACE(Ours) | **44.2** | **42.5** | 90.7* | **42.6** | **31.2** | 42.8 | **40.7** | 20.1 | 10.8 |
| | (Ours) | WFCE(Ours) | 43.60 | 42.3 | **90.8*** | 41.9 | 31.1 | **43.5** | 40.5 | 19.5 | 11.4 |
| | I-FGSM | CE | 32.1 | 29.7 | 100* | 45.0 | 17.0 | 33.2 | 31.4 | 9.1 | 6.0 |
| | | RCE | 28.9 | 28.0 | 100* | 35.2 | 16.2 | 29.7 | 27.6 | 8.4 | 5.7 |
| | QI-FGSM (Ours) | WACE(Ours) | **39.9** | **36.6** | 100* | 53.0 | 22.6 | 44.4 | 39.3 | 10.9 | 7.3 |
| R | | CE | 55.4 | 53.0 | 100* | 70.7 | 37.8 | 58.6 | 55.1 | 17.1 | 11.9 |
| e | MI-FGSM | RCE | 57.1 | 56.6 | 100* | 63.6 | 35.9 | 56.2 | 53.9 | 16.1 | 11.6 |
| s | | FECE(Ours) | 58.70 | 57.3 | 100* | 65.9 | 37.4 | 56.3 | 55.1 | 16.6 | 11.8 |
| N | QMI-FGSM | WACE(Ours) | 62.5 | **60.5** | 100* | 74.8 | **43.1** | 65.0 | **61.2** | **19.9** | 13.3 |
| e | (Ours) | WFCE(Ours) | **63.30** | 58.9 | 100* | **74.9** | 41.9 | **66.1** | 61.0 | 19.8 | **13.8** |
| t | DI-FGSM | CE | 52.6 | 49.2 | 100* | 62.7 | 36.9 | 56.0 | 51.5 | 11.7 | 9.3 |
| 5 | | RCE | 46.1 | 46.9 | 100* | 53.0 | 36.1 | 48.4 | 46.1 | 12.2 | 9.5 |
| 0 | QDI-FGSM (Ours) | WACE(Ours) | **61.0** | **56.9** | 100* | 68.9 | 41.5 | 64.5 | 58.6 | 16.9 | 12.7 |
| | | CE | 68.7 | 68.2 | 100* | 81.8 | 51.4 | 72.3 | 68.5 | 24.2 | 16.7 |
| | SINI-FGSM | RCE | 70.5 | 69.5 | 100* | 81.0 | 52.5 | 73.7 | 69.4 | 21.6 | 16.1 |
| | | FECE(Ours) | 72.30 | 71.1 | 100* | 84.0 | 52.5 | 74.2 | 70.8 | 23.7 | 16.4 |
| | QSINI-FGSM | WACE(Ours) | 73.1 | **72.0** | 100* | 87.1 | **57.0** | 78.9 | 73.6 | **26.1** | 21.0 |
| | (Ours) | WFCE(Ours) | **75.10** | 71.3 | 100* | **87.4** | **57.0** | **79.3** | **74.0** | 26.0 | **21.1** |
| | | CE | 69.1 | 68.5 | 99.9* | 83.9 | 53.1 | 72.1 | 69.3 | 21.4 | 16.9 |
| | VMI-FGSM | RCE | 69.9 | 70.9 | 100* | 79.5 | 52.1 | 71.8 | 68.8 | 22.1 | 18.7 |
| | | FECE(Ours) | 75.90 | 74.1 | 100* | 83.2 | 55.3 | 74.9 | 72.7 | 22.8 | 19.1 |
| | QVMI-FGSM | WACE(Ours) | 78.6 | 74.8 | **100*** | 88.2 | **60.8** | **82.3** | 76.9 | **27.3** | **23.2** |
| | (Ours) | WFCE(Ours) | **78.90** | **75.0** | 100* | **88.8** | 60.4 | 81.5 | 76.9 | 27.2 | 22.7 |

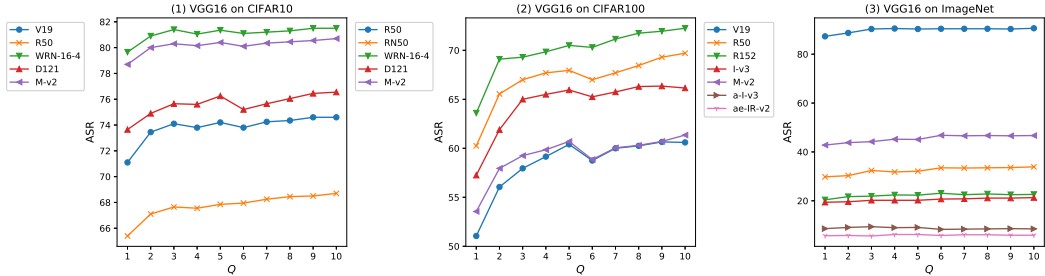

Figure 6: The untargeted attack success rates (%) on the victim models with adversarial examples generated by QI-FGSM ($\epsilon = 8/255$) for CIFAR10/100 and ImageNet (the surrogate model is VGG16) when varying the number of queries $Q$.

2016), ResNet152 (R152) (He et al., 2016), ResNext50 (RN50) (Xie et al., 2017), WideResNet-16-4 (WRN-16-4) (Zagoruyko & Komodakis, 2016), Inception-v3 (I-v3) (Szegedy et al., 2016), DenseNet121 (D121) (Huang et al., 2017) and MobileNet-v2 (M-v2) (Sandler et al., 2018) and two adversarially trained models, namely adversarial Inception-v3 (a-I-v3) and adversarial ensemble Inception-Resnet-v2 (ae-IR-v2) (Tramèr et al., 2018). We choose VGG16 and ResNet50 as source models for CIFAR10/100 and ImageNet, respectively. The CIFAR10/100 models are trained from scratch and the ImageNet models are the pretrained models in (Wightman, 2019; Huang, 2017).

**Baselines.** Several most recently proposed methods aiming at generating transferable adversarial examples are taken as baselines, i.e. FGSM (Goodfellow et al., 2015), I-FGSM (Kurakin et al., 2017), MI-FGSM (Dong et al., 2018), DI-FGSM (Xie et al., 2019), SI-NI-FGSM (Lin et al., 2020) and VMI-FGSM (Wang & He, 2021), which are implemented in a pytorch repository (Kim, 2020).

Table 4: The untargeted attack success rates (%) on six naturally trained models for CIFAR10 using various transfer attacks and two-query attacks with the attack strength $\epsilon = 16/255$. The adversarial examples are generated by ResNet50. $*$ denotes the attack success rates under white-box attacks. **Average** means to calculate the average value except $*$. Note that $Q = 1$ in Q-FGSM.

| Model | Attack | Loss | V16 | V19 | R50 | WRN-16-4 | D121 | M-v2 | **Average** |
|---|---|---|---|---|---|---|---|---|---|
| | Square($Q = 10$) | - | 42.10 | 43.50 | 45.00 | 57.70 | 57.30 | 66.20 | 51.97 |
| | PRGF($Q = 10$) | - | 9.25 | 10.40 | 43.45* | 19.25 | 16.55 | 15.40 | 14.17 |
| | FGSM | CE | 66.60 | 69.55 | 80.25* | 75.85 | 73.95 | 77.95 | 72.78 |
| | | RCE | 60.90 | 64.35 | 74.90* | 71.35 | 68.30 | 76.00 | 68.18 |
| | | FECE(Ours) | **73.45** | **75.50** | **84.65*** | **80.35** | **76.65** | **80.85** | **77.36** |
| | Q-FGSM(Ours) | WACE(Ours) | 71.25 | 71.00 | 82.70* | 77.35 | 74.05 | 79.30 | 74.59 |
| | | WFCE(Ours) | 70.75 | 71.35 | 83.00* | 76.60 | 73.90 | 79.75 | 74.47 |
| | I-FGSM | CE | 84.55 | 86.70 | 100* | 99.05 | 97.55 | 94.50 | 92.47 |
| | | RCE | 84.50 | 85.60 | 100* | 99.15 | 96.70 | 95.50 | 92.29 |
| | | FECE(Ours) | 88.70 | 90.10 | 100* | 99.60 | 98.20 | 96.40 | 94.60 |
| | QI-FGSM(Ours) | WACE(Ours) | 89.30 | 90.15 | 100* | 99.70 | **98.60** | 96.25 | 94.8 |
| | | WFCE(Ours) | **92.00** | **92.00** | **100*** | **99.80** | 98.55 | **97.55** | **95.98** |
| | MI-FGSM | CE | 95.50 | 96.45 | 99.95* | 99.50 | 98.75 | 97.20 | 97.48 |
| | | RCE | 94.30 | 95.35 | 99.85* | 98.90 | 97.90 | 96.25 | 96.54 |
| | | FECE(Ours) | 97.45 | **98.10** | 100* | **100** | **99.70** | 98.10 | 98.67 |
| | QMI-FGSM(Ours) | WACE(Ours) | 97.35 | 97.45 | 100* | 99.80 | 99.20 | 98.25 | 98.41 |
| ResNet50 | | WFCE(Ours) | **97.80** | 97.45 | **100*** | 99.75 | 99.30 | **99.10** | **98.68** |
| | DI-FGSM | CE | 93.10 | 94.20 | 99.95* | 99.05 | 97.80 | 96.45 | 96.12 |
| | | RCE | 91.35 | 91.65 | 99.45* | 98.15 | 97.05 | 96.40 | 94.92 |
| | | FECE(Ours) | 95.20 | 95.45 | 100* | 99.75 | 99.05 | 98.10 | 97.51 |
| | QDI-FGSM(Ours) | WACE(Ours) | **95.95** | 95.00 | 100* | **99.80** | **99.20** | 98.45 | 97.68 |
| | | WFCE(Ours) | 95.80 | **95.85** | **100*** | **99.80** | **99.20** | **98.75** | **97.88** |
| | SINI-FGSM | CE | 96.50 | 97.10 | 99.95* | 99.55 | 98.75 | 97.85 | 97.95 |
| | | RCE | 98.25 | 98.75 | 100* | 99.95 | 99.70 | 99.35 | 99.2 |
| | | FECE(Ours) | 98.35 | **98.90** | 100* | **100** | 99.75 | 99.45 | 99.29 |
| | QSINI-FGSM(Ours) | WACE(Ours) | 98.70 | 98.70 | 100* | **100** | **99.90** | 98.80 | 99.22 |
| | | WFCE(Ours) | **98.95** | 98.80 | **100*** | 99.95 | **99.90** | **98.90** | **99.30** |
| | VMI-FGSM | CE | 96.80 | 97.10 | 99.95* | 99.25 | 98.85 | 97.40 | 97.88 |
| | | RCE | 97.65 | 97.70 | 99.80* | 99.30 | 99.20 | 98.90 | 98.55 |
| | | FECE(Ours) | 97.45 | 97.85 | 100* | 99.70 | 99.25 | 97.70 | 98.39 |
| | QVMI-FGSM(Ours) | WACE(Ours) | **98.55** | **98.60** | **100*** | **99.75** | **99.70** | 99.30 | **99.18** |
| | | WFCE(Ours) | 98.40 | 98.30 | 100* | 99.65 | 99.55 | **99.40** | 99.06 |

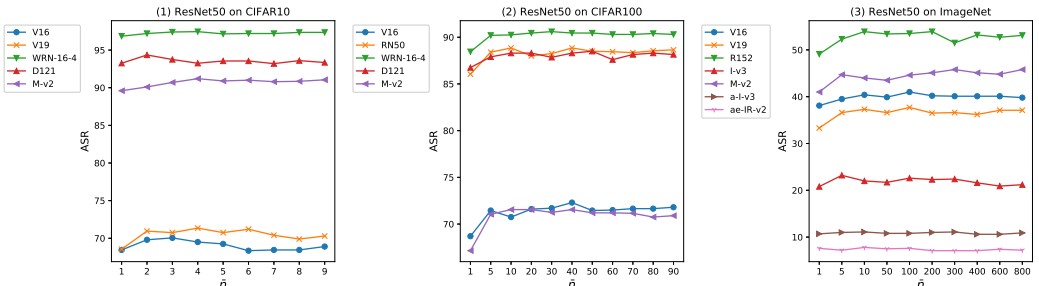

Figure 7: The untargeted attack success rates (%) on the victim models with adversarial examples generated by QI-FGSM ($\epsilon = 8/255$) for CIFAR10/100 and ImageNet (the surrogate model is ResNet50) when varying the number of the top-n wrong categories $\bar{n}$.

In addition, the RCE loss (Zhang et al., 2022a), which is integrated into the above transfer attacks instead of cross-entropy loss (CE), and two black-box query attacks, i.e. P-RGF (Cheng et al., 2019) and Square (Andriushchenko et al., 2020), are taken as baselines to further validate the effectiveness of our method.

**Hyper-parameters.** On CIFAR10/100 and ImageNet, we set the maximum perturbation, number of iteration and step size as $\epsilon, T, \alpha = 8/255, 10, 0.8/255$ or $16/255, 10, 1.6/255$. We set the decay factor $\mu = 1.0$ for MI-FGSM, SI-NI-FGSM and VMI-FGSM. The transformation probability is set to 0.5 for DI-FGSM. The number of scale copies is 5 for SI-NI-FGSM. The number of sampled examples in the neighborhood and the upper bound of neighborhood are 20 and 1.5, respectively. The number of query, which is the same as that of our query prior-based attacks, is set to $Q = 10$

Table 5: The untargeted attack success rates (%) on six naturally trained models for CIFAR100 using various transfer attacks and two query attacks with the attack strength $\epsilon = 16/255$. The adversarial examples are generated by ResNet50. $*$ denotes the attack success rates under white-box attacks. **Average** means to calculate the average value except $*$. Note that $Q = 1$ in Q-FGSM.

| Model | Attack | Loss | V16 | R50 | RN50 | WRN-16-4 | D121 | M-v2 | **Average** |
|---|---|---|---|---|---|---|---|---|---|
| | Square(Q=10) | - | 73.25 | 74.15 | 74.90 | 77.95 | 71.80 | 84.80 | 76.14 |
| | PRGF(Q=10) | - | 18.05 | 60.95* | 26.65 | 26.40 | 25.00 | 17.15 | 22.65 |
| | FGSM | CE | **84.75** | 91.15* | 86.05 | 88.25 | 85.80 | 86.60 | 86.29 |
| | | RCE | 83.55 | 90.90* | 85.80 | 87.65 | 85.25 | 86.55 | 85.76 |
| | Q-FGSM(Ours) | WACE(Ours) | 84.30 | **91.90*** | 87.00 | **89.30** | **86.00** | **87.40** | **86.8** |
| | I-FGSM | CE | 80.55 | 99.90* | 91.65 | 93.45 | 90.85 | 75.80 | 86.46 |
| | | RCE | 86.80 | 99.90* | 94.30 | 95.85 | 94.50 | 85.95 | 91.48 |
| | | FECE(Ours) | 87.75 | 99.85* | 93.95 | 95.50 | 94.15 | 84.90 | 91.25 |
| | QI-FGSM(Ours) | WACE(Ours) | 88.20 | 99.90* | **96.05** | 96.95 | 95.60 | 85.70 | 92.5 |
| | | WFCE(Ours) | **89.15** | 99.90* | 95.95 | **97.90** | **96.35** | **87.35** | **93.34** |
| | MI-FGSM | CE | 92.10 | 99.70* | 93.95 | 96.25 | 94.75 | 89.00 | 93.21 |
| | | RCE | 94.80 | 99.45* | 96.00 | 97.10 | 96.00 | **93.20** | 95.42 |
| | | FECE(Ours) | **95.30** | 99.55* | 95.75 | 97.00 | 95.95 | 92.25 | 95.25 |
| | QMI-FGSM(Ours) | WACE(Ours) | 94.70 | 99.80* | 97.05 | **98.55** | 97.00 | 91.90 | 95.84 |
| ResNet50 | | WFCE(Ours) | 94.80 | **99.85*** | **97.10** | 98.30 | **97.15** | 92.55 | **95.98** |
| | DI-FGSM | CE | 87.65 | 99.60* | 93.75 | 95.00 | 93.90 | 83.45 | 90.75 |
| | | RCE | 91.70 | 99.30* | 94.50 | 96.10 | 94.60 | 89.10 | 93.2 |
| | | FECE(Ours) | 92.20 | 99.45* | 95.05 | 96.15 | 95.65 | 89.90 | 93.79 |
| | QDI-FGSM(Ours) | WACE(Ours) | 92.85 | **99.80*** | **97.00** | **97.80** | **97.05** | 90.95 | **95.13** |
| | | WFCE(Ours) | **93.25** | 99.65* | 96.75 | 97.65 | 96.85 | **91.10** | 95.12 |
| | SINI-FGSM | CE | 91.95 | 99.70* | 93.70 | 96.40 | 95.50 | 91.40 | 93.79 |
| | | RCE | **97.50** | **100*** | 98.35 | 98.80 | 98.95 | **97.35** | **98.19** |
| | | FECE(Ours) | 97.45 | 99.95* | 98.05 | **98.90** | **99.00** | 96.35 | 97.95 |
| | QSINI-FGSM(Ours) | WACE(Ours) | 96.90 | **100*** | 98.65 | 98.65 | 98.85 | 95.25 | 97.66 |
| | | WFCE(Ours) | 97.05 | **100*** | **98.80** | 98.55 | 98.75 | 95.30 | 97.69 |
| | VMI-FGSM | CE | 95.85 | 99.75* | 96.20 | 98.15 | 97.20 | 94.15 | 96.31 |
| | | RCE | 97.25 | 99.70* | 97.90 | 98.30 | 97.80 | 95.50 | 97.35 |
| | | FECE(Ours) | 97.25 | 99.85* | 97.30 | 98.50 | 98.15 | 95.70 | 97.38 |
| | QVMI-FGSM(Ours) | WACE(Ours) | 97.40 | 99.85* | **98.05** | 99.25 | **98.60** | **96.30** | **97.92** |
| | | WFCE(Ours) | **97.65** | **99.90*** | 97.90 | **99.35** | 98.45 | 96.05 | 97.88 |

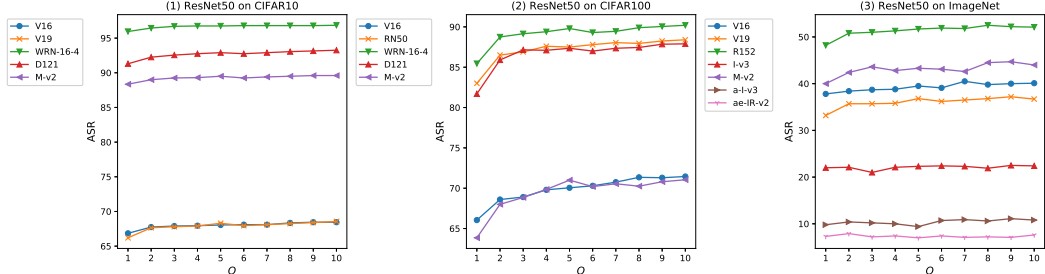

Figure 8: The untargeted attack success rates (%) on the victim models with adversarial examples generated by QI-FGSM ($\epsilon = 8/255$) for CIFAR10/100 and ImageNet (the surrogate model is ResNet50) when varying the number of queries $Q$.

for Square and P-RGF. For the proposed method, we set $\bar{n}=1$ and $Q = 10$ for CIFAR10, $\bar{n}=5$ and $Q = 10$ for CIFAR100 and ImageNet. Table 30 concludes the best parameter combination of $\mathcal{K}$ and $\mathcal{T}$ for the FECE loss on different transfer attacks and datasets with the ResNet50 as the surrogate model.

## B.2 COMPARISON WITH OR WITHOUT THE QUERY PRIORS ON THE UNTARGETED ATTACKS

### B.2.1 ATTACKING A NATURALLY TRAINED MODEL

To validate that the query priors can enhance the transferability of the transfer attacks, we perform six transfer attacks with or without the query priors to attack six naturally trained models for CIFAR10/100 and ImageNet. As shown in Tables 1, 2, 3, 7, 8 and 9, when the attack strength

Table 6: The untargeted attack success rates (%) on six naturally trained models and two adversarially trained models for ImageNet using various transfer attacks and two query attacks with the attack strength $\epsilon = 16/255$. The adversarial examples are generated by ResNet50. $*$ denotes the attack success rates under white-box attacks. **Avg.** means to calculate the average value of the naturally trained models except $*$. Note that $Q = 1$ in Q-FGSM.

| Model | Attack | Loss | V16 | V19 | R50 | R152 | I-v3 | M-v2 | **Avg.** | a-I-v3 | ae-IR-v2 |
|---|---|---|---|---|---|---|---|---|---|---|---|
| | Square(Q=0) | - | 77.8 | 73.2 | 42.2 | 30.7 | 40.4 | 72.2 | 56.1 | 35.1 | 30.4 |
| | Square(Q=1) | - | 78.5 | 74.6 | 44.0 | 33.5 | 41.9 | 73.0 | 57.6 | 37.3 | 32.2 |
| | Square(Q=10) | - | 83.8 | 80.6 | 55.8 | 43.4 | 51.9 | 79.5 | 65.8 | 47.3 | 40.1 |
| | PRGF(Q=10) | - | 12.1 | 11.0 | 88.9* | - | 6.7 | 13.6 | 10.9 | 5.9 | 3.6 |
| | | CE | 66.7 | 63.0 | **87.1*** | 52.8 | 41.4 | 61.4 | 57.1 | 32.7 | 19.0 |
| | FGSM | RCE | 62.8 | 61.3 | 80.6* | 44.4 | 38.1 | 60.7 | 53.5 | 31.8 | 18.6 |
| | | FECE(Ours) | 66.7 | 63.2 | 87.0* | **53.0** | 42.0 | 61.6 | 57.3 | 33.2 | 19.1 |
| | Q-FGSM | WACE(Ours) | **66.9** | 63.8 | 86.8* | 51.8 | 42.3 | 63.3 | **57.6** | **33.7** | **21.3** |
| | (Ours) | WFCE(Ours) | 66.2 | 63.6 | 86.9* | 51.4 | **42.8** | 63.5 | 57.5 | 33.2 | 20.3 |
| | | CE | 49.3 | 46.8 | 100* | 66.5 | 29.2 | 53.6 | 49.1 | 15.1 | 10.0 |
| | I-FGSM | RCE | 52.4 | 49.0 | 100* | 60.9 | 30.2 | 52.4 | 49.0 | 15.9 | 9.6 |
| | QI-FGSM (Ours) | WACE(Ours) | **61.8** | **57.1** | 100* | **75.6** | **36.4** | **64.6** | **59.1** | **17.4** | **11.2** |
| | | CE | 77.1 | 75.7 | 100* | 84.8 | 56.2 | 75.5 | 73.9 | 27.8 | 18.7 |
| R | MI-FGSM | RCE | 82.8 | 81.3 | 100* | 84.0 | 57.9 | 80.1 | 77.2 | 28.5 | 19.2 |
| e | | FECE(Ours) | **84.3** | **81.4** | 100* | 85.5 | 57.6 | 80.6 | 77.9 | 28.8 | 18.6 |
| s | QMI-FGSM | WACE(Ours) | 81.3 | 78.9 | 100* | 88.6 | 61.0 | 82.5 | 78.5 | 28.6 | **23.5** |
| N | (Ours) | WFCE(Ours) | 83.5 | 80.6 | **100*** | 88.8 | 63.2 | 83.8 | 80.0 | 30.2 | 22.4 |
| e | | CE | 73.1 | 71.0 | 100* | 83.7 | 56.7 | 75.7 | 72.0 | 19.3 | 13.5 |
| t | DI-FGSM | RCE | 75.8 | 72.3 | 100* | 81.3 | 54.6 | 74.8 | 71.8 | 18.7 | 13.8 |
| 5 | QDI-FGSM (Ours) | WACE(Ours) | **81.0** | **78.5** | 100* | **89.8** | **62.7** | **83.3** | **79.1** | **24.2** | **17.0** |
| 0 | | CE | 91.8 | 91.2 | 100* | 96.6 | 79.3 | 91.9 | 90.2 | **41.2** | 30.0 |
| | SINI-FGSM | RCE | 93.1 | 94.3 | 100* | **97.5** | 79.5 | **94.4** | 91.8 | 37.0 | 26.7 |
| | | FECE(Ours) | **93.9** | **94.5** | 100* | 97.5 | **80.2** | 94.3 | **92.08** | 40.1 | 28.8 |
| | QSINI-FGSM | WACE(Ours) | 90.5 | 89.2 | 100* | 96.3 | 78.0 | 92.8 | 89.4 | 38.8 | 31.4 |
| | (Ours) | WFCE(Ours) | 90.7 | 89.8 | 100* | 96.4 | 77.9 | 93.3 | 89.6 | 39.8 | **32.1** |
| | | CE | 89.3 | 88.3 | 100* | 95.7 | 74.2 | 88.6 | 87.2 | 38.1 | 32.6 |
| | VMI-FGSM | RCE | 91.7 | 91.2 | 100* | 96.5 | 80.1 | 91.2 | 90.1 | 44.2 | 38.9 |
| | | FECE(Ours) | **92.3** | **92.5** | 100* | 97.5 | 81.0 | 92.2 | 91.1 | 44.6 | 37.2 |
| | QVMI-FGSM | WACE(Ours) | 91.5 | 91.8 | 100* | **97.6** | **82.3** | **93.7** | 91.4 | 47.0 | 40.5 |
| | (Ours) | WFCE(Ours) | 92.1 | 91.8 | **100*** | 97.6 | 82.2 | 93.7 | **91.5** | **47.2** | **41.8** |

Table 7: The untargeted attack success rates (%) on six naturally trained models for CIFAR10 using various transfer attacks and two query attacks with the attack strength $\epsilon = 8/255$. The adversarial examples are generated by VGG16. $*$ denotes the attack success rates under white-box attacks. **Average** means to calculate the average value except $*$. Note that $Q = 1$ in Q-FGSM.

| Model | Attack | Loss | V16 | V19 | R50 | WRN-16-4 | D121 | M-v2 | **Average** |
|---|---|---|---|---|---|---|---|---|---|
| | Square ($Q = 10$) | - | 12.65 | 13.00 | 14.15 | 18.75 | 19.40 | 27.15 | 17.52 |
| | PRGF ($Q = 10$) | - | 23.75* | 4.30 | 4.20 | 5.00 | 5.20 | 4.90 | 4.72 |
| | FGSM | CE | 62.40* | 45.20 | 41.90 | 46.90 | 43.55 | 48.40 | 45.19 |
| | | RCE | 56.65* | 41.30 | 38.45 | 42.95 | 39.00 | 44.25 | 41.19 |
| | Q-FGSM (Ours) | WACE | **72.85*** | **49.30** | **45.50** | **51.60** | **47.05** | **54.65** | **49.62** |
| | I-FGSM | CE | 97.80* | 68.95 | 61.45 | 72.65 | 68.30 | 70.90 | 68.45 |
| | | RCE | 96.80* | 68.10 | 60.80 | 72.20 | 68.10 | 70.65 | 67.97 |
| | QI-FGSM (Ours) | WACE | **98.80*** | **74.35** | **68.65** | **80.90** | **76.50** | **80.35** | **76.15** |
| | MI-FGSM | CE | 93.35* | 75.75 | 70.75 | 76.05 | 74.00 | 74.10 | 74.13 |
| | | RCE | 90.70* | 72.60 | 68.85 | 72.90 | 71.40 | 72.55 | 71.66 |
| VGG16 | QMI-FGSM (Ours) | WACE | **96.65*** | **80.80** | **77.65** | **84.20** | **80.95** | **84.70** | **81.66** |
| | DI-FGSM | CE | 91.30* | 69.30 | 62.80 | 71.50 | 69.20 | 70.60 | 68.68 |
| | | RCE | 89.60* | 67.20 | 61.25 | 68.90 | 64.90 | 68.10 | 66.07 |
| | QDI-FGSM (Ours) | WACE | **95.85*** | **74.25** | **71.15** | **81.40** | **76.45** | **80.80** | **76.81** |
| | SI-NI-FGSM | CE | 94.00* | 72.10 | 68.10 | 77.90 | 72.85 | 75.55 | 73.3 |
| | | RCE | 98.60* | 85.60 | 80.30 | 87.45 | 85.45 | 86.35 | 85.03 |
| | QSI-NI-FGSM (Ours) | WACE | **99.25*** | **88.25** | **84.95** | **90.80** | **89.20** | **90.70** | **88.78** |
| | VMI-FGSM | CE | 94.90* | 80.00 | 76.60 | 81.60 | 78.40 | 79.95 | 79.31 |
| | | RCE | 91.70* | 77.30 | 75.50 | 79.10 | 76.65 | 78.75 | 77.46 |
| | QVMI-FGSM (Ours) | WACE | **97.10*** | **84.40** | **83.35** | **88.40** | **85.70** | **89.10** | **86.19** |

$\epsilon = 8/255$, the query prior-based attacks with the WACE loss can not only significantly improve the transfer attack success rate of the black-box setting but also improve the attack success rate of the white-box setting on different surrogate models and datasets. In comparison with the CE loss, the average increase of the ASR is 2.98 to 4.43% on Q-FGSM and 4.12 to 15.48% on the other five

Table 8: The untargeted attack success rates (%) on six naturally trained models for CIFAR100 using various transfer attacks and two query attacks with the attack strength $\epsilon = 8/255$. The adversarial examples are generated by VGG16. $*$ denotes the attack success rates under white-box attacks. **Average** means to calculate the average value except $*$. Note that $Q = 1$ in Q-FGSM.

| Model | Attack | Loss | V16 | R50 | RN50 | WRN-16-4 | D121 | M-v2 | **Average** |
|---|---|---|---|---|---|---|---|---|---|
| | Square ($Q = 10$) | - | 36.75 | 31.60 | 39.40 | 41.40 | 33.40 | 49.90 | 38.74 |
| | PRGF ($Q = 10$) | - | 46.20* | 6.90 | 8.10 | 8.00 | 6.95 | 6.95 | 7.38 |
| | FGSM | CE | **85.75*** | 63.15 | 64.30 | 68.65 | 66.25 | 66.45 | 65.76 |
| | | RCE | 83.55* | 61.05 | 63.25 | 67.20 | 65.25 | 64.40 | 64.23 |
| | Q-FGSM (Ours) | WACE | 85.30* | **63.20** | **65.65** | **68.75** | **67.00** | **67.65** | **66.45** |
| | I-FGSM | CE | 99.60* | 50.70 | 58.70 | 61.15 | 54.10 | 50.65 | 55.06 |
| | | RCE | 99.50* | 47.25 | 56.30 | 60.00 | 54.10 | 52.90 | 54.11 |
| | QI-FGSM (Ours) | WACE | **99.75*** | **60.70** | **69.45** | **72.05** | **66.70** | **61.50** | **66.08** |
| | MI-FGSM | CE | 99.30* | 69.30 | 72.75 | 76.35 | 71.10 | 69.35 | 71.77 |
| | | RCE | 99.10* | 68.20 | 72.50 | 76.05 | 72.55 | 70.25 | 71.91 |
| VGG16 | QMI-FGSM (Ours) | WACE | 99.30* | **76.50** | **79.80** | **82.50** | **77.70** | **76.45** | **78.59** |
| | DI-FGSM | CE | 98.60* | 59.30 | 64.05 | 66.35 | 60.25 | 58.50 | 61.69 |
| | | RCE | 97.30* | 57.60 | 63.90 | 67.70 | 59.70 | 62.25 | 62.23 |
| | QDI-FGSM (Ours) | WACE | **99.00*** | **67.05** | **73.45** | **75.60** | **69.45** | **69.40** | **70.99** |
| | SI-NI-FGSM | CE | 96.75* | 64.35 | 70.10 | 73.90 | 70.65 | 68.20 | 69.44 |
| | | RCE | 99.60* | 71.65 | 76.15 | 80.05 | 76.10 | 75.25 | 75.84 |
| | QSI-NI-FGSM (Ours) | WACE | **99.90*** | **77.15** | **81.40** | **83.70** | **80.55** | **77.75** | **80.11** |
| | VMI-FGSM | CE | **99.55*** | 77.70 | 80.00 | 82.85 | 79.70 | 76.85 | 79.42 |
| | | RCE | 99.05* | 79.20 | 80.20 | 83.90 | 80.30 | 79.65 | 80.65 |
| | QVMI-FGSM (Ours) | WACE | 99.35* | **83.30** | **85.50** | **87.75** | **84.40** | **81.40** | **84.47** |

Table 9: The untargeted attack success rates (%) on six naturally trained models and two adversarially trained models for ImageNet using various transfer attacks and two query attacks with the attack strength $\epsilon = 8/255$. The adversarial examples are generated by VGG16. $*$ denotes the attack success rates under white-box attacks. **Avg.** means to calculate the average value of the naturally trained models except $*$. Note that $Q = 1$ in Q-FGSM.

| Model | Attack | Loss | V16 | V19 | R50 | R152 | I-v3 | M-v2 | **Avg.** | a-I-v3 | ae-IR-v2 |
|---|---|---|---|---|---|---|---|---|---|---|---|
| | Square ($Q = 0$) | - | 40.2 | 34.5 | 16.5 | 12.4 | 17.9 | 34.1 | 25.9 | 12.3 | 11.2 |
| | Square ($Q = 1$) | - | 41.2 | 35.7 | 17.8 | 13.5 | 19.2 | 35.1 | 27.1 | 14.5 | 12.4 |
| | Square ($Q = 10$) | - | 47.7 | 41.6 | 23.2 | 16.2 | 23.6 | 41.3 | 32.3 | 20.3 | 16.8 |
| | PRGF ($Q = 10$) | - | 88.2* | 21.1 | 4.3 | - | 2.7 | 6.1 | 8.6 | 2.5 | 1.0 |
| | FGSM | CE | 95.6* | **72.8** | 32.8 | **24.6** | 26.0 | 42.0 | 39.6 | 18.0 | 10.1 |
| | | RCE | 95.0* | 67.0 | 27.6 | 18.2 | 21.1 | 38.3 | 34.4 | 17.0 | 7.9 |
| | Q-FGSM (Ours) | WACE | **96.4*** | 72.7 | **33.9** | 24.4 | **26.4** | **43.9** | **40.3** | **18.1** | **10.4** |
| | I-FGSM | CE | 99.4* | 87.9 | 27.8 | 18.4 | 15.0 | 37.3 | 37.3 | 7.8 | 3.9 |
| | | RCE | 100* | 82.2 | 18.4 | 12.2 | 10.4 | 28.7 | 30.4 | 7.0 | 3.6 |
| | QI-FGSM (Ours) | WACE | **100*** | **90.6** | **33.9** | **22.6** | **21.2** | **46.6** | **43.0** | **8.5** | **5.9** |
| | MI-FGSM | CE | 99.4* | 95.8 | 46.5 | 33.5 | 32.7 | 58.3 | 53.4 | 13.8 | 7.2 |
| | | RCE | 100* | 94.5 | 38.3 | 27.3 | 28.3 | 55.4 | 48.8 | 14.0 | 7.2 |
| VGG16 | QMI-FGSM (Ours) | WACE | **100*** | **96.5** | **52.4** | **39.5** | **36.7** | **65.0** | **58.0** | **16.1** | **9.2** |
| | DI-FGSM | CE | 99.6* | 94.1 | 38.1 | 26.2 | 27.0 | 52.3 | 47.5 | 8.8 | 5.4 |
| | | RCE | 99.9* | 90.9 | 27.6 | 17.7 | 20.2 | 44.8 | 40.2 | 9.0 | 4.5 |
| | QDI-FGSM (Ours) | WACE | **100*** | **94.9** | **45.6** | **33.5** | **34.0** | **61.0** | **53.8** | **10.7** | **7.8** |
| | SI-NI-FGSM | CE | 100* | 98.0 | 56.6 | 41.8 | 43.8 | 70.3 | 62.1 | 17.7 | 11.5 |
| | | RCE | 100* | **97.7** | 54.3 | 41.1 | 40.5 | 70.4 | 60.8 | 16.9 | 10.3 |
| | QSI-NI-FGSM (Ours) | WACE | **100*** | 97.5 | **64.5** | **51.6** | **47.3** | **75.4** | **67.3** | **21.1** | **13.9** |
| | VMI-FGSM | CE | 99.4* | 98.0 | 62.4 | 48.5 | 43.6 | 72.8 | 65.1 | 17.5 | 11.3 |
| | | RCE | 100* | 96.1 | 55.4 | 42.3 | 41.3 | 68.4 | 60.7 | 16.7 | 10.4 |
| | QVMI-FGSM (Ours) | WACE | **100*** | **98.1** | **69.2** | **54.6** | **50.5** | **78.2** | **70.1** | **19.8** | **14.5** |

query prior-based transfer attacks for CIFAR10, 0.69 to 0.82% on Q-FGSM and 3.69 to 11.18% on the other five query prior-based transfer attacks for CIFAR100, 0.7 to 1.3% on Q-FGSM and 4.6 to 7.9% on the other five query prior-based transfer attacks for ImageNet. In comparison with the RCE loss, the average increase of the ASR is 8.43 to 9.64% on CIFAR10 and 2.96 to 10.98% on the other five query prior-based transfer attacks for CIFAR10, 1.95 to 2.22% on Q-FGSM and 0.86 to 11.97% on the other five query prior-based transfer attacks for CIFAR100, 5.9 to 7.8% on Q-FGSM and 4.2 to 13.6% on the other five query prior-based transfer attacks for ImageNet.

In addition, as shown in Tables 4, 5, 6, 10, 11 and 12, when the attack strength $\epsilon = 16/255$, in comparison with the CE and RCE losses, the query prior-based attacks with the WACE loss can still effectively enhance the transferability of the gradient iterative-based attacks on different surrogate models and datasets.

Table 10: The untargeted attack success rates (%) on six naturally trained models for CIFAR10 using various transfer attacks and two query attacks with the attack strength $\epsilon = 16/255$. The adversarial examples are generated by VGG16. $*$ denotes the attack success rates under white-box attacks. **Average** means to calculate the average value except $*$. Note that $Q = 1$ in Q-FGSM.

| Model | Attack | Loss | V16 | V19 | R50 | WRN-16-4 | D121 | M-v2 | **Average** |
|---|---|---|---|---|---|---|---|---|---|
| | Square ($Q = 10$) | - | 44.80 | 42.55 | 43.50 | 57.75 | 58.10 | 65.05 | 51.96 |
| | PRGF ($Q = 10$) | - | 39.30* | 12.80 | 10.55 | 13.60 | 11.70 | 12.60 | 12.25 |
| | FGSM | CE | 70.50* | 64.65 | 65.70 | 67.60 | 61.75 | 76.25 | 67.19 |
| | | RCE | 66.35* | 62.05 | 63.00 | 64.75 | 59.40 | 74.20 | 64.68 |
| | Q-FGSM (Ours) | WACE | **77.70\*** | **67.50** | **69.75** | **71.45** | **67.25** | **79.35** | **71.06** |
| | I-FGSM | CE | 99.75* | 87.60 | 79.10 | 89.25 | 84.05 | 84.90 | 84.98 |
| | | RCE | 99.45* | 87.15 | 80.75 | 89.40 | 85.25 | 86.70 | 85.85 |
| | QI-FGSM (Ours) | WACE | **99.90\*** | **91.60** | **85.95** | **94.30** | **90.75** | **90.75** | **90.67** |
| | MI-FGSM | CE | 98.90* | 94.00 | 90.00 | 93.45 | 91.15 | 90.75 | 91.87 |
| | | RCE | 98.50* | 93.15 | 90.00 | 93.15 | 91.40 | 91.45 | 91.83 |
| VGG16 | QMI-FGSM (Ours) | WACE | **99.35\*** | **95.55** | **93.10** | **96.65** | **94.50** | **94.90** | **94.94** |
| | DI-FGSM | CE | 98.10* | 91.90 | 85.95 | 91.85 | 88.70 | 89.95 | 89.67 |
| | | RCE | 97.35* | 90.35 | 84.65 | 90.25 | 87.70 | 89.45 | 88.48 |
| | QDI-FGSM (Ours) | WACE | **99.00\*** | **94.25** | **91.50** | **96.00** | **94.35** | **95.60** | **94.34** |
| | SI-NI-FGSM | CE | 99.90* | 96.25 | 94.25 | 96.70 | 94.20 | 94.05 | 95.09 |
| | | RCE | 99.70* | 97.70 | 96.20 | 98.05 | 96.40 | 97.50 | 97.17 |
| | QSI-NI-FGSM (Ours) | WACE | **99.95\*** | **99.30** | **97.70** | **98.80** | **98.40** | **98.10** | **98.46** |
| | VMI-FGSM | CE | 99.85* | 97.35 | 95.10 | 96.80 | 95.40 | 96.15 | 96.16 |
| | | RCE | 99.50* | 97.30 | 96.25 | 97.40 | 96.85 | 97.25 | 97.01 |
| | QVMI-FGSM (Ours) | WACE | 99.80* | **98.70** | **97.80** | **99.25** | **98.35** | **99.00** | **98.62** |

Table 11: The untargeted attack success rates (%) on six naturally trained models for CIFAR100 using various transfer attacks and two query attacks with the attack strength $\epsilon = 16/255$. The adversarial examples are generated by VGG16. $*$ denotes the attack success rates under white-box attacks. **Average** means to calculate the average value except $*$. Note that $Q = 1$ in Q-FGSM.

| Model | Attack | Loss | V16 | R50 | RN50 | WRN-16-4 | D121 | M-v2 | **Average** |
|---|---|---|---|---|---|---|---|---|---|
| | Square ($Q = 10$) | - | 76.10 | 72.20 | 74.25 | 78.70 | 71.10 | 83.85 | 76.03 |
| | PRGF ($Q = 10$) | - | 63.65* | 15.45 | 18.35 | 17.60 | 14.30 | 14.85 | 16.11 |
| | FGSM | CE | 91.50* | 84.80 | 85.90 | 87.25 | 84.80 | 86.75 | 85.9 |
| | | RCE | 90.95* | 83.95 | 85.40 | 86.50 | 84.00 | 86.35 | 85.24 |
| | Q-FGSM (Ours) | WACE | 91.60* | **85.70** | **86.55** | **88.10** | **85.15** | **87.50** | **86.6** |
| | I-FGSM | CE | 100* | 68.25 | 74.50 | 78.40 | 71.35 | 69.60 | 72.42 |
| | | RCE | 100* | 66.75 | 73.50 | 78.60 | 72.20 | 74.25 | 73.06 |
| | QI-FGSM (Ours) | WACE | 100* | **77.85** | **83.40** | **86.30** | **81.30** | **79.40** | **81.65** |
| | MI-FGSM | CE | 100* | 87.95 | 88.50 | 91.10 | 87.35 | 86.15 | 88.21 |
| | | RCE | 100* | 88.85 | 90.15 | 92.85 | 89.50 | 89.50 | 90.17 |
| VGG16 | QMI-FGSM (Ours) | WACE | 100* | **92.65** | **92.20** | **95.35** | **92.00** | **90.85** | **92.61** |
| | DI-FGSM | CE | 99.75* | 77.55 | 81.25 | 86.25 | 80.10 | 79.50 | 80.93 |
| | | RCE | 99.70* | 78.40 | 82.75 | 86.30 | 81.10 | 83.00 | 82.31 |
| | QDI-FGSM (Ours) | WACE | **99.95\*** | **85.80** | **88.60** | **92.10** | **87.30** | **87.30** | **88.22** |
| | SI-NI-FGSM | CE | 99.85* | 88.90 | 90.40 | 92.45 | 89.40 | 89.60 | 90.15 |
| | | RCE | 100* | 91.10 | 92.55 | 93.30 | 92.10 | **93.40** | 92.49 |
| | QSI-NI-FGSM (Ours) | WACE | 100* | **93.10** | **92.85** | **95.95** | **93.30** | 92.75 | **93.59** |
| | VMI-FGSM | CE | **100\*** | 93.25 | 92.30 | 95.10 | 92.75 | 91.65 | 93.01 |
| | | RCE | 100* | 93.95 | 93.80 | 95.75 | 93.75 | 93.65 | 94.18 |
| | QVMI-FGSM (Ours) | WACE | 99.95* | **95.75** | **96.05** | **96.50** | **94.45** | **93.85** | **95.32** |

In conclusion, through the comparison with or without the query priors, at the low attack strength, i.e. $\epsilon = 8/255$, the query prior-based attacks can significantly enhance the transferability of adversarial examples to attack the naturally trained models. At the high attack strength, i.e. $\epsilon = 16/255$, most query prior-based attacks can enhance the transferability of adversarial examples, but the average ASR of QSI-NI-FGSM has a slight decrease on ImageNet with VGG16 as the surrogate model.

### B.2.2 ATTACKING AN ADVERSARIALLY TRAINED MODEL

Tables 3 and 9 perform six transfer attacks with or without the query priors to attack two adversarially trained models for ImageNet on different surrogate models when the attack strength $\epsilon = 8/255$. The results show that the query prior-based attacks with the WACE loss can enhance the transferability of the gradient iterative-based attacks when attacking the adversarially trained model. In comparison with the CE loss, the increase of the ASR is 0.1 to 0.3% on Q-FGSM and 0.7 to 6.3% on the other five transfer attacks for ImageNet (except for a slight decrease on Q-FGSM with ResNet50

Table 12: The untargeted attack success rates (%) on six naturally trained models and two adversarially trained models for ImageNet using various transfer attacks and two query attacks with the attack strength $\epsilon = 16/255$. The adversarial examples are generated by VGG16. $*$ denotes the attack success rates under white-box attacks. **Avg.** means to calculate the average value of the naturally trained models except $*$. Note that $Q = 1$ in Q-FGSM.

| Model | Attack | Loss | V16 | V19 | R50 | R152 | I-v3 | M-v2 | **Avg.** | a-I-v3 | ae-IR-v2 |
|-------|--------|------|-----|-----|-----|------|------|------|----------|--------|----------|
| VGG16 | Square ($Q = 0$) | - | 79.0 | 72.6 | 39.6 | 27.6 | 38.4 | 70.9 | 54.7 | 32.2 | 27.6 |
| | Square ($Q = 1$) | - | 79.4 | 73.3 | 41.1 | 29.8 | 39.7 | 71.7 | 55.8 | 34.8 | 29.1 |
| | Square ($Q = 10$) | - | 84.0 | 80.2 | 51.5 | 40.1 | 50.7 | 79.4 | 64.3 | 46.4 | 38.2 |
| | PRGF ($Q = 10$) | - | 94.5* | 38.9 | 10.3 | - | 5.7 | 13.1 | 17.0 | 5.8 | 2.1 |
| | FGSM | CE | 94.8* | **82.9** | 47.4 | **36.4** | **35.8** | 61.6 | **52.8** | 32.0 | **16.8** |
| | | RCE | 95.4* | 81.5 | 43.4 | 32.1 | 34.1 | 61.5 | 50.5 | 31.5 | 14.7 |
| | Q-FGSM (Ours) | WACE | **95.8*** | 82.4 | **47.9** | 35.9 | 34.8 | **62.8** | 52.8 | **32.7** | 15.9 |
| | I-FGSM | CE | 99.4* | 97.2 | 46.2 | 31.7 | 28.1 | 58.8 | 52.4 | 13.9 | 7.5 |
| | | RCE | 100* | 96.3 | 35.9 | 25.4 | 22.2 | 52.2 | 46.4 | 14.0 | 6.5 |
| | QI-FGSM (Ours) | WACE | **100*** | **97.6** | **53.4** | **39.2** | **32.4** | **66.4** | **57.8** | **15.4** | **8.2** |
| | MI-FGSM | CE | 99.4* | 98.7 | 69.0 | 54.2 | 51.1 | 79.8 | 70.6 | 24.5 | 14.6 |
| | | RCE | 100* | **99.2** | 66.7 | 50.9 | 48.2 | 81.1 | 69.2 | 24.2 | 13.0 |
| | QMI-FGSM (Ours) | WACE | **100*** | 99.1 | **73.0** | **60.7** | **54.9** | **81.7** | **73.9** | **25.5** | **16.1** |
| | DI-FGSM | CE | 99.6* | 98.6 | 60.2 | 44.8 | 44.6 | 73.8 | 64.4 | 15.6 | 9.2 |
| | | RCE | 100* | 97.9 | 50.2 | 38.1 | 37.5 | 68.3 | 58.4 | 14.4 | 8.3 |
| | QDI-FGSM (Ours) | WACE | **100*** | **98.7** | **67.1** | **53.3** | **50.1** | **80.9** | **70.0** | **17.6** | **12.3** |
| | SI-NI-FGSM | CE | 100* | **99.7** | **86.0** | **73.1** | **70.8** | **91.7** | **84.3** | 32.5 | 20.9 |
| | | RCE | 100* | 99.6 | 85.1 | 71.0 | 67.6 | 91.1 | 82.9 | 28.4 | 18.5 |
| | QSI-NI-FGSM (Ours) | WACE | **100*** | 99.5 | 84.7 | 72.3 | 68.4 | 90.2 | 83.0 | **33.5** | **22.5** |
| | VMI-FGSM | CE | 99.8* | 99.4 | 83.4 | 72.2 | 65.8 | 88.3 | 81.8 | 32.5 | 22.2 |
| | | RCE | 100* | 99.4 | 84.9 | 70.9 | 67.2 | 89.5 | 82.4 | 32.7 | 22.9 |
| | QVMI-FGSM (Ours) | WACE | **100*** | **99.7** | **87.2** | **78.4** | **70.3** | **90.6** | **85.2** | **37.3** | **28.0** |

Table 13: The untargeted attack success rates (%) on five naturally trained models for CIFAR10 using VMI-FGSM as the baseline with the attack strength $\epsilon = 8/255$. The adversarial examples are generated by VGG16 and ResNet50, which are the surrogate model and the query model, respectively. Note that ResNet50 is both the query model and the victim model.

| Surrogate Model | Query Model | Attack | Loss | V19 | R50 | WRN-16-4 | D121 | M-v2 | **Average** |
|-----------------|-------------|--------|------|-----|-----|----------|------|------|-------------|
| VGG16 | ResNet50 | VMI-FGSM | CE | 80.00 | 76.60 | 81.60 | 78.40 | 79.95 | 79.31 |
| | | | RCE | 77.30 | 75.50 | 79.10 | 76.65 | 78.75 | 77.46 |
| | | QVMI-FGSM(Ours) | WACE | **84.50** | **83.35** | **85.40** | **83.90** | **84.45** | **84.32** |

as the surrogate model). In comparison with the RCE loss, the increase of the ASR is 0.6 to 2.5% on Q-FGSM and 1.6 to 5.2% on the other five transfer attacks for ImageNet.

In addition, as shown in Tables 6 and 12, when the attack strength $\epsilon = 16/255$, in comparison with the CE and RCE losses, the query prior-based attacks with the WACE loss can still effectively enhance the transferability of the gradient iterative-based attacks to attack the adversarially trained models on different surrogate models (except for QSI-NI-FGSM with ResNet50 as the surrogate model to attack adversarial Inception-v3).

In conclusion, through the comparison with or without the query priors, at the low attack strength, i.e. $\epsilon = 8/255$, the query prior-based attacks except for Q-FGSM can enhance the transferability of adversarial examples to attack the adversarially trained models. At the high attack strength, i.e. $\epsilon = 16/255$, the query prior-based attacks except for Q-FGSM can enhance the transferability of adversarial examples to attack the adversarially trained models, but QSI-NI-FGSM reduces the attack success rate.

### B.2.3 ATTACKING THE OTHER MODELS

In all baseline methods, because VMI-FGSM has the highest overall performance, VMI-FGSM is selected as the baseline to further compare the transferability to the other models. Then, a surrogate model and a query model are used to generate adversarial examples to attack many other models where the query model is a target victim model queried by our query-prior based attack method. Here, VGG16 and ResNet50 (ResNet50 and VGG16) are used as the surrogate model and the query model respectively on CIFAR10/100 and ImageNet.

Table 14: The untargeted attack success rates (%) on five naturally trained models for CIFAR10 using VMI-FGSM as the baseline with the attack strength $\epsilon = 8/255$. The adversarial examples are generated by ResNet50 and VGG16, which are the surrogate model and the query model, respectively. Note that VGG16 is both the query model and the victim model.

| Surrogate Model | Query Model | Attack | Loss | V16 | V19 | WRN-16-4 | D121 | M-v2 | Average |
|---|---|---|---|---|---|---|---|---|---|
| ResNet50 | VGG16 | VMI-FGSM | CE | 80.40 | 82.20 | 94.60 | 90.80 | 88.95 | 87.39 |
| | | | RCE | 77.75 | 78.05 | 92.15 | 89.00 | 87.60 | 84.91 |
| | | QVMI-FGSM(Ours) | WACE | **88.60** | **87.10** | **96.30** | **93.75** | **92.60** | **91.67** |

Table 15: The untargeted attack success rates (%) on five naturally trained models for CIFAR100 using VMI-FGSM as the baseline with the attack strength $\epsilon = 8/255$. The adversarial examples are generated by VGG16 and ResNet50, which are the surrogate model and the query model, respectively. Note that ResNet50 is both the query model and the victim model.

| Surrogate Model | Query Model | Attack | Loss | R50 | RN50 | WRN-16-4 | D121 | M-v2 | Average |
|---|---|---|---|---|---|---|---|---|---|
| VGG16 | ResNet50 | VMI-FGSM | CE | 77.70 | 80.00 | 82.85 | 79.70 | 76.85 | 79.42 |
| | | | RCE | 79.20 | 80.20 | 83.90 | 80.30 | **79.65** | 80.65 |
| | | QVMI-FGSM(Ours) | WACE | **83.30** | **81.70** | **84.05** | **80.65** | 78.00 | **81.54** |

When VGG16 and ResNet50 are the surrogate model and the query model, respectively, as shown in Tables 13, 15 and 17, the attack success rate of QVMI-FGSM is almost higher than that of VMI-FGSM (with the CE loss). Specifically, the average increase of the ASR is 5.01% on CIFAR10, 2.12% on CIFAR100 and 1.78% on ImageNet. In addition, the ASR of QVMI-FGSM is higher than that of VMI-FGSM with the RCE loss.

When ResNet50 and VGG16 are the surrogate model and the query model, respectively, as shown in Tables 14, 16 and 18, the attack success rate of QVMI-FGSM is also almost higher than that of VMI-FGSM (with the CE loss). Specifically, the average increase of the ASR is 4.28% on CIFAR10, 0.64% on CIFAR100 and 3.42% on ImageNet. In addition, the ASR of QVMI-FGSM is also almost higher than that of VMI-FGSM with the RCE loss (except for CIFAR100 with ResNet50 as the surrogate model and VGG16 as the query model).

Overall, the adversarial examples generated by QVMI-FGSM not only perform better on the query model but also perform better on the other models.

*Why does the transferable attack success rate of the adversarial examples generated by the surrogate model and the query model improve on the other models?* Two points answer the question. First, there are similarities between models, which is also the reason why the adversarial examples have transferability. Second, attacking multiple models is similar to attacking an ensemble model. Because attacking the surrogate model and the query model at the same time is similar to attacking the ensemble model consisting of them and Liu et al. (2017) found that the adversarial examples generated by the ensemble model have higher transferability, QVMI-FGSM has higher transferability than VMI-FGSM to the other models.

### B.3 COMPARISON WITH OR WITHOUT THE FUZZY DOMAIN ELIMINATING TECHNIQUE ON THE UNTARGETED ATTACKS

#### B.3.1 ATTACKING A NATURALLY TRAINED MODEL

To validate that the fuzzy domain eliminating technique can enhance the transferability of the transfer attacks, we perform six transfer attacks with or without the fuzzy domain eliminating technique to attack six naturally trained models for CIFAR10, five gradient iterative-based attacks for CIFAR100, FGSM and three latest gradient iterative-based attacks (MI-FGSM, SI-NI-FGSM and VMI-FGSM) for ImageNet. Note that, according to Table 30, when $\mathcal{K} = 1$ and $\mathcal{T} = 1$, the FECE loss becomes the CE loss. As shown in Tables 1, 2 and 3, when the attack strength $\epsilon = 8/255$, our FECE loss can significantly enhance the transferability of the transfer attacks to attack the naturally trained models on different datasets. In comparison with the CE loss, the average increase of the ASR is

Table 16: The untargeted attack success rates (%) on five naturally trained models for CIFAR100 using VMI-FGSM as the baseline with the attack strength $\epsilon = 8/255$. The adversarial examples are generated by ResNet50 and VGG16, which are the surrogate model and the query model, respectively. Note that VGG16 is both the query model and the victim model.

| Surrogate Model | Query Model | Attack | Loss | V16 | RN50 | WRN-16-4 | D121 | M-v2 | **Average** |
|---|---|---|---|---|---|---|---|---|---|
| ResNet50 | VGG16 | VMI-FGSM | CE | 84.40 | 88.75 | 90.35 | 89.25 | 80.70 | 86.69 |
| | | | RCE | 84.00 | **90.45** | **91.20** | **90.55** | **82.70** | **87.78** |
| | | QVMI-FGSM(Ours) | WACE | **87.05** | 89.20 | 90.50 | 90.10 | 79.80 | 87.33 |

Table 17: The untargeted attack success rates (%) on five naturally trained models for ImageNet using VMI-FGSM as the baseline with the attack strength $\epsilon = 8/255$. The adversarial examples are generated by VGG16 and ResNet50, which are the surrogate model and the query model, respectively. Note that ResNet50 is both the query model and the victim model.

| Surrogate Model | Query Model | Attack | Loss | V19 | R50 | R152 | I-v3 | M-v2 | **Average** |
|---|---|---|---|---|---|---|---|---|---|
| VGG16 | ResNet50 | VMI-FGSM | CE | **98.0** | 62.4 | 48.5 | 43.6 | **72.8** | 65.06 |
| | | | RCE | 96.1 | 55.4 | 42.3 | 41.3 | 68.4 | 60.7 |
| | | QVMI-FGSM(Ours) | WACE | 97.6 | **69.2** | **50.8** | **44.1** | 72.5 | **66.84** |

6.03% on FGSM and 2.31 to 7.25% on the other five gradient iterative-based attacks for CIFAR10, 1.3 to 10.27% on the five gradient iterative-based attacks for CIFAR100, 0.4% on FGSM and 2.3 to 3.4% on the latest gradient iterative-based attacks (SI-NI-FGSM and VMI-FGSM) for ImageNet. In comparison with the RCE loss, the average increase of the ASR is 12.69% on FGSM and 0.64 to 11.14% on the other five gradient iterative-based attacks for CIFAR10, 0.21 to 0.6% on several gradient iterative-based attacks (MI-FGSM, DI-FGSM and VMI-FGSM) for CIFAR100 (the average ASR is kept on I-FGSM and SI-NI-FGSM), 6.9% on FGSM and 1.2 to 3.9% on the latest gradient iterative-based attacks for ImageNet.

In addition, as shown in Tables 4, 5 and 6, when the attack strength $\epsilon = 16/255$, in comparison with the CE loss, our FECE loss based attacks can still effectively enhance the transferability of the gradient iterative-based attacks on different datasets. In comparison with the RCE loss, our FECE loss based attacks can also still effectively enhance the transferability of the gradient iterative-based attacks on CIFAR10 and ImageNet, and keep the transferability of the gradient iterative-based attacks on CIFAR100.

In conclusion, through the comparison with or without the fuzzy domain eliminating technique, at the low attack strength, i.e. $\epsilon = 8/255$, our FECE loss can effectively enhance the transferability of adversarial examples to attack the naturally trained models on different datasets. At the high attack strength, i.e. $\epsilon = 16/255$, our FECE loss can effectively enhance the transferability of adversarial examples to attack the naturally trained models on CIFAR10 and ImageNet, and keep the transferability of adversarial examples on CIFAR100.

### B.3.2 ATTACKING AN ADVERSARIALLY TRAINED MODEL

Table 3 performs the three latest transfer attacks (MI-FGSM, SI-NI-FGSM and VMI-FGSM) with or without the fuzzy domain eliminating technique to attack two adversarially trained models for ImageNet when the attack strength $\epsilon = 8/255$. In comparison with different loss functions (the CE and RCE losses), our FECE loss can enhance the transferability of VMI-FGSM and keep (or slightly decrease) the transferability of the other transfer attacks.

In addition, as shown in Table 6, when the attack strength $\epsilon = 16/255$, in comparison with the CE loss, our FECE loss can enhance the transferability of FGSM, MI-FGSM and VMI-FGSM, and keep the transferability of SI-NI-FGSM. In comparison with the RCE loss, our FECE loss can enhance the transferability of FGSM and SI-NI-FGSM, and keep (or slightly decrease) on MI-FGSM and VMI-FGSM.

In conclusion, through the comparison with or without the fuzzy domain eliminating technique, at the low attack strength, i.e. $\epsilon = 8/255$, our FECE loss can steadily improve the transferability of

Table 18: The untargeted attack success rates (%) on five naturally trained models for ImageNet using VMI-FGSM as the baseline with the attack strength $\epsilon = 8/255$. The adversarial examples are generated by ResNet50 and VGG16, which are the surrogate model and the query model, respectively. Note that VGG16 is both the query model and the victim model.

| Surrogate Model | Query Model | Attack | Loss | V16 | V19 | R152 | I-v3 | M-v2 | **Average** |
|---|---|---|---|---|---|---|---|---|---|
| ResNet50 | VGG16 | VMI-FGSM | CE | 69.1 | 68.5 | 83.9 | **53.1** | 72.1 | 69.34 |
| | | | RCE | 69.9 | 70.9 | 79.5 | 52.1 | 71.8 | 68.84 |
| | | QVMI-FGSM(Ours) | WACE | **78.6** | **72.5** | **85.2** | 52.5 | **75.0** | **72.76** |

Table 19: The untargeted attack success rates (%) on adversarial Inception-v3 for ImageNet using various transfer attacks and a query attack with the attack strength $\epsilon = 8/255$. The adversarial examples are generated by adversarial ensemble Inception-Resnet-v2. Note that $Q = 1$ in Q-FGSM and $Q = 5$ in the other attacks.

| | | | | a-I-v3 | | | |
|---|---|---|---|---|---|---|---|
| Attack | Square | Q-FGSM | QI-FGSM | QMI-FGSM | QDI-FGSM | QSI-NI-FGSM | QVMI-FGSM |
| ae-IR-v2 | 16.9 | 23.2 | 13.4 | 22.9 | 17.7 | 24.5 | **28.7** |

VMI-FGSM. At the high attack strength, i.e. $\epsilon = 16/255$, the CE, RCE and FECE losses have their own advantages on different transfer attacks.

### B.3.3 COMBINATION OF THE QUERY PRIORS AND FUZZY DOMAIN ELIMINATING TECHNIQUE

As shown in Tables 1, 2, 3, 4, 5 and 6, whether the attack strength $\epsilon = 8/255$ or $16/255$, when attacking the naturally trained model, in comparison with our WACE and FECE losses, our WFCE loss can further improve the transferability of the gradient iterative-based attacks on different datasets. In addition, when attacking the adversarially trained model and the attack strength $\epsilon = 16/255$, in comparison with our WACE and FECE losses, our WFCE loss can further improve the transferability of the latest VMI-FGSM on ImageNet.

### B.4 COMPARISON WITH CURRENT BLACK-BOX QUERY ATTACKS ON THE UNTARGETED ATTACKS

As shown in Tables 1, 2, 3, 4, 5, 6, 7, 8, 9, 10, 11 and 12, whether the attack strength $\epsilon = 8/255$ or $16/255$, when the allowed query number $Q = 10$, the attack success rate of our QVMI-FGSM is much larger than that of Square and PRGF when attacking the naturally trained models on different surrogate models and datasets.

As shown in Tables 3, 6, 9 and 12, whether the attack strength $\epsilon = 8/255$ or $16/255$, when the allowed query number $Q = 10$, the attack success rate of our QVMI-FGSM is much larger than that of PRGF when attacking the adversarially trained models on different surrogate models for ImageNet.

As shown in Table 3, when the attack strength $\epsilon = 8/255$ and the allowed query number $Q = 10$, the attack success rate of our QVMI-FGSM is larger than that of Square when attacking the adversarially trained models on ImageNet.

To highlight the advantages of the query prior-based attacks for attacking the adversarially trained models when compared with Square, we set adversarial ensemble Inception-Resnet-v2 as the surrogate model rather than the naturally trained models (i.e. VGG16 or ResNet50) and adversarial Inception-v3 as the victim model, and reduce the query number from 10 to 5 (i.e. $Q = 5$). As shown in Table 19, when the attack strength is 8/255, by comparing the best query prior-based attacks with Square, the increase of the ASR is 11.8%. As shown in Table 20, when the attack strength is 16/255, the increase of the ASR is 7.5%.

In conclusion, (i) through the comparison with Square, whether the attack strength is low or high, the ASR of the query prior-based attacks is far greater than that of Square for attacking six naturally trained models. When attacking two adversarially trained models, at the low attack strength, i.e.

Table 20: The untargeted attack success rates (%) on adversarial Inception-v3 for ImageNet using various transfer attacks and a query attack with the attack strength $\epsilon = 16/255$. The adversarial examples are generated by adversarial ensemble Inception-Resnet-v2. Note that $Q = 1$ in Q-FGSM and $Q = 5$ in the other attacks.

| | | | | a-I-v3 | | | |
|---|---|---|---|---|---|---|---|
| Attack | Square | Q-FGSM | QI-FGSM | QMI-FGSM | QDI-FGSM | QSI-NI-FGSM | QVMI-FGSM |
| ae-IR-v2 | 39.4 | 41.3 | 21.4 | 36.3 | 28.3 | 38.5 | **46.9** |

Table 21: The untargeted attack success rates (%) on five naturally trained models for CIFAR10 using I-FGSM as the baseline with the attack strength $\epsilon = 8/255$. VGG16 is the surrogate model and the query number of QI-FGSM is 1, i.e., $Q = 1$.

| Model | Attack | Loss | V19 | R50 | WRN-16-4 | D121 | M-v2 | **Average** |
|---|---|---|---|---|---|---|---|---|
| VGG16 | I-FGSM | CE | 68.95 | 61.45 | 72.65 | 68.30 | 70.90 | 68.45 |
| | | RCE | 68.10 | 60.80 | 72.20 | 68.10 | 70.65 | 67.97 |
| | QI-FGSM(Q=1) | WACE | **74.60** | **68.70** | **81.50** | **76.55** | **80.70** | **76.41** |

$\epsilon = 8/255$, some query prior-based attacks are better than Square ($Q = 10$). At the high attack strength, i.e. $\epsilon = 16/255$, Square ($Q = 10$) is better than the query prior-based attacks and Square ($Q = 0$) is better than the transfer attacks (i.e., the family of FGSMs). However, when we use the adversarially trained model as the surrogate model and reduce the query number, regardless of whether the attack strength is low or high, the ASR of the query prior-based attacks is greater than that of Square for attacking the other adversarially trained models. (ii) P-RGF is inefficient at limits of a few queries on six naturally trained models and two adversarially trained models for CIFAR10/100 and ImageNet. The ASR of the query prior-based attacks is far greater than that of P-RGF.

## B.5 ABLATION STUDY ON THE UNTARGETED ATTACKS

### B.5.1 DIFFERENT NUMBERS OF THE TOP-N WRONG CATEGORIES

Figures 5 and 7 respectively evaluate the effect of different $\bar{n}$ on the attack success rates of five naturally trained victim models and two adversarially trained victim models when these victim models are attacked by QI-FGSM ($\epsilon = 8/255$) with VGG16 and ResNet50 for CIFAR10/100 and ImageNet. As shown in Figures 5 and 7, when $\bar{n}$ is greater than a certain threshold, the attack success rate will not be improved, e.g., Figure 5 shows that the threshold is 2 for CIFAR10, 10 for CIFAR100 and 5 for ImageNet approximately, and Figure 7 shows that the threshold is 2 for CIFAR10, 5 for CIFAR100 and 10 for ImageNet approximately. Because increasing $\bar{n}$ increases the calculation time of the gradient, $\bar{n}$ is not the bigger the better.

### B.5.2 DIFFERENT QUERY NUMBERS

Figures 6 and 8 respectively evaluate the effect of different $Q$ on the attack success rates of five naturally trained victim models and two adversarially trained victim models when these victim models are attacked by QI-FGSM ($\epsilon = 8/255$) with VGG16 and ResNet50 for CIFAR10/100 and ImageNet. On the victim models, the more the query, the greater the attack success rate. As shown in Figure 6, when the query number increases from 1 to 10, the attack success rate increases by 3.5% at most for CIFAR10, 10% at most for CIFAR100 and 5% at most for ImageNet approximately, and the increased attack success rate is mainly increased in the first five queries. As shown in Figure 8, when the query number increases from 1 to 10, the attack success rate increases by 3% at most for CIFAR10, 8% at most for CIFAR100 and 5% at most for ImageNet approximately, and the increased attack success rate is mainly increased in the first five queries.

### B.5.3 COMPARISON WITH OR WITHOUT THE QUERY PRIORS WHEN $Q = 1$

When $Q = 0$, the query prior-based methods will be transformed into the usual methods, e.g., QI-FGSM $\rightarrow$ I-FGSM. To further explore the effectiveness of the query-prior based method, we set the query number $Q$ as 1 and I-FGSM is selected as the baseline.

Table 22: The untargeted attack success rates (%) on five naturally trained models for CIFAR10 using I-FGSM as the baseline with the attack strength $\epsilon = 8/255$. ResNet50 is the surrogate model and the query number of QI-FGSM is 1, i.e., $Q = 1$.

| Model | Attack | Loss | V16 | V19 | WRN-16-4 | D121 | M-v2 | **Average** |
|---|---|---|---|---|---|---|---|---|
| ResNet50 | I-FGSM | CE | 59.85 | 62.55 | 93.95 | 88.70 | 84.35 | 77.88 |
| | | RCE | 58.05 | 60.10 | 90.60 | 86.15 | 82.25 | 75.43 |
| | QI-FGSM(Q=1) | WACE | **68.45** | **68.55** | **96.85** | **93.25** | **89.60** | **83.34** |

Table 23: The untargeted attack success rates (%) on five naturally trained models for CIFAR100 using I-FGSM as the baseline with the attack strength $\epsilon = 8/255$. VGG16 is the surrogate model and the query number of QI-FGSM is 1, i.e., $Q = 1$.

| Model | Attack | Loss | R50 | RN50 | WRN-16-4 | D121 | M-v2 | **Average** |
|---|---|---|---|---|---|---|---|---|
| VGG16 | I-FGSM | CE | 50.70 | 58.70 | 61.15 | 54.10 | 50.65 | 55.06 |
| | | RCE | 47.25 | 56.30 | 60.00 | 54.10 | 52.90 | 54.11 |
| | QI-FGSM(Q=1) | WACE | **58.95** | **68.35** | **70.55** | **64.25** | **58.95** | **64.21** |

As shown in Tables 21, 22, 23, 24, 25 and 26, even if the number of query $Q$ is 1, the query prior-based method can still significantly improve the transferability of the baseline method on different surrogate models and different datasets with attack strength $\epsilon = \frac{8}{255}$.

### B.5.4 FURTHER VERIFY THE EFFECTIVENESS OF THE QUERY PRIOR-BASED ATTACKS

To further verify the effectiveness of the query prior-based attacks, we make a more fair comparison that QVMI-FGSM compared with the combination of VMI-FGSM and Square. As shown in Tables 27, 28 and 29, when the attack strength $\epsilon = 8/255$ and the allowed query number $Q = 10$, the results of our QVMI-FGSM have higher performance than the combination of VMI-FGSM and Square to attack five naturally trained models on CIFAR10/100 and ImageNet.

### B.5.5 DIFFERENT SIZES OF THE TEMPERATURE PARAMETER

Figure 9 evaluates the effect of different $\mathcal{K}$ on the attack success rates of ResNet50 to VGG16 using various transfer attacks for CIFAR10/100 and ImageNet. As shown in Figure 9, with the increase of $\mathcal{K}$, the attack success rates of the gradient iterative-based attacks are significantly increased on CIFAR10 except for SI-NI-FGSM. When the $\mathcal{K}$ increases to 2, the performance of all gradient iterative-based attacks is almost optimal on CIFAR10.

### B.5.6 DIFFERENT SIZES OF THE PENALTY PARAMETER

Figure 10 evaluates the effect of different $\mathcal{T}$ on the attack success rates of ResNet50 to VGG16 using various transfer attacks for CIFAR10/100 and ImageNet. As shown in Figure 10, with the increase of $\mathcal{T}$, the attack success rate of the SI-NI-FGSM is increased on CIFAR10, the attack success rates of all gradient iterative-based attacks are significantly increased on CIFAR100 and the attack success rates of the latest gradient iterative-based attacks (MI-FGSM, SI-NI-FGSM and VMI-FGSM) are increased by a reasonable $\mathcal{T}$ on ImageNet.

In addition, Figure 11 further explores the optimal parameter combinations of $\mathcal{K}$ and $\mathcal{T}$ on different datasets, which are summarized in Table 30.

### B.6 COMPARISON WITH OR WITHOUT THE FUZZY DOMAIN ELIMINATING TECHNIQUE ON THE TARGETED ATTACKS

As shown in Figure 12, slightly decreasing $\mathcal{K}$ from 1 can slightly increase the targeted attack success rates of several gradient iterative-based attacks on CIFAR10/100. As shown in Figure 13, with the increase of the $\mathcal{T}$, the targeted attack success rates of almost all the FECE ($\mathcal{K} = 1$) based attacks are increased and close to that of the RCE based attacks, which are theoretically analyzed in Propositions 4 and 7.

Table 24: The untargeted attack success rates (%) on five naturally trained models for CIFAR100 using I-FGSM as the baseline with the attack strength $\epsilon = 8/255$. ResNet50 is the surrogate model and the query number of QI-FGSM is 1, i.e., $Q = 1$.

| Model | Attack | Loss | V16 | RN50 | WRN-16-4 | D121 | M-v2 | **Average** |
|---|---|---|---|---|---|---|---|---|
| ResNet50 | I-FGSM | CE | 61.45 | 81.40 | 82.05 | 81.10 | 58.05 | 72.81 |
| | | RCE | **69.45** | 84.10 | 85.80 | 84.35 | **68.00** | 78.34 |
| | QI-FGSM(Q=1) | WACE | 68.70 | **86.05** | **88.45** | **86.75** | 67.15 | **79.42** |

Table 25: The untargeted attack success rates (%) on five naturally trained models for ImageNet using I-FGSM as the baseline with the attack strength $\epsilon = 8/255$. VGG16 is the surrogate model and the query number of QI-FGSM is 1, i.e., $Q = 1$.

| Model | Attack | Loss | V19 | R50 | R152 | I-v3 | M-v2 | **Average** |
|---|---|---|---|---|---|---|---|---|
| VGG16 | I-FGSM | CE | **87.9** | 27.8 | 18.4 | 15.0 | 37.3 | 37.3 |
| | | RCE | 82.2 | 18.4 | 12.2 | 10.4 | 30.4 | 30.4 |
| | QI-FGSM(Q=1) | WACE | 87.4 | **31.1** | **21.7** | **20.2** | **42.4** | **40.6** |

**Proposition 7** *With the increase of the $\mathcal{T}$, the targeted attack success rates of almost all the FECE ($\mathcal{K} = 1$) based attacks are close to that of the RCE based attacks.*

**Proof** *In the targeted attacks, the RCE and FECE losses respectively are Eq. 49 and 50:*

$$L_{RCE}(x, y_\tau; \theta) = -L_{CE}(x, y_\tau; \theta) + \frac{1}{C} \sum_{c=1}^{C} L_{CE}(x, y_c; \theta) \tag{49}$$

$$L_{FECE}(x, y_\tau; \theta, \mathcal{T}, 1) = \log FESoftmax(z_\tau; \mathcal{T}, 1) \tag{50}$$

*where $\mathcal{K} = 1$ in the FECE loss. To explore the targeted attack success rates of almost all the FECE ($\mathcal{K} = 1$) based attacks are close to that of the RCE based attacks with the increase of the $\mathcal{T}$, the derivation formula of $L_{RCE}(x, y_\tau; \theta)$ and $L_{FECE}(x, y_\tau; \theta, \mathcal{T}, 1)$ w.r.t. the input x respectively are Eq. 51 and 52:*

$$\frac{\partial L_{RCE}}{\partial x} = \frac{\partial L_{RCE}}{\partial z_\tau} \cdot \left( \frac{\partial z_\tau}{\partial x} \right) + \sum_{i=1(i \neq \tau)}^{C} \frac{\partial L_{RCE}}{\partial z_i} \cdot \frac{\partial z_i}{\partial x}$$

$$= \frac{1}{\ln 2} \cdot \left( \left( \frac{\partial z_\tau}{\partial x} \right) - \sum_{i=1}^{C} \frac{1}{C} \cdot \frac{\partial z_i}{\partial x} \right) \tag{51}$$

$$\frac{\partial L_{FECE}}{\partial x} = \frac{\partial L_{FECE}}{\partial z_\tau} \cdot \left( \frac{\partial z_\tau}{\partial x} \right) + \sum_{i=1(i \neq \tau)}^{C} \frac{\partial L_{FECE}}{\partial z_i} \cdot \frac{\partial z_i}{\partial x}$$

$$= \frac{1}{\mathcal{T} \cdot \ln 2} \cdot \left( \left( 1 - \frac{e^{z_\tau/\mathcal{T}}}{\sum_{i=1}^{C} e^{z_i/\mathcal{T}}} \right) \cdot \left( \frac{\partial z_\tau}{\partial x} \right) - \sum_{i=1(i \neq \tau)}^{C} \frac{e^{z_i/\mathcal{T}}}{\sum_{j=1}^{C} e^{z_j/\mathcal{T}}} \cdot \frac{\partial z_i}{\partial x} \right). \tag{52}$$

*In Eq. 52, with the increase of $\mathcal{T}$, when $\mathcal{T} \to +\infty$, $\left( 1 - \frac{e^{z_\tau/\mathcal{T}}}{\sum_{i=1}^{C} e^{z_i/\mathcal{T}}} \right) \approx \frac{C-1}{C}$ and $\frac{e^{z_i/\mathcal{T}}}{\sum_{j=1}^{C} e^{z_j/\mathcal{T}}} \approx \frac{1}{C}$. When $C$ is large, $\frac{C-1}{C} \approx 1$. Then,*

$$sign\left( \frac{\partial L_{RCE}}{\partial x} \right) \approx sign\left( \frac{\partial L_{FECE}}{\partial x} \right) \tag{53}$$

*where $sign(\cdot)$ is the sign function. Therefore, with the increase of the $\mathcal{T}$, the targeted attack success rates of almost all the FECE ($\mathcal{K} = 1$) based attacks are close to that of the RCE based attacks.*

*Therefore, according to Proposition 4, the high performance of the RCE loss in the targeted transfer attacks can be explained as the fuzzy domain eliminating technique.*

Table 26: The untargeted attack success rates (%) on five naturally trained models for ImageNet using I-FGSM as the baseline with the attack strength $\epsilon = 8/255$. ResNet50 is the surrogate model and the query number of QI-FGSM is 1, i.e., $Q = 1$.

| Model | Attack | Loss | V16 | V19 | R152 | I-v3 | M-v2 | Average |
|---|---|---|---|---|---|---|---|---|
| ResNet50 | I-FGSM | CE | 32.1 | 29.7 | 45.0 | 17.0 | 33.2 | 31.4 |
| | | RCE | 28.9 | 28.0 | 35.2 | 16.2 | 29.7 | 27.6 |
| | QI-FGSM(Q=1) | WACE | **38.1** | **33.3** | **49.1** | **20.8** | **41.0** | **36.5** |

Table 27: The untargeted attack success rates (%) on five naturally trained models for CIFAR10 using the combination of VMI-FGSM and Square as the baseline with the attack strength $\epsilon = 8/255$. VGG16 is the surrogate model and the query number of QVMI-FGSM and Square is 10, i.e., $Q = 10$.

| Model | Attack | Loss | V19 | R50 | WRN-16-4 | D121 | M-v2 | Average |
|---|---|---|---|---|---|---|---|---|
| VGG16 | VMI-FGSM & Square | CE | 80.05 | 77.40 | 81.65 | 79.15 | 80.35 | 79.72 |
| | | RCE | 77.45 | 75.35 | 79.25 | 77.25 | 79.85 | 77.83 |
| | QVMI-FGSM(Ours) | WACE | **84.40** | **83.35** | **88.40** | **85.70** | **89.10** | **86.19** |

## B.7 LIMITATION

The query prior-based attacks are effective for the untargeted attack. However, because Proposition 2 is more conducive to the exploration of the untargeted attack than the targeted attack, the proposed query prior-based attacks are designed as the untargeted attacks, which may not work in the targeted attack. In the future, the design of the query prior-based targeted attack is still a problem that needs to be studied.

## C THE DETAILED CONTRIBUTION INTRODUCTION OF THE QUERY PRIOR-BASED ATTACKS

The detailed contributions of the query prior-based attacks as follows.

First, we propose Proposition 1 and Corollary 1, which explore the relationship between the $\cos \vartheta$ ($\vartheta$ is the gradient angle between the surrogate model and the victim model) and the transferability on the same surrogate model and victim model pair using different transfer attack methods. In addition, we propose Proposition 2, which finds the preference property of deep neural networks. The Theoretical and Empirical Proofs of Propositions 1, 2 and Corollary 1 are represented in the appendices A.1 and A.2.

Second, by utilizing Propositions 1, 2 and Corollary 1, we designed a simple WACE loss function. Theorem 1 and Figure 4 proved that the WACE loss is better than CE and RCE losses on reducing the gradient angle between the surrogate model and victim model. Based on the WACE loss, we designed the query prior-based attacks, which solved two problems. First, compared with several latest transfer attack methods, the query prior-based attacks significantly improve the transferable attack success rate on the target victim model for CIFAR10/100 and ImageNet, and effectively improve the transferable attack success rate on the other models for CIFAR10/100 and ImageNet. Second, compared with two latest effective query attack methods, when the number of query is reduced to 10, the attack success rate of our QVMI-FGSM still remains high and is much higher than them.

Third, as far as we know, our query-prior based attack method is the first try to solve the problem of black-box attack that allows a few queries (i.e., less or equal to 10).

## D THE DETAILED MOTIVATION OF THE DESIGNED QUERY PRIOR-BASED ATTACKS

To the best of our knowledge, we can divide the current black-box attacks into three scenarios.

Table 28: The untargeted attack success rates (%) on five naturally trained models for CIFAR100 using the combination of VMI-FGSM and Square as the baseline with the attack strength $\epsilon = 8/255$. VGG16 is the surrogate model and the query number of QVMI-FGSM and Square is 10, i.e., $Q = 10$.

| Model | Attack | Loss | R50 | RN50 | WRN-16-4 | D121 | M-v2 | **Average** |
|-------|--------|------|-----|------|----------|------|------|-------------|
| VGG16 | VMI-FGSM & Square | CE | 80.25 | 82.05 | 84.80 | 81.95 | 80.25 | 81.86 |
| | | RCE | 80.25 | 81.50 | 85.05 | 82.30 | 81.30 | 82.08 |
| | QVMI-FGSM(Ours) | WACE | **83.30** | **85.50** | **87.75** | **84.40** | **81.40** | **84.47** |

Table 29: The untargeted attack success rates (%) on five naturally trained models for ImageNet using the combination of VMI-FGSM and Square as the baseline with the attack strength $\epsilon = 8/255$. ResNet50 is the surrogate model and the query number of QVMI-FGSM and Square is 10, i.e., $Q = 10$.

| Model | Attack | Loss | V16 | V19 | R152 | I-v3 | M-v2 | **Average** |
|-------|--------|------|-----|-----|------|------|------|-------------|
| ResNet50 | VMI-FGSM & Square | CE | 74.9 | 74.0 | 84.7 | 56.4 | 75.1 | 73.0 |
| | | RCE | 75.1 | 74.6 | 80.1 | 53.9 | 75.6 | 71.9 |
| | QVMI-FGSM(Ours) | WACE | **78.6** | **74.8** | **88.2** | **60.8** | **82.3** | **76.9** |

**The first scenario** is the query-free transfer-based attack, i.e., the allowed number of query $Q = 0$. The adversarial examples are generated by the surrogate model without any knowledge of any target model. For example, the current transfer-based attacks are the query-free transfer-based attack, i.e., FGSM, I-FGSM, MI-FGSM, DI-FGSM, SI-NI-FGSM and VMI-FGSM.

**The second scenario** is the query-based attack without transfer prior, i.e., a sufficient number of query and without transfer prior. The adversarial examples are generated by gradient estimation or random search. For example, a typical effective algorithm is Square.

**The third scenario** is the query-based attack with transfer prior, i.e., a sufficient number of query and with transfer prior. The adversarial examples are generated by the combination of the transfer prior and gradient estimation (or random search) where the transfer prior is used to improve the efficiency of gradient estimation and reduce the number of query. For example, a typical effective algorithm is PRGF.

In our paper, we explore a novel scenario, i.e., the fourth scenario. **The fourth scenario** is the transfer-based attack with a few queries, i.e., the allowed number of query $Q \leq 10$. The adversarial examples are generated by the surrogate model with a few query outputs of a target victim model (the number of query $Q \leq 10$).

The fourth scenario is reasonable, and there is no black-box attack algorithm specifically belonging to the fourth scenario at present. *Why is the fourth scenario reasonable?* There are two reasons to answer the question and the two reasons are also the problems that existed in the first, second and third scenarios.

First, in the second and third scenarios, although the number of queries in the current query-based attacks is decreasing, it still needs hundreds of queries. Even if the number of query $Q \leq 10$, the attack success rates of the query-based attacks with or without transfer prior are significantly reduced, and are far lower than the query-free transfer-based attack in the first scenario, which can be found in our experimental results.

Second, Proposition 1 and Corollary 1 of our paper explore the reason why the attack success rate of the current query-free transfer-based attack in the first scenario is increasing (i.e., when the step size $\alpha$ is small, the better the transferability of the transfer-based attack, the smaller the gradient angle $\vartheta$ between the surrogate model and the victim model). To reduce the angle $\vartheta$ for improving the transferability, Proposition 2 of our paper explores the preference of deep neural network implemented classification models after being attacked by the gradient-based attack algorithm (i.e., the successful attacked adversarial examples prefer to be classified as the wrong categories with higher probability). By utilizing Propositions 1, 2 and Corollary 1, we can design an algorithm to reduce the angle $\vartheta$ for enhancing the transferability of the generated adversarial examples with a few query outputs of a target victim model.

Table 30: The optimal parameter of the FECE loss on the combination of different methods and datasets with ResNet50 as the surrogate model.

|  | CIFAR10 | | CIFAR100 | | ImageNet | |
| --- | --- | --- | --- | --- | --- | --- |
|  | $\mathcal{T}$ | $\mathcal{K}$ | $\mathcal{T}$ | $\mathcal{K}$ | $\mathcal{T}$ | $\mathcal{K}$ |
| FGSM | 1.0 | 2.0 | 1.0 | 1.0 | 1.0 | 1.02 |
| BIM | 1.0 | 3.5 | 4.5 | 1.15 | 1.0 | 1.0 |
| MIFGSM | 1.0 | 3.0 | 3.0 | 1.05 | 3.5 | 1.015 |
| DIFGSM | 1.0 | 2.5 | 3.5 | 1.0 | 1.0 | 1.0 |
| SINIFGSM | 5.5 | 1.0 | 3.5 | 1.0 | 2.0 | 1.0 |
| VMIFGSM | 1.0 | 2.5 | 2.0 | 1.05 | 2.0 | 1.0 |

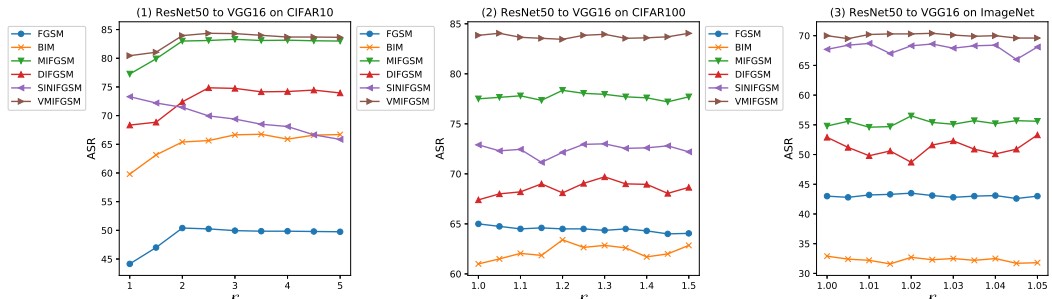

Figure 9: The untargeted attack success rates (%) of ResNet50 to VGG16 using various transfer attacks for CIFAR10/100 and ImageNet when varying the size of $\mathcal{K}$ ($\mathcal{K} > 1$) in the FECE loss. Note that $\mathcal{T}$ in the FECE loss is 1.

Therefore, by utilizing Propositions 1, 2 and Corollary 1, we design a simple WACE loss function. Theorem 1 and Figure 4 prove that the WACE loss is better than the CE and RCE losses on reducing the gradient angle between the surrogate model and the victim model. Based on the WACE loss, we design the query prior-based attacks, which solves the above two problems and are verified by the extended experiments. Overall, in the fourth scenario (i.e., the allowed number of query $Q \leq 10$), our method has the highest attack success rate when compared with the current black-box attacks.

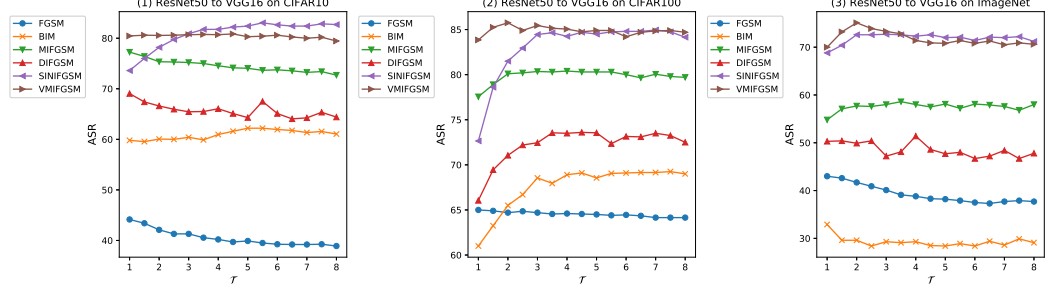

Figure 10: The untargeted attack success rates (%) of ResNet50 to VGG16 using various transfer attacks for CIFAR10/100 and ImageNet when varying the size of $\mathcal{T}$ ($\mathcal{T} > 1$) in the FECE loss. Note that $\mathcal{K}$ in the FECE loss is 1.

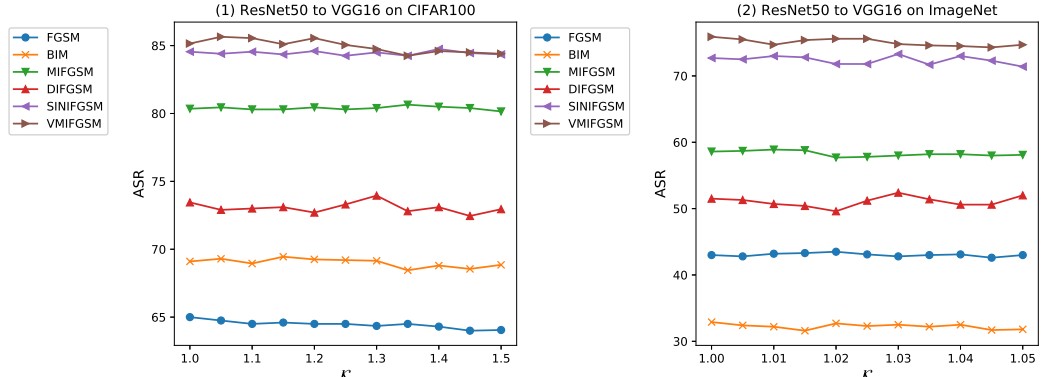

Figure 11: The untargeted attack success rates (%) of ResNet50 to VGG16 using various transfer attacks for CIFAR10/100 and ImageNet when varying the size of $\mathcal{K}$ ($\mathcal{K} > 1$) in the FECE loss. Note that $\mathcal{T}$ in the FECE loss is the optimal value.

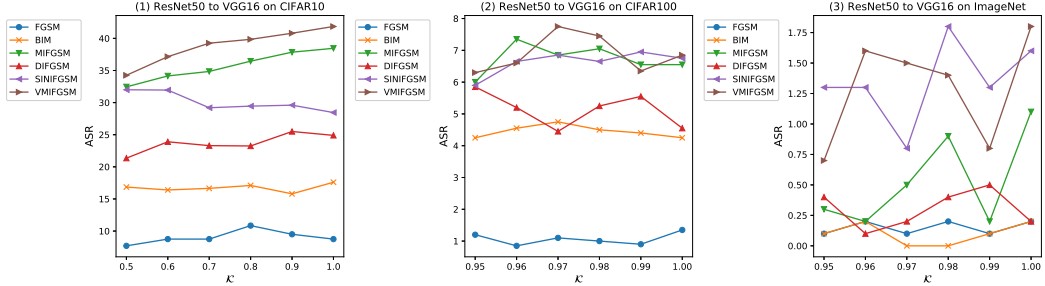

Figure 12: The targeted attack success rates (%) of ResNet50 to VGG16 using various transfer attacks for CIFAR10/100 and ImageNet when varying the size of $\mathcal{K}$ ($0 < \mathcal{K} < 1$) in the FECE loss. Note that $\mathcal{T}$ in the FECE loss is 1.

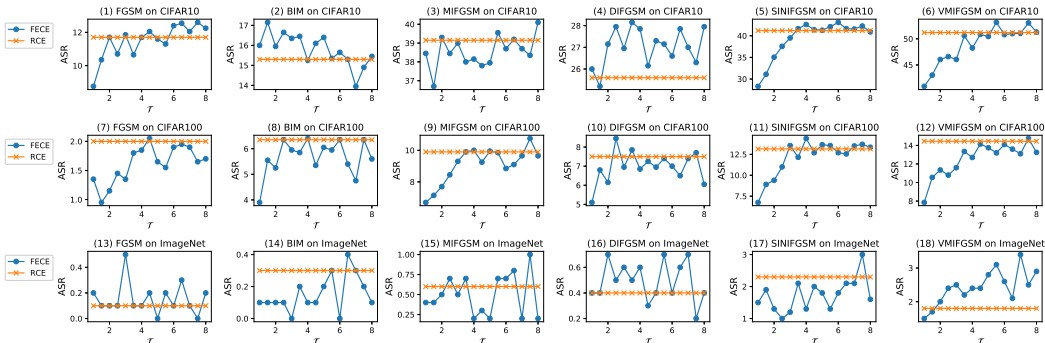

Figure 13: The targeted attack success rates (%) of ResNet50 to VGG16 using various transfer attacks for CIFAR10/100 and ImageNet when varying the size of $\mathcal{T}$ ($\mathcal{T} > 1$) in the FECE loss. Note that $\mathcal{K}$ in the FECE loss is 1.

