# OpenReview forum: "Enhancing the Transferability of Adversarial Examples via a Few Queries and Fuzzy Domain Eliminating"
_ICLR.cc/2023/Conference — Submitted to ICLR 2023_

### Official Review · Reviewer_g4Mo · 2022-10-23

**Confidence:** 5
**Correctness:** 2
**Technical Novelty And Significance:** 2
**Empirical Novelty And Significance:** 2
**Recommendation:** 1

**Clarity, Quality, Novelty And Reproducibility:**

The idea of this work is clear. However, the theoretical novelty is limited and experimental settings are problematic.


**Strength And Weaknesses:**

[ Strengths]
+ The idea of this work is easy to follow.

[ Weaknesses & Suggestions to Authors]

- The contribution of the theoretical analysis seems limited. Proposition1 and proposition 2 are well-known heuristic understandings. For example, [cite 1] has analyzed the effect of gradient alignment between the source and target models. If the attacking directions of the source model and the target model are similar, adversarial transferability will be high. Thus, I think the theoretical contribution of this paper is limited.

[cite 1] Ambra Demontis, Marco Melis, Maura Pintor, Matthew Jagielski, Battista Biggio, Alina Oprea, Cristina Nita-Rotaru, and Fabio Roli. Why do adversarial attacks transfer? explaining transferability of evasion and poisoning attacks. In 28th USENIX Security Symposium USENIX Security 19), pp. 321–338, 2019.

- The theoretical analysis is not rigorous.
(1). The formulation of the proposition/theorem is not rigorous and mathematical. Take Proposition 1 for an example, the concept of “the transferability of the transfer black-box attack” is not clearly defined by the mathematical formula in the paper.
(2). The proof is not rigorous. I go through the proof and find that some key steps in the proof are ambiguous. Take Theorem 1 for an example, the theorem states that the angle between (a) the gradient of the WACE loss on the source model and (b) the gradient of the CE loss on the target model is smaller than (c) the gradient of the CE loss on the source model and (b) the gradient of the CE loss on the target model. Then, in the proof, after giving the solutions of (a)(b)(c). The proof does not continue to mathematically compare the angles between these gradients but ends with some intuitive explanations. In other words, the theorem is not mathematically proven. In this way, I suggest the authors not write the claims as theorems, but as conjectures.

- The main objective of this paper is not clear. The title claims that this paper focuses on the transferability of adversarial examples, while the introduction indicates that the authors aim to improve the black-box attack. If the focus is adversarial transferability, the information of the target model is usually unknown in transfer-based attacks. If the focus is the black-box attack, then the comparison with other black-box methods is insufficient (see the following concerns about experiments).

- Experiments are problematic.
(1) Comparisons with the transfer-based attack are not fair. The proposed method accesses queries of the target model during attacking, while other transfer-based methods have no information about the target model. Using information of target models to guide the attack can obviously enhance the transferability. Thus, comparisons between the proposed method and other transfer-based methods are unfair. A fair comparison requires at least two attackers to access the same information.
(2) Comparisons with the query-based attack are not sufficient. Only two query-based attacks are chosen as baseline methods for comparisons, which is not sufficient to demonstrate the effectiveness of the proposed method.
(3) Baseline methods combining transfer-based and query-based attacks are required.
Since the proposed method combines transfer-based and query-based attacks, there should be baseline methods also combining these two kinds of attacks, for fair comparisons. For example, a naïve baseline method can be designed as the linear interpolation of perturbations generated with transfer-based and perturbations generated with query-based attacks. Moreover, transferable prior-based query attacks [cite 2, 3] should be compared.
(4) Experimental results cannot demonstrate the correctness of the theoretical analysis. In the analysis of fuzzy domain elimination, it is claimed (such as in Proposition 3, 4) that the fuzzy domain can be partly eliminated by setting the temperature scaling factor greater than 1. However, as Table 30 shows, the optimal temperature scaling factor in many settings is 1. In this way, I doubt that the experimental results may be better if the temperature scaling factor is allowed to be less than 1, which contradicts the proposed theoretical analysis.
(5) Some results of the proposed methods are not reported in the table. For example, the results of WACE and WFCE in the Q-FGSM in Table 2 are not reported. Therefore, I doubt the effectiveness of the proposed methods in these cases.

[cite 2] Shuyu Cheng, Yinpeng Dong, Tianyu Pang, Hang Su, and Jun Zhu. Improving black-box adversarial attacks with a transfer-based prior. In Advances in Neural Information Processing Systems 32: Annual Conference on Neural Information Processing Systems 2019, NeurIPS 2019, December 8-14, 2019, Vancouver, BC, Canada, pp. 10932–10942, 2019

[cite 3] Jiancheng Yang, Yangzhou Jiang, Xiaoyang Huang, Bingbing Ni, and Chenglong Zhao. Learning black-box attackers with transferable priors and query feedback. In Advances in Neural Information Processing Systems 33: Annual Conference on Neural Information Processing Systems 2020, NeurIPS 2020, December 6-12, 2020, virtual, 2020.


[ Minors]
- This main paper is not self-contained. All experimental results are put in the appendix, which makes the main paper not self-contained.


**Summary Of The Paper:**

This work proposes two methods to enhance adversarial transferability. The first method is called the query prior-based method, which enhances adversarial transferability by using a few queries of the target model. The second method uses temperature scaling and fuzzy scaling to scale the network output before SoftMax operation, in order to eliminate the fuzzy domain defined in the paper.

**Summary Of The Review:**

The theoretical novelty is limited, and experimental settings are problematic. Comparisons with the transfer-based attack are not fair, and comparisons with the query-based attack are not sufficient. Based on these, I tend to reject this paper.

---

### Official Review · Reviewer_XmMZ · 2022-10-24

**Confidence:** 3
**Correctness:** 4
**Technical Novelty And Significance:** 4
**Empirical Novelty And Significance:** Not applicable
**Recommendation:** 5

**Clarity, Quality, Novelty And Reproducibility:**

The authors’ contributions look to be novel. However, the paper has some issues with the writing style, especially with its verbosity and material organization. The authors definitely have made many contributions to the field; however, concisely writing the findings is also their responsibility.

**Strength And Weaknesses:**

[[Strengths]]
1. The suggested idea of gradient angle provides valuable insights into attack transferability.
2. The number of experiments is ample enough to support the power of the suggested method.

[[Weaknesses]]
1. The writing style is unnecessarily lengthy, making the paper harder to read. I suggest rewriting some statements to be more concise. Let me provide some examples. First, the authors defined the term gradient angle for simplicity, but Theorem 1 uses the phrase ‘the angle between A and B’ multiple times with complicated parameters. Defining a simpler notation for the gradient angle would shorten the statement. Also, Definitions 3 and 4 overlap a lot in their sentences and notations, meaning the authors can just describe what the readers should make replacements to get Definition 4 from Definition 3.
2. The paper writing is quite disorganized. For example, some jargons or Greek symbols appear before they define them.
3. The authors keep all the tables in the Appendix and briefly describe the results in the main body. If there are too many experimental results, put a few representative results worth highlighting in the main body, then refer to the Appendix for more results.
4. It is better to separate Appendix A into two sections, one for formal proofs and the other for empirical verification of the statements. (Just demonstrating from experiments is not considered valid proof.)


**Summary Of The Paper:**

This paper is mainly about enhancing the power of black-box attacks by delving into phenomena regarding the transferability of adversarial examples. The authors’ approach is twofold: inducing smaller gradient angles and removing fuzzy domains. The authors defined new loss terms (WACE and FECE), then combined them to design their strongest attack. Experimental results are briefly described in the paper. (All the details are in the Appendix.)

**Summary Of The Review:**

I believe the paper achieved nontrivial contributions to the field. First, the concept of gradient angle seems a novel and valuable idea for understanding attack transferability. Second, the concept of fuzzy domain also makes sense, and it explains the performance gap between white-box attacks and black-box attacks.
However, The paper writing is far from the standard and could be a reason for rejection. The authors put no details about the experiments in the main body, then referred to tables in the Appendix. Doing this even seems that the authors are exploiting the unlimited Appendix page limit. As a reviewer, I believe this is out of a standard paper writing practice, so I cannot score the recommendation above the acceptance threshold. (If a better reason is needed, I don’t see any detailed experimental result in the main body, and reviewers are not obliged to read Appendix to check it further.) The authors should choose the essential parts carefully and showcase them within the 10-page limit.

---

### Official Review · Reviewer_19nG · 2022-10-26

**Confidence:** 2
**Correctness:** 2
**Technical Novelty And Significance:** 2
**Empirical Novelty And Significance:** 2
**Recommendation:** 3

**Clarity, Quality, Novelty And Reproducibility:**

There are severe issues with the paper writing.
- All the experiment results and the proof are in the appendix, making the main part of the paper incomplete.
- The Empirical proof seems computer generated and redundant. And it is just evidence, not proof.



**Strength And Weaknesses:**

Strength:

- This paper tries to improve the transferability of adversarial examples in both theoretical and empirical perspectives.

Weaknesses:

- In Eq (10), is multiplying w_i necessary?
- The proof of Theorem 1 seems incorrect. (I could be wrong.) The theorem says the “angle” between WACE and CE is smaller than others. However, in the proof, the angle or the cosine of the angle is not computed.
- Lemma 1 and Assumption 2 are very strong assumptions considering that neural networks are highly non-linear, and alpha may be relatively large.


**Summary Of The Paper:**

Existing blackbox adversarial attack methods still make lots of queries to create an adversarial example. This paper addresses the problem by proposing the query prior-based attacks and fuzzy domain eliminating technique to enhance adversarial example transferability. Theoretical analysis and extensive experiments demonstrate that this method could significantly improve the transferability of gradient-based adversarial attacks.


**Summary Of The Review:**

The main contribution of the paper is spread out over 43 pages, and it is beyond the commitment of a reviewer. Meanwhile, the main part does not contain any experiment results (all results are in appendix), therefore it is incomplete.

---

### Official Review · Reviewer_9Tuj · 2022-10-26

**Confidence:** 3
**Correctness:** 3
**Technical Novelty And Significance:** 3
**Empirical Novelty And Significance:** 3
**Recommendation:** 5

**Clarity, Quality, Novelty And Reproducibility:**

The presentation quality needs to improve. In fact, I need to read the abstract and introduction twice to understand what the authors want to say. They want to use information from query to improve transfer-based attack. This message is not clear.

They put all the experimental results in appendix and only mentioned what experiments they did in the main text. It is not ideal. If the authors expect others to read the appendix like the main text, please improve the quality and connections with main text. Furthermore, 43 pages are abnormally long. Although I do see some ICRL papers with over 20 pages, over 40 pages is very a lot.

The proposed method is novelty with good theoretical support. However, I don’t read the proof.


**Strength And Weaknesses:**

Strength:

The proposed methods outperformed the CE loss and RCE loss.

The proposed designs are justified theoretically.


Weaknesses:

Assumption 1 is not mathematically well defined.

The experimental results on CIFAR100 are not necessary to me.

In the experiments, authors show that proposed methods can improve the transferability of gradient-based approaches. However, many state-of-the-art methods are not purely gradient approach. Some exploit the skip link and some attacks the key features based on attribution maps. The authors are better to check last two years ICRL and NIPS. Comparing with these methods are expected. These methods are based on smoothing. Some of the techniques in the proposed methods are related to smoothing. Will it be over-smoothed by applying these state-of-the art transfer-based methods with the proposed method. In some previous cases, we have seen that combining two strong transfer attacks do not always improve the results.


**Summary Of The Paper:**

The authors exploit the properties of adversarial examples to design weighted augmented CE loss to improve transferability. Then, they add two parameters, i.e., fuzzy scaling and fuzzy domain to eliminate some fuzzy region in the searching. Thus, adversarial example with higher transferability is more likely to be found. The authors provided solid justifications for their designs.

**Summary Of The Review:**

The proposed designs are justified theoretically.

Presentation and organization should be improved.

Comparsion with the state-of-the-art transfer-based attack methods is expected.

---

### Decision · Program_Chairs · 2023-01-20

**Decision:**

Reject

**Justification For Why Not Higher Score:**

See above.

**Justification For Why Not Lower Score:**

N/A

**Metareview: Summary, Strengths And Weaknesses:**

This work aimed to improve the adversarial transferability through weighted augmented cross-entropy loss and fuzzy domain eliminating technique.

The reviewers have presented several important concerns about this work, covering the motivation, theoretical analysis, writing and experiments. The authors didn't provide responses.